# Control Consistency Losses for Diffusion Bridges

**Samuel Howard** [1]   **Nikolas Nüsken** [2]   **Jakiw Pidstrigach** [1]

## Abstract

Simulating the conditioned dynamics of diffusion processes, given their initial and terminal states, is an important but challenging problem in the sciences. The difficulty is particularly pronounced for rare events, for which the unconditioned dynamics rarely reach the terminal state. In this work, we propose a novel approach for learning diffusion bridges based on a self-consistency property of the optimal control. The resulting algorithm learns the conditioned dynamics in an iterative online manner, and exhibits strong performance in a range of empirical settings without requiring differentiation through simulated trajectories. Beyond the diffusion bridge setting, we draw connections between our self-consistency framework and recent advances in the wider stochastic optimal control literature.

## 1. Introduction

**Diffusion Processes**   Diffusion processes are ubiquitous throughout mathematics, science, and, more recently, machine learning. A diffusion process $(X_t)_{t \in [0,T]}$ in $\mathbb{R}^d$ is governed by a stochastic differential equation (SDE),

$$\mathrm{d}X_t = b(t, X_t)\,\mathrm{d}t + \sigma(t, X_t)\,\mathrm{d}B_t, \qquad X_0 = x_0, \quad (1)$$

for a drift coefficient $b : [0, T] \times \mathbb{R}^d \to \mathbb{R}^d$ and diffusion coefficient $\sigma : [0, T] \times \mathbb{R}^d \to \mathbb{R}^{d \times d}$. In this work we will consider elliptic diffusions, so $\sigma\sigma^\top$ is of full-rank, and make the standard assumptions that $b, \sigma$ are Lipschitz continuous and of linear growth.

**Diffusion Bridges**   Given the SDE dynamics (1), one can use standard time-discretisation schemes such as the Euler–Maruyama method to simulate samples of the unconditioned diffusion process $X_t$. However, often in applications one is instead interested in simulating the *conditioned* dynamics of this underlying base process, conditional on terminating at a known location $X_T = x_T$ (Baudoin, 2002). Such *diffusion bridges* play an important role in a wide range of domains, including chemistry (Bolhuis et al., 2002; E & Vanden-Eijnden, 2010), finance (Elerian et al., 2001; Durham & Gallant, 2002), and shape evolution (Arnaudon et al., 2022) to name a few. Sampling diffusion bridges is typically challenging, particularly when the conditioned event in question occurs rarely under the base dynamics (1).

**Existing Approaches**   Traditional approaches for diffusion bridge sampling have commonly used MCMC or SMC-based methods (Beskos et al., 2008; Fearnhead et al., 2008). More recently, neural approximations have been used to learn the controlled drift of the conditioned dynamics, leveraging tools from score-matching (Heng et al., 2025; Baker et al., 2025) or Malliavin calculus (Pidstrigach et al., 2025). Once trained, such neural approaches enable direct simulation of samples from the diffusion bridge. However, these methods learn the neural approximation using samples from uncontrolled dynamics, limiting their success for rare conditioning events as the training samples may not adequately cover the region near the terminal state $x_T$. Motivated by this, Yang et al. (2025) learn a neurally-guided bridge by minimising the KL divergence to the true conditioned dynamics, though they require backpropagation through the simulated trajectories which can impact scalability.

**Contributions**   In this work, we derive a *self-consistency property* of the diffusion bridge solution in terms of the control drift $u(t, X_t)$ and the Jacobian process $J_{t|s}$ along the solution trajectories (Theorem 2.1, Figure 1). In combination with a *terminal condition* that enforces bridging to the known endpoint $x_T$ (Theorem 3.1, Figure 2), we train a neural-controlled diffusion process using a proposed family of *self-consistency losses* based on the above property of the solution. These fixed-point-style losses are trained in an online manner, and provide scalable training without the need to backpropagate through the simulated trajectories. We demonstrate strong empirical performance in a range of settings, competitive with state-of-the-art existing methods at a significantly reduced computational cost. Additionally, we explain how our self-consistency framework extends to wider stochastic optimal control problems, offering new perspectives and connections to recent developments.

---

[1]Department of Statistics, University of Oxford [2]Department of Mathematics, King's College London. Correspondence to: Samuel Howard <howard@stats.ox.ac.uk>.

*Proceedings of the 43$^{rd}$ International Conference on Machine Learning*, Seoul, South Korea. PMLR 306, 2026. Copyright 2026 by the author(s).

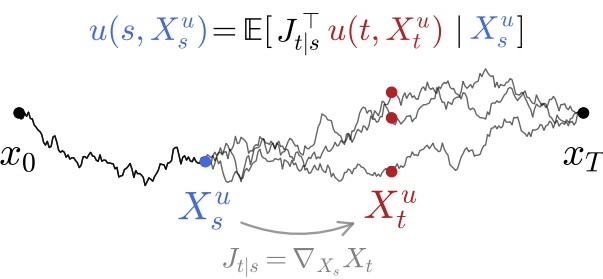

*Figure 1.* **Theorem 2.1:** The self-consistency property (SC1) relates the control value at an earlier time $s$ to its value at a later time $t$, along the solution trajectories.

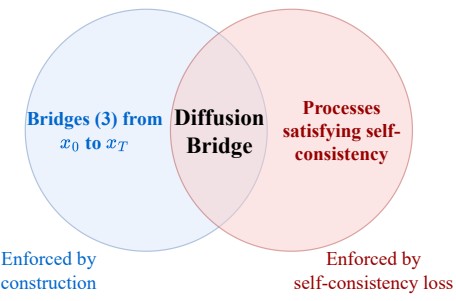

*Figure 2.* **Theorem 3.1:** The diffusion bridge is the unique process of form (3) bridging from $x_0$ to $x_T$ with $u$ of gradient form, that satisfies the self-consistency property (SC1).

## 2. Background

**Controlled Diffusions**   We begin with a more general setting, from which diffusion bridges will be recovered as a limiting case. Let us denote by $\mathbb{P}$ the path measure associated with the unconditioned reference process $X_t$ in (1), and consider a 'target' path measure $\mathbb{Q}$ induced by a reweighting density function $F(X_T)$ defined on the terminal state,

$$\frac{\mathrm{d}\mathbb{Q}}{\mathrm{d}\mathbb{P}}(X) \propto F(X_T). \tag{2}$$

It is a well-known result from stochastic analysis (Eberle, 2019, Section 1.3) that samples from $\mathbb{Q}$ can be obtained by simulating a *controlled* diffusion process $X_t^u$, with dynamics given by the following SDE started at $X_0^u = x_0$,

$$\mathrm{d}X_t^u = \big(b + (\sigma\sigma^\top)u\big)(t, X_t^u)\,\mathrm{d}t + \sigma(t, X_t^u)\,\mathrm{d}B_t. \tag{3}$$

Moreover, the *optimal control*[1] $u^*$ (that is, the control achieving (2)), is given by Doob's $h$-transform,

$$u^*(t, x) = \nabla_x \log \mathbb{E}_{\mathbb{P}}[F(X_T)|X_t = x]. \tag{4}$$

Observe that diffusion bridges are recovered from this framework by using the singular reweighting function $F(X_T) = \delta_{x_T}(X_T)$, which intuitively corresponds to 'retaining only trajectories that terminate at $x_T$'. Plugging this into (4), Doob's $h$-transform takes the more familiar form $u^*(t, x) = \nabla_x \log p(T, x_T|t, x)$, where $p(t, y|s, x)$ denotes the transition densities of the base process $X_t$.

**Self-Consistency of the Control**   In order to explain our approach, we now introduce the *Jacobian*, or *derivative*, process $J_{t|s} = \nabla_{X_s} X_t$ (Rogers & Williams, 2000, Chapter V.13), which evolves for $t > s$ according to the SDE

$$\mathrm{d}J_{t|s} = \nabla b(t, X_t) J_{t|s}\mathrm{d}t + \nabla \sigma(t, X_t) J_{t|s} \mathrm{d}B_t, \quad J_{s|s} = Id. \tag{5}$$

---
[1] Henceforth, we will use the term *optimal control* to refer to the $u$ that implements the conditioning $X_T = x_T$; the connection to optimal control more broadly will be explained in Appendix F.

Intuitively, $J_{t|s}$ measures the sensitivity of $X_t$ to perturbations to $X_s$ at an earlier time $s < t$, along trajectories of the uncontrolled process $X_t$. Importantly, as a consequence of the chain rule, the Jacobian process satisfies the semigroup property: for $s < r < t$ we have $J_{t|s} = J_{t|r}J_{r|s}$.

The expectation in (4) is over the uncontrolled dynamics $\mathbb{P}$ (that is, using trajectories $X_t$ sampled from (1)), which is not well-suited for learning the control for rare events. Using the derivative process $J_{t|s}$ (and assuming for now that $F$ is differentiable) we can perform the following calculations to turn this expectation over $\mathbb{P}$ into an expectation over $\mathbb{Q}$,

$$u^*(t, x) = \nabla_x \log \mathbb{E}_{\mathbb{P}}[F(X_T) \mid X_t = x] \tag{6}$$

$$\overset{\nabla \log f = \frac{\nabla f}{f}}{=} \frac{\nabla_x \mathbb{E}_{\mathbb{P}}[F(X_T) \mid X_t = x]}{\mathbb{E}_{\mathbb{P}}[F(X_T) \mid X_t = x]} \tag{7}$$

$$\overset{\substack{\text{interchange} \\ \nabla_x \text{ and } \mathbb{E}_{\mathbb{P}}}}{=} \frac{\mathbb{E}_{\mathbb{P}}[J_{T|t}^\top \nabla F(X_T) \mid X_t = x]}{\mathbb{E}_{\mathbb{P}}[F(X_T) \mid X_t = x]} \tag{8}$$

$$\overset{\nabla \log f = \frac{\nabla f}{f}}{=} \frac{\mathbb{E}_{\mathbb{P}}[J_{T|t}^\top F(X_T) \nabla \log F(X_T) \mid X_t = x]}{\mathbb{E}_{\mathbb{P}}[F(X_T) \mid X_t = x]} \tag{9}$$

$$\overset{\text{Lemma D.1}}{=} \mathbb{E}_{\mathbb{Q}}[J_{T|t}^\top \nabla \log F(X_T) \mid X_t = x], \tag{10}$$

In the final line, we use a conditional version of the change of measure in (2); see Lemma D.1. Interchanging the derivative and the expectation in (8) is intuitive and well-known in stochastic analysis, and is often referred to as the 'reparameterisation trick' in the machine learning literature (Kingma & Welling, 2014). It can be made rigorous using the theory of stochastic flows (Kunita, 1984; Elworthy et al., 1999), by considering the endpoint $X_T$ as a deterministic function of $X_s$ for a fixed Brownian trajectory $(B_t)_{t \in [0,T]}$; details are given in Appendix D.1. We emphasise here that the Jacobian term $J_{T|t}$ is for the *base* dynamics $\mathbb{P}$, while the expectation is now over the *target* dynamics $\mathbb{Q}$.

Unfortunately, for diffusion bridges we cannot make use of the representation in (10) directly, as this expression is not well defined for the singular reweighting $F = \delta_{x_T}$. To avoid this singular behaviour at the terminal time $T$, we can 'invert' the property at an earlier time $t < T$, using the tower property and the semigroup property of $J_{t|s}$,

$$
\begin{aligned}
u^*(s,x) &= \mathbb{E}_{\mathbb{Q}}[J_{T|s}^\top \nabla \log F(X_T)|X_s = x] \\
&= \mathbb{E}_{\mathbb{Q}}\big[\mathbb{E}_{\mathbb{Q}}[J_{t|s}^\top J_{T|t}^\top \nabla \log F(X_T)|X_{[s,t]}]\,|X_s = x\big] \\
&= \mathbb{E}_{\mathbb{Q}}\big[J_{t|s}^\top \mathbb{E}_{\mathbb{Q}}[J_{T|t}^\top \nabla \log F(X_T)|X_t]\,|X_s = x\big] \\
&= \mathbb{E}_{\mathbb{Q}}\big[J_{t|s}^\top u^*(t, X_t)|X_s = x\big].
\end{aligned}
$$

This results in the following *self-consistency* property of the optimal control $u^*$.

**Theorem 2.1** (**Self-consistency property (I)**). *For $0 < s < t < T$, the optimal control $u^*$ satisfies*

$$
u^*(s,x) = \mathbb{E}[J_{t|s}^\top u^*(t, X_t^{u^*})|X_s^{u^*} = x]. \qquad \text{(SC1)}
$$

Intuitively, this property characterises how the control at an earlier time $s$ should relate to its value at later times $t$ along the solution trajectories (see Figure 1). While (10) held only for differentiable $F$, this self-consistency property holds in more general cases, including for the singular choice $F = \delta_{x_T}$ corresponding to the diffusion bridge; we show this rigorously in Appendix D.1.

**A Family of Self-Consistency Properties** The property (SC1) is in fact an instance of a wider class of self-consistency properties, which we will detail further in Section 4 and Appendix C. It is these self-consistency properties that will form the basis for our approach. In particular, under settings with more unstable base dynamics, the Jacobian terms $J_{t|s}$ used in (SC1) can become large; we will later consider a second self-consistency property variant (SC2) which provides more stable behaviour in such scenarios.

**Combining Self-Consistency with Terminal Conditions** Theorem 2.1 establishes that self-consistency is a *necessary* property of the optimal control $u^*$, but it falls short of sufficiency; indeed, from the derivation we can see it holds for any choice of $F$. For sufficiency, we must combine self-consistency with an additional *terminal condition* that incorporates information about the solution's behaviour at the terminal time $T$. In the next section, we will show how this combined *self-consistency framework* can be used to design methods for learning diffusion bridge dynamics.

**Connections to SOC Methods** The self-consistency framework outlined above also provides a class of methods for solving more general Stochastic Optimal Control (SOC) problems. We remark that the recently introduced *adjoint*

*matching* objective (Domingo-Enrich et al., 2025), which has proven highly effective for fine-tuning flow-based generative models, can be recovered from this self-consistency framework by taking $t = T$ in (SC1) and imposing the terminal condition $u(T, x) = \nabla \log F(x)$. The adjoint matching objective was originally derived through the adjoint equation and removing terms that are zero at the solution; here, we see that it arises naturally from the derivation above. We comment more on connections to adjoint matching and related stochastic optimal control methods in Appendix F.

## 3. Control Consistency Losses for Diffusion Bridges

In this section, we utilise the self-consistency property (SC1) to introduce a novel methodology for learning diffusion bridges. In light of the discussion above, we must incorporate additional information regarding the known behaviour at terminal time $T$. The following theorem shows that the diffusion bridge control can be characterised as the unique gradient-form control that forces the dynamics to end at $x_T$, while satisfying the self-consistency property.

**Theorem 3.1.** *Consider the controlled dynamics* (3) *with control drift* $u : [0, T] \times \mathbb{R}^d \to \mathbb{R}^d$. *Suppose that:*

  *(i) $u$ satisfies the self-consistency property* (SC1)*;*

  *(ii) $u$ is such that it forces the controlled dynamics to terminate at $X_T^u = x_T$;*

  *(iii) $u$ is of gradient form for some time $t$, (that is, $u(t, \cdot) = \nabla \phi(\cdot)$ for some scalar function $\phi$).*

*Then the controlled process $X_t^u$ is the diffusion bridge.*

The proof of the theorem is provided in Appendix D. We remark that the gradient-form condition *(iii)* is a technical requirement to rule out rotational components in the control drift $u$. The true control drift is of gradient form, however it could be possible for a self-consistent $u$ to include rotational terms. Enforcing the gradient condition at a single time $t$ is sufficient to prevent such terms arising.

Theorem 3.1 provides the theoretical justification for our proposed approach. Namely, we propose to (see Figure 2):

  • optimise for the *self-consistency property* using a fixed-point style self-consistency loss.

  • enforce the *bridging* and *gradient-form* properties by construction.

Below, we outline how these two aspects can be implemented to give a practical algorithm.

## 3.1. Self-Consistency Losses

To encourage satisfying the self-consistency property (SC1) we wish to minimise the expression

$$\mathbb{E}\big[\|u(s, X_s^u) - \mathbb{E}[J_{t|s}^\top u(t, X_t^u)|X_s^u]\|^2\big]$$
$$= \mathbb{E}\big[\|u(s, X_s^u) - J_{t|s}^\top u(t, X_t^u)\|^2\big] + C, \qquad (11)$$

where $C$ is a constant independent of $u$. There are two dependencies on $u$ in (11): through $u$ itself, but also implicitly through the simulated trajectories $X_t^u$. We thus propose to train using a *fixed-point* style loss, akin to similar fixed-point losses used in Domingo-Enrich et al. (2025). Namely, we regress the control values $u(s, X_s^u)$ towards the current estimates of the targets $J_{t|s}^\top \bar{u}(t, X_t^{\bar{u}})$, obtained using samples of the current controlled process $X_t^{\bar{u}}$; here, $\bar{u}$ denotes the stopgrad operator, so that the regression targets are temporarily fixed during the optimisation. Importantly, the resulting iterative algorithm does not require propagation of the gradients through the simulated trajectories, which provides significant scalability benefits.

Note that the relation (SC1) holds for any $s < t$; one can therefore integrate over the values of $t$ according to a chosen weighting schedule $\alpha_t$, which governs how strongly the self-consistency property is enforced relative to different future times. This results in the following family of training objectives,

$$\mathcal{L}(\theta) = \mathop{\mathbb{E}}_{s \sim \mathbf{U}_{[0,T]}} \left[\left\|u_\theta(s, X_s^{\bar{\theta}}) - \mathcal{U}_s\right\|^2\right], \qquad (12)$$

where $\mathcal{U}_s := \frac{1}{A_s} \int_s^T \alpha_t J_{t|s}^\top u_{\bar{\theta}}(t, X_t^{\bar{\theta}}) \, \mathrm{d}t. \qquad (13)$

Here, we use the short-hand notation $X_t^\theta := X_t^{u_\theta}$, $\bar{\theta}$ denotes the stopgrad operator as before, $\alpha_t$ can be any weighting function (see Section 4 for a discussion), and $A_s = \int_s^T \alpha_t \, \mathrm{d}t$ is a normalisation factor.

**Solving for Training Targets** We need to compute the training targets $\mathcal{U}_s$ in (13), using trajectories simulated with the current control $u_\theta$. In order to implement this efficiently, we can solve for a reverse SDE backwards in time along the simulated paths, started from the known final control $\mathcal{U}_{T-\delta t} = u(T - \delta t, X_{T-\delta t})$ (see Section 3.2; this is similar to Pidstrigach et al. (2025, Appendix A), Li et al. (2020); Domingo-Enrich et al. (2025)). To do so, we write the target $\mathcal{U}_s$ in terms of the neighbouring target $\mathcal{U}_{s+\delta t}$ by utilising the semigroup property of the Jacobian terms $J_{t|s}$ (see Appendix B.2), giving the recursive formula

$$\mathcal{U}_s = \frac{1}{A_s} \int_s^T \alpha_t J_{t|s}^\top u(t, X_t) \, \mathrm{d}t$$
$$\approx J_{s+\delta t|s}^\top \left[\frac{A_{s+\delta t}}{A_s}\mathcal{U}_{s+\delta t} + (\delta t)\frac{\alpha_{s+\delta t}}{A_s}u(s + \delta t, X_{s+\delta t})\right]$$
$$= J_{s+\delta t|s}^\top \left[\lambda_s \mathcal{U}_{s+\delta t} + (1 - \lambda_s)u(s + \delta t, X_{s+\delta t})\right], \quad (14)$$

where in the final line we set $\lambda_s = \frac{A_{s+\delta t}}{A_s}$. Each target $\mathcal{U}_s$ is constructed as a weighted combination of the 'running' next-step target $\mathcal{U}_{s+\delta t}$ and the next-step control $u(s + \delta t, X_{s+\delta t})$, before transforming with the single-step Jacobian $J_{s+\delta t|s} \approx \mathrm{Id} + \nabla b(s, X_s)(\delta t) + \nabla \sigma(s, X_s) \cdot \delta B_s$ (approximated with a time discretisation of (5)). This procedure can be carried out using vector-Jacobian products, meaning that the training targets $\mathcal{U}_s$ can be computed in an efficient and scalable manner without the need for evaluating the Jacobian matrices explicitly.

## 3.2. Parameterisation of Bridging Processes

In light of Theorem 3.1 we must also enforce the bridging and gradient-form properties from conditions *(ii)* and *(iii)*. We will do so by construction, through an appropriate choice of neural parameterisation and by leveraging knowledge of the endpoint $x_T$. In this section, we consider neural parameterisations of the *full* drift $f = b + (\sigma\sigma^\top)u$ of the controlled process; as $\sigma\sigma^\top$ is invertible by assumption, we can compute the corresponding control $u_\theta$ used in the training objective (12).

**Bridging Property** Since the termination point is known, in the final discretisation step we can jump directly to $x_T$. By doing so, we directly enforce the bridging property, and the control function $u(T - \delta t, \cdot)$ at the penultimate timestep is completely determined by construction. This provides the required terminal information, which is then propagated backwards by the self-consistency loss (12) to learn the control at the earlier timesteps.

**Choice of Neural Parameterisation** There is considerable freedom in the drift parameterisation; the exact form can be chosen by the user and may be application-dependent. A simple and natural choice that we found to perform well in practice is to parameterise the drift as

$$f_\theta(t, X_t) = \frac{x_T - X_t}{T - t} + \sigma(t, X_t)\, \eta_\theta(t, X_t), \qquad (15)$$

where $\eta$ is a neurally-parameterised vector field. The first term is the drift of a Brownian bridge which guides the controlled process towards the terminal point $x_T$, and the second is a neural 'adjustment' that we learn. Proposition B.1 in Appendix B.1 shows that the singularity of the optimal drift $u^*$ is fully captured by $\frac{x_T - X_t}{T - t}$; the learned correction $\eta_\theta$ is locally bounded and Lipschitz, and thus amenable to approximation by neural networks.

In many applications, the reference drift $b$ is of gradient form and the diffusion term $\sigma$ is independent of $x$. In such cases, jumping to the endpoint at the final step enforces the gradient-form property of $u$ at time $T - \delta t$, satisfying the requirements of Theorem 3.1. We often found it beneficial to enforce the gradient form of $u$ for all $t$ directly, by taking

**Alg. 1** Control Consistency Diffusion Bridge (CCDB)

**Input:** Self-consistency property (e.g. (SC1) or (SC2)), drift parameterisation $f_\theta = b + (\sigma\sigma^\top)u_\theta$, number of iterations $N$, batch size $B$, weighting schedule $\alpha$.

**for** $n = 0$ **to** $N - 1$ **do**

**Path simulation:** Simulate $B$ paths of $(X_t^\theta)_{t \in [0,T]}$ according to the controlled dynamics (3), using the current control $u_\theta$;

**Target computation:** For the chosen self-consistency property, compute training targets $\mathcal{U}_s$ backwards along the obtained paths (using e.g. (14) or (55));

**Parameter update:** Perform gradient step on $\theta$ using the self-consistency loss (12).

**end for**

$\eta = \nabla\psi_\theta$ for a scalar-valued neural network $\psi$. When the true drift $b$ is not of gradient form, one can instead ensure $u$ is of gradient-form by including the base drift in the parameterisation. One can also make suitable adjustments to cover other cases; we provide an extended discussion of parameterisation choice in Appendix B.1.

**Algorithm** The resulting procedure, which we term *Control Consistency Diffusion Bridge* (CCDB), is outlined in Algorithm 1. We provide additional implementation details in the next section and in Appendix B.

## 4. A General Class of Self-Consistency Losses

The presentation so far provides a clean and intuitive self-consistency property (SC1) that can be used to learn diffusion bridges, and in Section 5 we will see that the resulting algorithm often performs extremely well in practice. Here, we shed light on potential issues that can arise in certain settings and present strategies to mitigate them, including a generalisation of the self-consistency property.

**Behaviour of the Jacobian Process** Recall that the Jacobian process $J_{t|s}$ evolves as

$$\mathrm{d}J_{t|s} = \nabla b(t, X_t)J_{t|s}\mathrm{d}t + \nabla\sigma(t, X_t)J_{t|s}\mathrm{d}B_t, \quad J_{s|s} = \mathrm{Id}.$$

Note that the increments depend on the current value of $J_{t|s}$. In particular, when $\nabla b$ or $\nabla\sigma$ have positive eigenvalues then the value will increase along those directions. Compounding of such changes over time can lead to explosion of the value of $J_{t|s}$ and subsequently in the targets $J_{t|s}^\top u(t, X_t^u)$, potentially causing numerical issues during training.

To illustrate this effect conceptually, consider overdamped Langevin dynamics $\mathrm{d}X_t = -\nabla U(X_t)\mathrm{d}t + \sigma\mathrm{d}B_t$. The process traverses the scalar potential $U$ randomly, with a tendency to move 'downhill'. In such a setting, positive eigenvalues occur passing over a peak or ridge. Recalling

that $J_{t|s} = \nabla_{X_s}X_t$, this is indeed intuitive—a perturbation of $X_s$ within a valley has little effect on the later position $X_t$, but a perturbation made while crossing a peak can compound to cause large deviations in the subsequent trajectory.

**Role of $\alpha$ Schedule** Recall that the self-consistency loss (12) depends on a choice of weighting function $\alpha$. Intuitively, the function $\alpha$ determines how 'far into the future' the algorithm looks when trying to enforce the self-consistency property. The $\alpha$ schedule therefore provides a trade-off; looking far into the future provides better propagation of the known information from the final step at $T - \delta t$, while a shortened schedule can mitigate growth in the training targets, which can help stabilise training in cases where the Jacobian terms would become large. Following Pidstrigach et al. (2025), we consider three types of $\alpha$ schedule, which differ in how far into the future the algorithm attempts to enforce self-consistency:

- *Next-step prediction*: Fits to the training target from only the next discretisation step, so $\mathcal{U}_s = J_{s+\delta t|s}^\top u(s+\delta t, X_{s+\delta t}^u)$. This corresponds to $\lambda = 0.0$ in (14).

- *Average prediction*: Fits to the average over the subsequent times $t > s$, so $\mathcal{U}_s = \frac{1}{T-s}\int_s^T J_{t|s}^\top u(t, X_t^u)\mathrm{d}t$. This corresponds to a constant $\alpha$ schedule.

- *Final-step prediction*: Fits to the training target from only the final step, so $\mathcal{U}_s = J_{T-\delta t|s}^\top u(T - \delta t, X_{T-\delta t}^u)$. This corresponds to $\lambda = 1.0$ in (14).

Generally, we found the average schedule to strike a good balance across a variety of experimental settings, so we use this in our experiments unless otherwise stated. We provide an empirical comparison of $\alpha$-schedules in Appendix E.7.

**Generalising the Self-Consistency Property** We now present an alternative mechanism to combat troublesome behaviour in the Jacobian $J_{t|s}$, by deriving a generalisation of our self-consistency property that retains the same theoretical guarantees.

Recall from the derivation in (8) that the Jacobian $J_{t|s}$ arises from taking the derivative through the $\mathbb{P}$-expectation. We can instead consider a change of measure before performing this operation. Let $\tilde{\mathbb{P}}$ be an 'auxiliary' measure associated to the process $\mathrm{d}\tilde{X}_t = (b+\tilde{b})(t, \tilde{X}_t)\mathrm{d}t + \sigma(t, \tilde{X}_t)\mathrm{d}B_t$, in which we include an additional drift term $\tilde{b}$. In the derivation from Section 2, we can instead perform the change of measure

$$\nabla_x \mathbb{E}_{\mathbb{P}}[F(X_T)|X_t = x] = \nabla_x \mathbb{E}_{\tilde{\mathbb{P}}}[F(X_T)\frac{\mathrm{d}\mathbb{P}}{\mathrm{d}\tilde{\mathbb{P}}}|X_t = x],$$

where the Radon-Nikodym derivative is given by the Girsanov weights $\frac{\mathrm{d}\mathbb{P}}{\mathrm{d}\tilde{\mathbb{P}}} = \exp(-\int_t^T \sigma^{-1}\tilde{b}(r, X_r)^\top \mathrm{d}B_r - \int_t^T \frac{(\sigma\sigma^\top)^{-1}}{2}\|\tilde{b}(r, X_r)\|^2 \mathrm{d}r)$ (Øksendal, 2003, Theorem

8.6.8). Continuing the calculation as in Section 2, we can obtain a *generalised self-consistency property* (SC-gen) (given in Appendix) which instead includes Jacobian terms $\tilde{J}_{t|s}$ for the auxiliary process $\tilde{X}$ rather than the base process $X$, while also picking up additional 'running cost' terms.

A natural choice of auxiliary process is to take $\tilde{b} = -b$, so that $\tilde{X}$ has no drift term—when the diffusion coefficient is independent of $x$, this will mean $\tilde{J}_{t|s} = \text{Id}$. In this case, (SC-gen) simplifies to the following property, which we find gives a significant improvement in settings where the growth of the base Jacobian $J_{t|s}$ would cause numerical issues.

**Theorem 4.1** (**Self-consistency property (II)**). *For* $0 < s < t < T$, *the control drift* $u^*$ *of the solution satisfies,*

$$u^*(s,x) = \mathbb{E}\Big[ \int_s^t \nabla b(r, X_r^{u^*})^\top u^*(r, X_r^{u^*})\, dr$$
$$+ u^*(t, X_t^{u^*}) \,|\, X_s^{u^*} = x \Big]. \quad \text{(SC2)}$$

Importantly, we retain the same theoretical guarantees when using such instances of the generalised self-consistency property (SC-gen) as we do in the previous case.

**Theorem 4.2.** *The result of Th. 2.1 holds with the generalised self-consistency property* (SC-gen) *in place of* (SC1).

A complete derivation of the generalised self-consistency property and corresponding proofs are given in Appendix D. In Appendix F we also discuss connections with recent developments in the stochastic optimal control literature, and remark that similar variations might be useful for wider SOC problems; we leave investigating this to future work.

**Algorithm** As in Section 3, the generalised self-consistency properties can be used to construct a fixed-point objective used for training. As they hold for any $s < t$, we can again integrate over $t$ according to a weighting schedule $\alpha_t$. In the case of (SC2), this gives the training objectives

$$\mathcal{L}(\theta) = \mathop{\mathbb{E}}_{s \sim \mathbf{U}_{[0,T]}} \left[ \left\| u_\theta(s, X_s^{\bar{\theta}}) - \mathcal{U}_s \right\|^2 \right], \quad (16)$$

$$\text{for } \mathcal{U}_s := \frac{1}{A_s} \int_s^T \alpha_t\, \Phi_{s,t}\, dt, \quad (17)$$

$$\text{where } \Phi_{s,t} = \int_s^t \nabla b(r, X_r^{u^*})^\top u^*(r, X_r^{u^*})\, dr$$
$$+ u^*(t, X_t^{u^*}). \quad (18)$$

Similarly to before, the training targets $\mathcal{U}_s$ can be computed via a recursive formula backwards along the simulated trajectories; for full details, see Appendix C.

**Connection to the $\alpha$-weighted objective** A natural question is how the loss in (17) relates to the family of $\alpha$-weighted losses (13). In fact, one can choose a *matrix-valued* $\alpha$ to recover (12); see Appendix D.4 for details.

## 5. Experiments

To assess the performance of our approach, we consider applying our method CCDB with the two self-consistency properties (SC1) and (SC2), and use the average $\alpha$-schedule (results for other $\alpha$-schedules are reported in Appendix E.7). We first consider four experimental settings commonly used in the diffusion bridge literature, before finally testing on more challenging examples from computational chemistry. When the ground-truth bridge $\mathbb{P}^*$ is available, we report the value of $\text{KL}(\mathbb{P}^* \| \mathbb{P}^\theta)$. When it is not, we instead report $\text{KL}(\mathbb{P}^\theta \| \mathbb{P})$ relative to the base process; amongst processes that bridge between the starting and terminal states, the diffusion bridge is the one that minimises this divergence.

We focus our comparisons on the recent neural-guided method of Yang et al. (2025) (NGDB); this is the most directly comparable method, as it similarly learns a neural control of the drift in an online manner, enabling applications for rare event scenarios in which the alternative neural methods would struggle. Note that we do not aim to necessarily beat the NGDB method, as this method differentiates through the trajectory which provides an advantage. We use NGDB here as a strong baseline and aim to achieve comparable results at a lower computational cost, since our approach does not differentiate through the trajectory. In Table 2, we report runtimes of our algorithm compared to NGDB, and observe an approximate threefold improvement in training speed.

We also include comparisons with the methods *score-diffusion bridge* (SDB) (Heng et al., 2025), *forward-bridge* (FB) (Baker et al., 2025), and BEL (Pidstrigach et al., 2025), which learn from uncontrolled simulations. As expected, we find these approaches generally perform well when the terminal state $x_T$ occurs frequently under the base dynamics, but struggle otherwise, consistent with findings in Yang et al. (2025).

Full details of each experimental setting, additional results, and further discussion of findings are given in Appendix E.

### 5.1. Ornstein-Uhlenbeck Bridges

We first consider sampling bridges of Ornstein-Uhlenbeck processes $dX_t = -\alpha X_t dt + \sigma dB_t$. This is a standard setting in the literature for quantitatively evaluating performance (Heng et al., 2025; Baker et al., 2025; Yang et al., 2025), as the true control $u^*$ is available in closed-form.

In Figure 3, we compare performance against the NGDB method in various OU bridge settings. We include experimental details and a full discussion of results in Appendix E. In Figure 3a, we report how the KL divergence relative to the ground truth evolves during training, and see that our method displays improved convergence to the solution in this example. In Figures 3b and 3c, we include plots to illus-

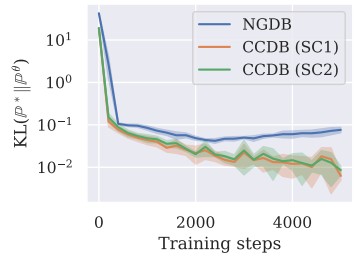
*(a)* KL to solution during training.

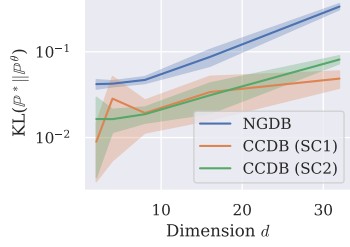
*(b)* Effect of dimension, for $\alpha = 2$

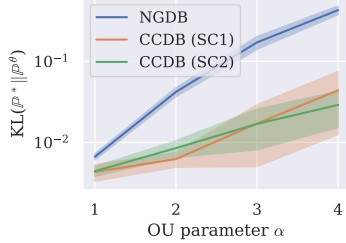
*(c)* Effect of OU parameter, $d = 2$.

*Figure 3.* Comparison of our proposed approach CCDB (using self-consistency properties (SC1) and (SC2)), and Neural Guided Diffusion Bridge (NGDB) (Yang et al., 2025), in the Ornstein-Uhlenbeck bridge experiment (mean±std, over 5 runs).

*Table 1.* Quantitative results for experiments (mean±std over 5 runs). Methods that perform strongly are indicated by •, those with reasonable performance by ▲, and methods that fail to converge to the correct solution by ▪. We note the result marked with an asterisk* is likely unreliable as it gave an extremely large control towards the endpoint, causing the integration to underestimate the correct value.

| | CIR model, $\text{KL}(\mathbb{P}^*\|\mathbb{P}^\theta) \downarrow$ | Double Well, $v = 3.0$, $\text{KL}(\mathbb{P}^*\|\mathbb{P}^\theta) \downarrow$ | Cell Diffusion, $\text{KL}(\mathbb{P}^\theta\|\mathbb{P}) \downarrow$ | | |
| --- | --- | --- | --- | --- | --- |
| | | | Normal | Rare | Multimodal |
| **CCDB (Ours)**, using (SC1) | 0.023±0.010 • | 0.148±0.073 •/▲ | 6.53±0.14 • | 66.1±0.9 • | 792.8±0.9 •/▲ |
| **CCDB (Ours)**, using (SC2) | 0.055±0.009 •/▲ | 0.041±0.013 • | 6.53±0.17 • | 66.0±0.5 • | 792.4±1.0 •/▲ |
| NGDB (Yang et al., 2025) | 0.051±0.012 •/▲ | 0.098±0.047 •/▲ | 6.48±0.15 • | 65.5±0.6 • | 783.4±0.8 • |
| SDB (Heng et al., 2025) | 0.104±0.011 ▲ | 0.229±0.024 ▲ | 5.56±0.39* | 75.0±5.1 ▲ | 1845±230 ▪ |
| FB (Baker et al., 2025) | 0.074±0.011 ▲ | - | 6.51±0.14 • | 237±98 ▪ | 4836±954 ▪ |
| BEL (Pidstrigach et al., 2025) | 0.038±0.009 • | 0.134±0.051 ▲ | 6.90±0.11 •/▲ | 94.3±2.2 ▲ | 885.1±6.0 ▲ |

trate the dependence on dimension and the OU parameter $\alpha$. We found that SDB, FB, and BEL did not learn the bridges successfully in these experiments, which is unsurprising given that they involve rare events under the base dynamics.

## 5.2. Cox–Ingersoll–Ross model

We now consider computing bridges for the Cox-Ingersoll-Ross (CIR) model (Cox et al., 1985), which is used in mathematical finance to model the evolution of interest rates. The CIR process evolves according to the SDE $dX_t = a(b - X_t)dt + \varepsilon\sqrt{X_t}dB_t$. We consider this example as it allows us to verify our method on an SDE with a *spatially-dependent* diffusion coefficient $\sigma(t, x) = \varepsilon\sqrt{x}$, and also because the ground-truth is again tractable. We find that using (SC1) in our method performs slightly better than using (SC2), which in turn performs comparably with NGDB. The methods SDB, FB and BEL also learn well in this example as the terminal state is well-covered by the base dynamics, though SDB and FB are still outperformed by the methods that learn from controlled dynamics.

## 5.3. Double-Well Potential

Next, we consider overdamped Langevin dynamics $dX_t = -\nabla U(X_t)dt + \sigma dB_t$ using a classical double-well potential $U(x) = v(x^2 - 1)^2$, and aim to learn the bridge

from $x_0 = 1$ to $x_1 = -1$; this setting was previously considered in Pidstrigach et al. (2025). In light of the discussion in Section 4, this presents a potentially more troublesome problem setting for our method, as the concave peak between the two metastable states can lead to large Jacobian terms. We consider setting $v = 3.0$ which results in a strong peak between the states, and again report $\text{KL}(\mathbb{P}^*\|\mathbb{P}^\theta)$ relative to the ground-truth, which in 1 dimension can be approximated by numerically solving the backwards Kolmogorov equation.

In this example, using (SC2) improves substantially over (SC1), reflecting the discussion in Section 4 regarding large Jacobians. This effect is exacerbated by increased well barrier height $v$; see Appendix E.3 for a comparison for different barrier heights. The SDB method was significantly outperformed by the approaches that used controlled dynamics, but performed reasonably for wells with lower heights (see Appendix E.3). We found that FB failed to learn reasonable dynamics in this setting. We also remark that using 'sticking-the-landing' adjustments (Roeder et al., 2017; Domingo-Enrich, 2024) provided substantial further improvements for our method in this setting; see Appendix E.8.

## 5.4. Cell Diffusion Example

For an example where the reference drift $b$ is not of gradient form, we consider the cell differentiation model of Wang

*Table 2.* Comparison of runtimes for our method and NGDB.

|  | Ours | NGDB |
|---|---|---|
| OU ($\alpha = 2.0, d = 2$) | 230s | 800s |
| CIR | 100s | 280s |
| Double Well ($v = 3.0$) | 180s | 520s |
| Cell Diffusion | 60s | 170s |
| Müller-Brown ($2d$) | 350s | 1460s |
| Müller-Brown ($100d$) | 1080s | 5300s |

*Table 3.* Comparison of results for the Müller–Brown experiment. The values for $\mathrm{KL}(\mathbb{P}^\theta \| \mathbb{P})$ show mean±std over 5 runs, while the Max Energy values report the average mean and average std over the 5 runs.

|  | $\mathrm{KL}(\mathbb{P}^\theta \| \mathbb{P}) \downarrow$ | Max Energy |
|---|---|---|
| **CCDB (Ours)**, using (SC2) | 27.8±1.7 | -33.6±6.1 |
| NGDB (Yang et al., 2025) | 28.6±0.4 | -35.9±4.9 |
| DL (Du et al., 2024) | 38.6±0.2 | -32.5±5.9 |

et al. (2011), which has previously been used to provide a qualitative evaluation of diffusion bridge methods in Heng et al. (2025); Baker et al. (2025); Yang et al. (2025). We consider the three settings used in Yang et al. (2025): a *normal* event, a *rare* event, and a *multi-modal* example. There is no ground-truth available in this setting, so we report $\mathrm{KL}(\mathbb{P}^\theta \| \mathbb{P})$ relative to the base process. We also visualise the obtained trajectories in Figure 6, and observe very similar results for CCDB and NGDB in each case. All methods performed similarly well in the 'normal' setting. In the rare and multimodal settings, NGDB appears to perform slightly better than CCDB, but both performed strongly and significantly outperformed the methods that use unconditioned dynamics. SDB and BEL performed reasonably in the rare setting but failed in the multimodal setting, and FB failed to learn the dynamics accurately in both settings.

### 5.5. Transition Path Sampling

So far, we have demonstrated strong performance of our method in standard experimental settings considered in the diffusion bridge literature, comparable with the strongest existing method NGDB at a significantly reduced computational cost. In the remainder, we apply our method to more challenging examples not previously considered by neural diffusion bridge methods, and aim to assess whether our algorithm remains stable in these more demanding examples.

Transition path sampling (TPS) is a fundamental problem in computational chemistry, in which one aims to simulate the conformational changes of molecules between different metastable states. We focus here on using overdamped Langevin dynamics as we consider elliptic diffusions in this work, and remark that extending our method to the more common underdamped setting is a promising direction for future work (see Appendix E.6). We emphasise that we do not aim to achieve state-of-the-art performance for TPS; there are many works devoted solely to this problem. Rather, we use this problem as a demanding test case to assess the stability of our method in a more challenging setting.

**Müller-Brown Potential** We first consider using the Müller-Brown potential, a classical benchmark for TPS methods. While only two-dimensional, the steep slopes of

the potential make this a difficult test case. We compare against NGDB, and also to the recent Doob's Lagrangian method (DL) (Du et al., 2024); this method uses a Gaussian path parameterisation and is specifically designed for TPS, making it a strong baseline. These approaches impose similar endpoint constraints to our method, making them suitable for comparison. We use the experimental setup of Du et al. (2024); for details, see Appendix E.5. In CCDB, we use (SC2) to avoid large Jacobian terms; indeed, using (SC1) generally failed to learn the correct dynamics (see Appendix E.5), consistent with the discussion in Section 4.

The results are shown in Table 3, with trajectories for our method visualised in Figure 4. CCDB and NGDB learn similar values of $\mathrm{KL}(\mathbb{P}^\theta \| \mathbb{P})$ relative to the base process, though CCDB has a slightly larger variance across runs. These values are lower than for DL, which is likely because the Gaussian path parameterisation cannot capture the true dynamics as accurately. Their parameterisation does however mean that DL can train significantly faster than CCDB and NGDB, as it does not require simulations. All of the methods learn qualitatively similar trajectories, and have similar maximum energies along the transition paths.

In Appendix E.5, we also consider a 100-dimensional Müller-Brown example, in which we augment the standard potential with quadratic potentials in the additional dimensions. We find that performance remains largely unchanged, demonstrating that problem difficulty is largely dictated by the complexity of the transition rather than dimensionality, and that our method scales well as dimension increases.

**Alanine Dipeptide** For an application to a real molecule, we consider transitioning between the $C5$ and $C7ax$ conformation states of alanine dipeptide. As we consider overdamped dynamics, our results are not directly comparable with other TPS methods which use underdamped dynamics; nevertheless, the obtained trajectories displayed in Figure 4 resemble those reported in other works. We provide additional details, comparisons with other methods, and discussion of findings in Appendix E.

**Choice of Self-Consistency Objective** The experiments shown here demonstrate strong performance of our self-

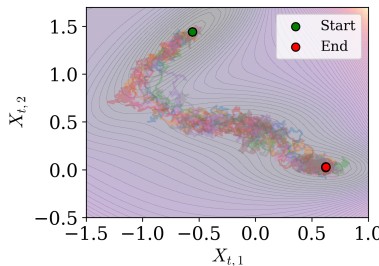 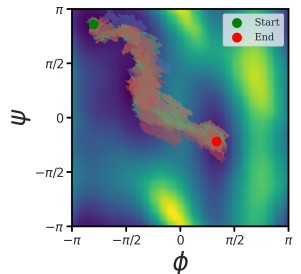 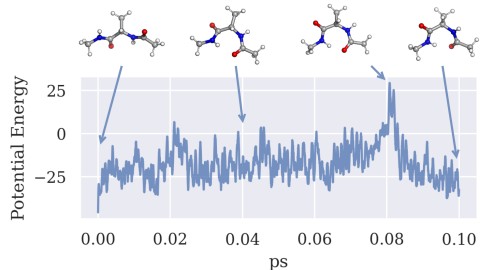

*Figure 4. (Left)* Learned trajectories for the Müller–Brown potential, for our method using (SC2). *(Middle)* Ramachandran plot showing obtained paths for alanine dipeptide. *(Right)* Energy of an individual transition path for alanine dipeptide.

consistency approach for learning diffusion bridges, but also highlight that the best choice of self-consistency property often depends on the behaviour of the Jacobian process. In stable settings without significant growth in the Jacobian terms (the OU, CIR, and cell diffusion examples), the objectives (SC1) and (SC2) perform similarly, whereas in more unstable settings with large Jacobians (double well, Müller-Brown, and Alanine Dipeptide), then (SC2) performs better than (SC1). We provide further experiments to illustrate this effect in Appendix E, which can help to guide practitioners in choosing the best objective for their problem setting.

## 6. Discussion

**Related Work**   Diffusion bridges are a well-studied problem with an extended literature spanning several decades (Clark, 1990; Delyon & Hu, 2006; Schauer et al., 2017). Traditional methods have used MCMC (Stuart et al., 2004; Beskos et al., 2008) or SMC-based approaches Lin et al. (2010). Our work contributes to the line of recent methods using neural networks to learn the control of the bridging dynamics (Heng et al., 2025; Baker et al., 2025; Pidstrigach et al., 2025; Yang et al., 2025). An extended discussion of related work is given in Appendix A.

The self-consistency framework provided in this work is also closely related to adjoint matching (Domingo-Enrich et al., 2025) and other recent developments in stochastic optimal control (Domingo-Enrich et al., 2024; Domingo-Enrich, 2024); we discuss these connections in Appendix F.

**Limitations**   Our approach requires ellipticity assumptions on the base process in order to compute the control terms $u$ in the training objective, which restricts its applicability in certain settings compared to NGDB. The fixed-point nature of our algorithm may be less stable than the KL-based objective used in NGDB, though this could be an inherent trade-off when avoiding differentiation through trajectories. We also require simulations of the current process during training, in contrast to the simulation-free method of Du et al. (2024), though their Gaussian path parameterisation means it does not learn the true dynamics in general.

**Conclusion and Future Directions**   In this work, we have presented a novel algorithm for learning diffusion bridges based on a self-consistency property of the optimal control $u^*$. Our method demonstrates strong performance across a range of diffusion bridge problems, showing competitive performance with the current strongest method at a lower computational cost. Future work could investigate extensions to hypoelliptic diffusions, and target specific applications in computational chemistry for which we report promising findings. In certain settings we observed a significant benefit to using the self-consistency property (SC2); future research could investigate to what extent similar benefits transfer to wider stochastic optimal control problems.

## Impact Statement

This paper presents work whose goal is to advance the field of Machine Learning. There are many potential societal consequences of our work, none which we feel must be specifically highlighted here.

## Acknowledgements

SH is supported by the EPSRC CDT in Modern Statistics and Statistical Machine Learning [EP/S023151/1]. JP was supported by the Engineering and Physical Sciences Research Council [EP/Y018273/1].

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

# Appendix

The Appendix is structured as follows. In Appendix A, we present an extended discussion of related work. In Appendix B, we give more details regarding the proposed method and its implementation. In Appendix C we give further details regarding the generalised form of the self-consistency property discussed in Section 4. In Appendix D we provide the proofs of the results in the paper. In Appendix E, we report experimental details and additional results. In Appendix F, we discuss connections to recent approaches in the stochastic optimal control and generative modelling literature. Finally, in Appendix G we provide the licenses and links to the assets used in the work.

## A. Related Work

Diffusion bridge simulation has been the subject of extensive study over recent decades; we highlight a selection of works from the literature here. A common approach is to use simulations from tractable dynamics, and correct using importance-sampling (Papaspiliopoulos & Roberts, 2012) or Metropolis-Hastings steps. Pedersen (1995) use the unconditional dynamics, and Clark (1990); Delyon & Hu (2006) utilise the Brownian bridge to ensure termination at the correct state. Schauer et al. (2017) consider a class of guided diffusions that includes an additional drift term while preserving tractability. We also highlight the works Stuart et al. (2004); Beskos et al. (2008) who consider Langevin-type algorithms on the space of paths, Bierkens et al. (2021) who utilise a piecewise deterministic Monte-Carlo approach, and Lin et al. (2010) who use an SMC-based approach. Often, diffusion bridge approaches are used for inferring the parameters of a discretely-observed diffusion process (Elerian et al., 2001; Roberts & Stramer, 2001; Durham & Gallant, 2002; Stramer & Yan, 2007; Golightly & Wilkinson, 2008; Bladt & Sørensen, 2014; Bladt et al., 2015; van der Meulen & Schauer, 2017).

More recently, neural approximations have been used to learn the control term $u(t,x)$. Once trained, this enables the simulation of samples from the diffusion bridge without the need for MCMC-style procedures. Heng et al. (2025) utilised the expression of the control drift in the time-reversed bridge dynamics as a score function, and trained the drift using score-matching on simulated trajectories of the unconditioned forwards process. The conditioned forwards dynamics can subsequently be learned via a time-reversal. This approach was extended in Baker et al. (2025) to learn the forwards dynamics directly, using trajectories of an adjoint process simulated backwards in time. Pidstrigach et al. (2025) uses a novel characterisation of the control based on Malliavin calculus, which is learned from simulations of the unconditioned process. Such approaches enable direct simulation of the conditioned dynamics using the learned drifts. However, as they rely on simulations of the unconditioned dynamics, they are limited in their ability to learn accurate dynamics for *rare* events, as these will hardly ever arise in the trajectories used for training. Recent work Yang et al. (2025) aims to address this by using a neural guiding drift in addition to the linear drift from Schauer et al. (2017), and optimising a KL-based objective to encourage towards the true conditioned dynamics. While exhibiting strong performance, the need for backpropagation through simulated trajectories can pose challenges for scalability. It is not clear how to modify their training objective to avoid backpropagation through the trajectories, as simply placing a stop-gradient on the samples from the trajectory will not in general converge to the correct solution. We also highlight the recent work Du et al. (2024), which introduces a variational formulation to learn a Gaussian mixture path parameterisation. Of these methods, the most related to our approach is the recent work of Pidstrigach et al. (2025), as the self-consistency property that we use can also be derived from their Malliavin calculus framework.

Finally, we highlight that the self-consistency framework that we use in this work has many connections to recent advances in the stochastic optimal control literature (Domingo-Enrich et al., 2024; 2025; Domingo-Enrich, 2024), though their focus is on optimal control for rewards with gradient information rather than on diffusion bridges. We provide an extended discussion of these connections in Appendix F.

## B. Method Details and Discussion

The presentation of our method in the main text (given in Algorithm 1) is based on the two most natural forms of the self-consistency property, namely (SC1) and (SC2). Below, we include additional implementation considerations regarding our proposed approach, and a detailed description of the general version of the algorithm using the generalised self-consistency property discussed in Section 4.

### B.1. Choice of Neural Parameterisation

Recall from the discussion in Section 3 that we use the neural parameterisation to guide towards the terminal state, and to enforce the gradient-form property. Aside from these considerations, there is considerable freedom in the choice of neural parameterisation, though it is clearly beneficial to direct the samples towards the endpoint to avoid large jumps in the final step. As discussed in Section 3, in cases with a gradient-form base drift $b$ we use the parameterisation

$$f_\theta(t, X_t) = \frac{x_T - X_t}{T - t} + \sigma(t, X_t)\, \eta_\theta(t, X_t), \tag{19}$$

for a neurally-parameterised vector field $\eta$. Recalling the result in Theorem 3.1, we only need to enforce the gradient property at a single point in time to give a theoretically-grounded algorithm. In cases with gradient-form $b$ and diffusion term $\sigma$ independent of $x$ (which covers a significant proportion of use cases), this is ensured by our construction where we directly jump to the terminal state in the final step; this enforces that the control $u(T - \delta t, \cdot)$ is of gradient-form. In practice, rotational terms may accrue if the optimisation is not performed perfectly; in many settings we found improved performance by enforcing the gradient property throughout time by setting $\eta_\theta = \nabla \psi_\theta$ for scalar-valued network $\psi_\theta$.

For base drifts $b$ not of gradient-form, we include the drift $b$ in the parameterisation,

$$f_\theta(t, X_t) = b(t, X_t) + \frac{x_T - X_t}{T - t} + \sigma(t, X_t)\, \eta_\theta(t, X_t).$$

As above, we can enforce the gradient-form property directly throughout time by setting $\eta_\theta = \nabla \psi_\theta$. If one wishes to use a more general neural parameterisation $\eta$, note that for time close to $T$ the samples are close to the terminal point $x_T$ and thus the drift $b$ can be well-approximated by its linearisation about $x_T$. Thus, jumping to the terminal state in the final step approximately enforces $u(T - \delta t, \cdot)$ to be of gradient-form similarly to above (and we note that we found such design decisions to have minimal effect in practice).

Certain choices of spatially-dependent diffusion coefficients $\sigma$ may require further modifications, and the appropriate adjustments can again be made to ensure the gradient-form conditions are satisfied in these cases too. There are also further possibilities for the parameterisation, such as incorporating the linearised guiding drift used in Mider et al. (2021); Yang et al. (2025).

**Regularity of the Residual Parameterisation**   Our choice of parameterisation above can be motivated by the fact that the residual correction to the Brownian bridge drift is in fact locally bounded and Lipschitz, and is thus amenable to approximation using neural networks. For simplicity, we present the following result in the case $\sigma \equiv 1$.

**Proposition B.1** (Boundedness and Lipschitz regularity of the bridge residual)**.** *Assume that $b$ is smooth with bounded derivatives, $b \in C_b^\infty([0, T] \times \mathbb{R}^d; \mathbb{R}^d)$, let $p(T, x_T \mid t, x)$ be the transition density of the uncontrolled process*

$$\mathrm{d}X_t = b(t, X_t)\, \mathrm{d}t + \mathrm{d}B_t, \tag{20}$$

*and fix the terminal point $x_T \in \mathbb{R}^d$. By Doob's h-transform, the optimal drift is given by*

$$u^\star(t, x; x_T) := \nabla_x \log p(T, x_T \mid t, x).$$

*Then the residual*

$$\eta^\star(t, x; x_T) := u^\star(t, x; x_T) - \frac{x_T - x}{T - t}$$

*is locally bounded and locally Lipschitz, uniformly as $t \to T$. More precisely, for every compact set $K \subset \mathbb{R}^d$, there exist constants $C < \infty$ and $\tau_0 > 0$ such that, for all $0 < T - t \le \tau_0$,*

$$\|\eta^\star(t, \cdot; x_T)\|_{L^\infty(K)} + \mathrm{Lip}_K\big(\eta^\star(t, \cdot; x_T)\big) \le C.$$

**Intuition**   For small $\tau := T - t$, the uncontrolled process (20) can be approximated by the Euler step

$$X_T \approx x + \tau\, b(t, x) + \sqrt{\tau}\, Z, \qquad Z \sim N(0, I).$$

Therefore, the transition density is approximately

$$p(T, x_T \,|\, t, x) \approx (2\pi\tau)^{-d/2} \exp\left(-\frac{|x_T - x - \tau b(t,x)|^2}{2\tau}\right).$$

Taking the logarithm and differentiating with respect to $x$ gives

$$\nabla_x \log p(T, x_T \,|\, t, x) = \frac{x_T - x}{\tau} + \mathcal{O}(1),$$

where the $\mathcal{O}(1)$ terms depend on $b$ and its derivatives, but they do not blow up as $t \to T$ if $b$ is sufficiently regular.

*Proof.* The proof makes the intuition from above precise, using heat kernel asymptotics (i.e., short time expansions of solutions to parabolic PDEs) from Azencott (1984); Aït-Sahalia (2008). For convenience, set

$$\tau := T - t, \qquad \delta := x_T - x,$$

so that the heat kernel can be written as

$$g_\tau(\delta) := (2\pi\tau)^{-d/2} \exp\left(-\frac{|\delta|^2}{2\tau}\right). \tag{21}$$

The transition density admits the expansion (Aït-Sahalia, 2008, Theorem 1)

$$\log p(T, x_T \,|\, t, x) = \log g_\tau(x_T - x) + A(t, x, x_T) + \tau R(t, x, x_T, \tau), \tag{22}$$

where the first correction term is

$$A(t, x, x_T) := \int_0^1 (x_T - x) \cdot b\big(t, x + \theta(x_T - x)\big)\, \mathrm{d}\theta, \tag{23}$$

and $R$ is bounded together with its first two spatial derivatives (locally on compact sets). In other words, the leading order contribution to the transition density is the standard heat kernel (21), and the first-order correction term has the explicit form (23).

The claim now follows by differentiating (22),

$$\nabla_x \log p(T, x_T \,|\, t, x) = \nabla_x \log g_\tau(x_T - x) + \nabla_x A(t, x, x_T) + \tau \nabla_x R(t, x, x_T, \tau),$$

and observing that the derivatives of $A$ and $R$ only contribute locally Lipschitz and locally bounded terms (because of the smoothness and boundedness assumptions on $b$). $\square$

### B.2. Solving for the training targets

Recall that we compute the training targets $\mathcal{U}_s := \frac{1}{A_s} \int_s^T \alpha_t J_{t|s}^\top u(t, X_t) \mathrm{d}t$ by solving backwards along the simulated trajectories. To do so, we can write a recursive formula for the target $\mathcal{U}_s$ in terms of the next-step target $\mathcal{U}_{s+\delta t}$. We include the full calculations below. Note that here (and in similar computations throughout the Appendix) we write $X_t$ for notational simplicity, and recall that these computations are performed using the controlled dynamics $X_t^{u_\theta}$.

$$\mathcal{U}_s = \frac{1}{A_s} \int_s^T \alpha_t J_{t|s}^\top u(t, X_t) \mathrm{d}t \tag{24}$$

$$= \frac{1}{A_s} \int_{s+\delta t}^T \alpha_t J_{t|s}^\top u(t, X_t) \mathrm{d}t + \frac{1}{A_s} \int_s^{s+\delta t} \alpha_t J_{t|s}^\top u(t, X_t) \mathrm{d}t \tag{25}$$

$$= \frac{1}{A_s} \int_{s+\delta t}^T \alpha_t (J_{t|s+\delta t} J_{s+\delta t|s})^\top u(t, X_t) \mathrm{d}t + \frac{1}{A_s} \int_s^{s+\delta t} \alpha_t J_{t|s}^\top u(t, X_t) \mathrm{d}t \tag{26}$$

$$= \frac{A_{s+\delta t}}{A_s} J_{s+\delta t|s}^\top \mathcal{U}_{s+\delta t} + \frac{1}{A_s} \int_s^{s+\delta t} \alpha_t J_{t|s}^\top u(t, X_t) \mathrm{d}t \tag{27}$$

$$\approx J_{s+\delta t|s}^\top \left[\frac{A_{s+\delta t}}{A_s} \mathcal{U}_{s+\delta t} + (\delta t) \frac{\alpha_{s+\delta t}}{A_s} u(s + \delta t, X_{s+\delta t})\right], \tag{28}$$

where in the final line we approximate the second integral using its value at the larger timestep. By relabelling $\lambda_s = \frac{A_{s+\delta t}}{A_s}$, this can be written as

$$\mathcal{U}_s = J_{s+\delta t|s}^\top \left[ \lambda_s \mathcal{U}_{s+\delta t} + (1 - \lambda_s) u_{s+\delta t}(X_{s+\delta t}) \right]. \tag{29}$$

This is intuitive; each target $\mathcal{U}_s$ is constructed as a weighted combination of the 'running' next-step target $\mathcal{U}_{s+\delta t}$ and the next-step control $u(s + \delta t, X_{s+\delta t})$, and is then transformed by the single-step Jacobian $J_{s+\delta t|s} \approx \mathrm{Id} + \nabla b(s, X_s)(\delta t) + \nabla \sigma(s, X_s) \cdot \delta B_s$. The choice of $\alpha$-schedule therefore determines how 'far into the future' the algorithm looks to enforce the self-consistency property. Following Pidstrigach et al. (2025), we consider three different choices of $\alpha$-schedule: *next-step*, *average*, and *endpoint* prediction. For clarity, we explicitly write out the recursions for these three $\alpha$-schedules below.

- *Next-step prediction:* This choice of schedule chooses to only enforce the self-consistency property at consecutive timesteps; that is, it tries to enforce $u(s, X_s^u) = \mathbb{E}[J_{s+\delta t|s}^\top u(s + \delta t, X_{s+\delta t}^u)|X_s^u]$. This corresponds to taking $\lambda_s = 0$ in the above expression (29), so the targets are calculated as

$$\mathcal{U}_s = J_{s+\delta t|s}^\top u(s + \delta t, X_{s+\delta t}^u). \tag{30}$$

  This choice results in small target variance—$u(s, X_s)$ is likely close to $\mathcal{U}_s$, as there is only a single step for variance to accrue. As we enforce the correct control at the final step, this choice can be thought of as 'passing the information along the trajectory one-by-one'. As a result, next-step prediction can cause slower learning, but can give more stable training in settings where the training target variance might become large due to large Jacobians (for example, see the double-well experiments in Appendix E.7).

- *Average prediction:* This choice instead aims to enforce the self-consistency uniformly across the remaining time $t \in [s, T]$, so $\mathcal{U}_s = \frac{1}{T-s} \int_s^T J_{t|s}^\top u(t, X_t^u) \mathrm{d}t$. In this case, the recursive formula becomes

$$\mathcal{U}_s = J_{s+\delta t|s}^\top \left[ \frac{T-(s+\delta t)}{T-s} \mathcal{U}_{s+\delta t} + \frac{(\delta t)}{T-s} u(s + \delta t, X_{s+\delta t}^u) \right]. \tag{31}$$

  We found this choice to provide a good balance between training target variance and propagation of the terminal information, and it worked consistently well across the experimental settings (see results in Appendix E.7). For this reason, the results reported in the main body use the average $\alpha$-schedule.

- *Endpoint prediction:* This choice chooses to only enforce the self-consistency property to the terminal step; that is, it tries to enforce $u(s, X_s^u) = \mathbb{E}[J_{T-\delta t|s}^\top u(T - \delta t, X_{T-\delta t}^u)|X_s^u]$. This corresponds to taking $\lambda_s = 1$ in (29), so the targets are calculated as

$$\mathcal{U}_s = J_{s+\delta t|s}^\top \mathcal{U}_{s+\delta t}. \tag{32}$$

  While this choice focuses on using the known terminal information, variance can accrue in the many intermediate steps. Moreover, target variance is exacerbated in the case of diffusion bridges, as the control grows sharply towards the terminal time. In the experiments in Appendix E.7, we consistently found this choice to perform worse than average prediction.

We remark that other choices of $\alpha$-schedule are possible and can be chosen by the practitioner. The method for computing the training targets for a single trajectory in the generalised case is presented in Algorithm 3.

The above recursions can be understood as a backwards discretisation of continuous dynamics. If we consider the integral $I_s = \int_s^T \alpha_t J_{t|s}^\top u(t, X_t^u) \, \mathrm{d}t$, then this has the continuous dynamics $\mathrm{d}I_s = -\alpha_s u(s, X_s^u) \, \mathrm{d}s - \nabla b(s, X_s^u)^\top I_s \, \mathrm{d}s$, for $\sigma$ not dependent on $x$ (with an additional term if $\sigma$ has $x$-dependence).

## B.3. Sticking-the-Landing adjustments

Here, we explain how 'Sticking-the-Landing' (STL) adjustments (Roeder et al., 2017) can be included in the training target recursions, similar to the adjustments discussed in Domingo-Enrich (2024). The idea of the STL adjustments is to remove the variance in the training targets at the solution, by leveraging the knowledge that the corresponding potential must solve the appropriate HJB equation. Below, we provide a sketch proof showing how the STL adjustments arise in our framework, and show how they can be incorporated into our algorithm. For full details, see the proof in Appendix D. Note that this presentation is for a scalar noise term $\sigma$; the general case follows similarly by making the appropriate changes.

It is known that the control $u^*$ of the solution is of the form $u^* = \nabla\phi$, where $\phi$ solves the HJB equation,

$$\partial_t \phi + \mathcal{L}\phi + \tfrac{1}{2}|\sigma\nabla\phi|^2 = 0. \tag{HJB}$$

Taking gradients of the HJB equation and using $u = \nabla\phi$, we see that

$$0 = \partial_t u + \nabla\mathcal{L}\phi + \tfrac{\sigma^2}{2}\nabla|u|^2 = \partial_t u + \nabla(b \cdot u) + \tfrac{\sigma^2}{2}\Delta u + \tfrac{\sigma^2}{2}\nabla|u|^2. \tag{Grad-HJB}$$

By applying the product rule of Itô stochastic calculus to $J_{t|s}^\top u(t, X_t^u)$ (for full details see the proofs in Appendix D), we have

$$\mathrm{d}(J_{t|s}^\top u(t, X_t^u)) = J_{t|s}^\top \underbrace{\left(\partial_t u + \nabla(b \cdot u) + \tfrac{\sigma^2}{2}\Delta u + \tfrac{\sigma^2}{2}\nabla|u|^2\right)(t, X_t^u)}_{\text{RHS of (Grad-HJB)}} \mathrm{d}t + \sigma J_{t|s}^\top \nabla u(t, X_t^u) \cdot \mathrm{d}B_t. \tag{33}$$

Observe that the deterministic term contains precisely the right-hand side of (Grad-HJB), so is zero at the solution. Integrating over time, the optimal control $u^*$ therefore satisfies the following *path-wise* self-consistency property, which holds almost surely along trajectories of the solution,

$$u(s, X_s^u) = J_{t|s}^\top u(t, X_t^u) - \sigma \int_s^t J_{r|s}^\top \nabla u(r, X_r^u) \cdot \mathrm{d}B_r. \tag{34}$$

By taking expectations, we recover the standard self-consistency property (SC1) used in the standard version of our algorithm. However, numerically speaking, working with the expectation introduces variance, as the stochastic term must be averaged out across samples. Rather than working with the standard self-consistency property, 'sticking-the-landing' means instead working with this *path-wise* property directly, which at optimality will be satisfied almost surely.

**Computing the STL training targets**   To incorporate the STL adjustment into our algorithm, we instead fit to the STL training targets

$$\mathcal{U}_s := \frac{1}{A_s} \int_s^T \alpha_t J_{t|s}^\top u(t, X_t^u)\,\mathrm{d}t - \frac{1}{A_s} \int_s^T \alpha_t \left(\sigma \int_s^t J_{r|s}^\top \nabla u(r, X_r^u) \cdot \mathrm{d}B_r\right)\mathrm{d}t. \tag{35}$$

The first term is the same as in the usual case, and the second term constitutes the STL adjustments. It remains to show the recursion that the STL targets satisfy, in order to compute the target backwards along the trajectories. To do so, it is first helpful to rearrange the noise term as follows (using the fact $\int_s^T \mathbf{1}_{r\in[s,t]}\alpha_t\,\mathrm{d}t = \int_r^T \alpha_t\,\mathrm{d}t = A_r$),

$$\frac{1}{A_s}\int_s^T \alpha_t \left(\int_s^t J_{r|s}^\top \nabla u(r, X_r^u) \cdot \mathrm{d}B_r\right)\mathrm{d}t = \frac{1}{A_s}\int_s^T \int_s^T \mathbf{1}_{r\in[s,t]}\alpha_t J_{r|s}^\top \nabla u(r, X_r^u) \cdot \mathrm{d}B_r\,\mathrm{d}t$$

$$= \frac{1}{A_s}\int_s^T \left(\int_s^T \mathbf{1}_{r\in[s,t]}\alpha_t\,\mathrm{d}t\right) J_{r|s}^\top \nabla u(r, X_r^u) \cdot \mathrm{d}B_r$$

$$= \frac{1}{A_s}\int_s^T A_r J_{r|s}^\top \nabla u(r, X_r^u) \cdot \mathrm{d}B_r.$$

The STL training targets can then be computed according to the following recursion, mirroring the calculation in the standard case.

$$\mathcal{U}_s = \frac{1}{A_s}\int_s^T \alpha_t J_{t|s}^\top u(t, X_t)\mathrm{d}t + \frac{\sigma}{A_s}\int_s^T A_r J_{r|s}^\top \nabla u(r, X_r) \cdot \mathrm{d}B_r \tag{36}$$

$$= \left[\frac{1}{A_s}\int_{s+\delta t}^T \alpha_t J_{t|s}^\top u(t, X_t)\mathrm{d}t + \frac{\sigma}{A_s}\int_{s+\delta t}^T A_r J_{r|s}^\top \nabla u(r, X_r) \cdot \mathrm{d}B_r\right] \tag{37}$$

$$+ \left[\frac{1}{A_s}\int_s^{s+\delta t} \alpha_t J_{t|s}^\top u(t, X_t)\mathrm{d}t + \frac{\sigma}{A_s}\int_s^{s+\delta t} A_r J_{r|s}^\top \nabla u(r, X_r) \cdot \mathrm{d}B_r\right] \tag{38}$$

$$= J_{s+\delta t|s}^\top \left[\frac{1}{A_s}\int_{s+\delta t}^T \alpha_t J_{t|s+\delta t}^\top u(t, X_t)\mathrm{d}t + \frac{\sigma}{A_s}\int_{s+\delta t}^T A_r J_{r|s+\delta t}^\top \nabla u(r, X_r) \cdot \mathrm{d}B_r\right] \tag{39}$$

$$+ \left[\frac{1}{A_s}\int_s^{s+\delta t} \alpha_t J_{t|s}^\top u(t, X_t)\mathrm{d}t + \frac{\sigma}{A_s}\int_s^{s+\delta t} A_r J_{r|s}^\top \nabla u(r, X_r) \cdot \mathrm{d}B_r\right] \tag{40}$$

$$\approx \sigma\nabla u(s, X_s) \cdot \delta B_s + J_{s+\delta t|s}^\top \left[\frac{A_{s+\delta t}}{A_s}\mathcal{U}_{s+\delta t} + (\delta t)\frac{\alpha_{s+\delta t}}{A_s}u(s+\delta t, X_{s+\delta t})\right]. \tag{41}$$

---

**Algorithm 2** Control Consistency Diffusion Bridge (Generalised version)

---

**Input:** Additional drift $\tilde{b}$ for auxiliary process, drift parameterisation $f_\theta = b + (\sigma\sigma^\top)u_\theta$, number of iterations $N$, batch size $B$, weighting schedule $\alpha_t$.

**for** $n = 0$ **to** $N - 1$ **do**

Simulate $B$ paths of $(X_t^\theta)_{t\in[0,T]}$ using the current control,

$$\mathrm{d}X_t^\theta = \big(b + (\sigma\sigma^\top)u_\theta\big)(t, X_t^\theta)\mathrm{d}t + \sigma(t, X_t^\theta)\mathrm{d}B_t, \tag{43}$$

Solve for training targets $\mathcal{U}_s$ (47) backwards along the obtained paths $(X_t^\theta)$, using Algorithm 3.
Perform a gradient step on $\theta$ using the squared loss function

$$\mathcal{L}(\theta) = \mathop{\mathbb{E}}_{s\sim\mathbf{U}_{[0,T]}} \left[\left\|u_\theta(s, X_s^{\bar\theta}) - \mathcal{U}_s\right\|^2\right]. \tag{44}$$

**end for**

---

---

**Algorithm 3** Computing training targets along a single trajectory (generalised version)

---

**Input:** Trajectory $X = \{X_0, X_{\delta t}, X_{2\delta t}, ..., X_T\}$, Noise increments $\delta B = \{\delta B_0, \delta B_{\delta t}, \delta B_{2\delta t}, ..., \delta B_{T-\delta t}\}$, additional drift $\tilde{b}$, weighting schedule $\alpha_t$, optionally include STL adjustments.

**do:** Initialise at known final control

$$\mathcal{U}_{T-\delta t} = u(T - \delta t, X_{T-\delta t}) = (\sigma\sigma^\top)(T - \delta t, X_{T-\delta t})^{-1}\Big(\frac{X_T - X_{T-\delta t}}{\delta t} - b(T - \delta t, X_{T-\delta t})\Big) \tag{45}$$

**for** $s = T - 2\delta t, T - 3\delta t, ..., \delta t, 0$ **do**

Compute next training target $\mathcal{U}_s$ as

$$\mathcal{U}_s = -(\delta t)\nabla\tilde{b}(s, X_s)^\top u(s, X_s) + \tilde{J}_{s+\delta t|s}^\top \left[\frac{(\delta t)\alpha_{s+\delta t}}{A_s}u(s + \delta t, X_{s+\delta t}) + \frac{A_{s+\delta t}}{A_s}\mathcal{U}_{s+\delta t}\right] \tag{46}$$

$$+(\nabla u\,\sigma + u\,\nabla\sigma)(s, X_s^u)\cdot\delta B_t \qquad\qquad \text{(Optional STL adjustment)}$$

using the approximation $\tilde{J}_{s+\delta t|s} \approx \mathrm{Id} + \nabla(b + \tilde{b})(s, X_s)(\delta t) + \nabla\sigma(s, X_s)\cdot\delta B_s$.

**end for**

---

The only change to the standard case is the inclusion of the additional STL adjustment $\sigma\nabla u(s, X_s)\cdot\delta B_s$ in the recursion, which is consistent with the calculations in Domingo-Enrich (2024). In the case of a possibly spatially-dependent diffusion coefficient $\sigma$, performing similar calculations gives the path-wise self-consistency property

$$u(s, X_s^u) = J_{t|s}^\top u(t, X_t^u) - \int_s^t J_{r|s}^\top(\nabla u\,\sigma + u\,\nabla\sigma)(r, X_r^u)\cdot\mathrm{d}B_r, \tag{42}$$

and the STL adjustment is $(\nabla u\,\sigma + u\,\nabla\sigma)(r, X_r^u)\cdot\delta B_r$.

**Performance using STL adjustments**   We compare performance with and without the STL adjustments in Appendix E.8, and observe that it generally improves performance by a small amount, with the largest improvements seen in cases where large Jacobian terms arise (that is, the double-well and Müller-Brown experiments). The additional terms $(\nabla u\,\sigma + u\,\nabla\sigma)(r, X_r^u)\cdot\delta B_r$ can be computed efficiently using Jacobian-vector products, however this still adds an additional computational cost; generally we found that including the STL adjustments made the training steps approximately 1.5 times slower.

## C. The Generalised Self-Consistency Property and Algorithm

In this section, we provide more details regarding the generalised form of the self-consistency property outlined in Section 4.

Recall we can perform a change of measure

$$\nabla_x \mathbb{E}_{\mathbb{P}}[F(X_T)|X_t = x] = \nabla_x \mathbb{E}_{\tilde{\mathbb{P}}}[F(X_T)\tfrac{\mathrm{d}\mathbb{P}}{\mathrm{d}\tilde{\mathbb{P}}}|X_t = x],$$

where $\tilde{\mathbb{P}}$ is associated to an 'auxiliary' process $\mathrm{d}\tilde{X}_t = (b + \tilde{b})(t, \tilde{X}_t)\mathrm{d}t + \sigma(t, \tilde{X}_t)\mathrm{d}B_t$, in which we include an additional drift term $\tilde{b}$. The Radon-Nikodym derivative is given by the Girsanov weights $\frac{\mathrm{d}\mathbb{P}}{\mathrm{d}\tilde{\mathbb{P}}} = \exp(-\int_s^T \sigma^{-1}\tilde{b}(r, X_r)^\top \mathrm{d}B_r - \int_s^T \frac{(\sigma\sigma^\top)^{-1}}{2}\|\tilde{b}(r, X_r)\|^2 \mathrm{d}r)$ (Øksendal, 2003, Theorem 8.6.8). Continuing the calculations from Section 2 (see Appendix D.3 for a full derivation), we obtain the following *generalised self-consistency property*.

**Theorem C.1** (**Generalised self-consistency property**). *For any differentiable $\tilde{b}$ satisfying the conditions of Girsanov's theorem, the control drift $u^*$ of the solution satisfies*

$$u^*(s, x) = \mathbb{E}\Big[-\int_s^t \tilde{J}_{r|s}^\top \nabla\tilde{b}(r, X_r^{u^*})^\top u^*(r, X_r^{u^*})\mathrm{d}r + \tilde{J}_{t|s}^\top u^*(t, X_t^{u^*})|X_s^{u^*} = x\Big], \quad for\ 0 < s < t < T. \quad \text{(SC-gen)}$$

When $\sigma$ is independent of $x$, taking $\tilde{b} = -b$ yields the simplified form (SC2) presented in the main paper. Theorem 4.2 states that we can instead use (SC-gen) in place of the self-consistency property (SC1) in our proposed method, while retaining the same theoretical guarantees. We now provide details for implementing this generalised version of our algorithm.

**Solving for the training targets** Unlike in the standard case, the generalised self-consistency property includes 'running costs'. As before, we can construct the targets by weighting according to an $\alpha$-schedule. Denoting the training targets as

$$\mathcal{U}_s = \frac{1}{A_s} \int_s^T \alpha_t\big(-\int_s^t \tilde{J}_{r|s}^\top \nabla\tilde{b}(r, X_r)^\top u(r, X_r)\mathrm{d}r + \tilde{J}_{t|s}^\top u(t, X_t)\big)\mathrm{d}t, \tag{47}$$

we can modify the reverse recursions for computing $\mathcal{U}_s$. For completeness, we include the computations below.

$$\mathcal{U}_s = \frac{1}{A_s} \int_s^T \alpha_t\big(-\int_s^t \tilde{J}_{r|s}^\top \nabla\tilde{b}(r, X_r)^\top u(r, X_r)\mathrm{d}r + \tilde{J}_{t|s}^\top u(t, X_t)\big)\mathrm{d}t \tag{48}$$

$$= \frac{1}{A_s} \int_s^{s+\delta t} \alpha_t\big(-\int_s^t \tilde{J}_{r|s}^\top \nabla\tilde{b}(r, X_r)^\top u(r, X_r)\mathrm{d}r + \tilde{J}_{t|s}^\top u(t, X_t)\big)\mathrm{d}t$$
$$+ \frac{1}{A_s} \int_{s+\delta t}^T \alpha_t\big(-\int_s^t \tilde{J}_{r|s}^\top \nabla\tilde{b}(r, X_r)^\top u(r, X_r)\mathrm{d}r + \tilde{J}_{t|s}^\top u(t, X_t)\big)\mathrm{d}t \tag{49}$$

$$= \frac{1}{A_s} \int_s^{s+\delta t} \alpha_t\big(-\int_s^t \tilde{J}_{r|s}^\top \nabla\tilde{b}(r, X_r)^\top u(r, X_r)\mathrm{d}r + \tilde{J}_{t|s}^\top u(t, X_t)\big)\mathrm{d}t$$
$$+ \frac{1}{A_s} \int_{s+\delta t}^T \alpha_t\big(-\int_s^{s+\delta t} \tilde{J}_{r|s}^\top \nabla\tilde{b}(r, X_r)^\top u(r, X_r)\mathrm{d}r\big)\mathrm{d}t \tag{50}$$
$$+ \frac{1}{A_s} \int_{s+\delta t}^T \alpha_t\big(-\int_{s+\delta t}^t \tilde{J}_{r|s}^\top \nabla\tilde{b}(r, X_r)^\top u(r, X_r)\mathrm{d}r + \tilde{J}_{t|s}^\top u(t, X_t)\big)\mathrm{d}t \tag{51}$$

$$= \frac{1}{A_s} \int_s^{s+\delta t} \alpha_t\big(-\int_s^t \tilde{J}_{r|s}^\top \nabla\tilde{b}(r, X_r)^\top u(r, X_r)\mathrm{d}r + \tilde{J}_{t|s}^\top u(t, X_t)\big)\mathrm{d}t$$
$$+ \frac{1}{A_s} \int_{s+\delta t}^T \alpha_t\big(-\int_s^{s+\delta t} \tilde{J}_{r|s}^\top \nabla\tilde{b}(r, X_r)^\top u(r, X_r)\mathrm{d}r\big)\mathrm{d}t \tag{52}$$
$$+ \frac{1}{A_s} \tilde{J}_{s+\delta t|s}^\top \int_{s+\delta t}^T \alpha_t\big(-\int_{s+\delta t}^t \tilde{J}_{r|s+\delta t}^\top \nabla\tilde{b}(r, X_r)^\top u(r, X_r)\mathrm{d}r + \tilde{J}_{t|s+\delta t}^\top u(t, X_t)\big)\mathrm{d}t \tag{53}$$

$$\approx -(\delta t)\nabla\tilde{b}(s, X_s)^\top u(s, X_s) + \tilde{J}_{s+\delta t|s}^\top \Big[\tfrac{(\delta t)\alpha_{s+\delta t}}{A_s} u(s+\delta t, X_{s+\delta t}) + \tfrac{A_{s+\delta t}}{A_s}\mathcal{U}_{s+\delta t}\Big] \tag{54}$$

When using the second form of the self-consistency property, (SC2), this simplifies to

$$\mathcal{U}_s = (\delta t)\nabla b(s, X_s^u)^\top u(s, X_s^u) + \Big[\tfrac{(\delta t)\alpha_{s+\delta t}}{A_s} u(s+\delta t, X_{s+\delta t}^u) + \tfrac{A_{s+\delta t}}{A_s}\mathcal{U}_{s+\delta t}\Big] \tag{55}$$

**Algorithm** A description of the resulting generalised version of the algorithm is given in Algorithm 2. The method for solving for the training targets $\mathcal{U}_s$ backwards along the trajectories in the generalised case is given in Algorithm 3. In the paper, we focus on the two most natural forms of the self-consistency property, namely (SC1) and (SC2), which we anticipate will be suitable for the vast majority of applications.

**Sticking-the-Landing Adjustments** By examining the proof of Theorem C.1 in Appendix D, we see that the STL adjustments for the generalised version of the algorithm remain the same as the standard case discussed in Appendix B.3. They are included in blue in Algorithm 3.

## D. Proofs

In this section, we provide proofs for the results stated in the paper.

### D.1. Details of Sketch Derivation in Section 2

In Section 2, we provided an introductory derivation of the self-consistency property (SC1); here, we fill in the technical details of the calculations. We remark that (SC1) can also be derived directly from the computations in the proof of Theorem 4.2 provided in the next section.

Recall that we consider making a change of measure $\frac{d\mathbb{Q}}{d\mathbb{P}}(X) \propto F(X_T)$. We first provide a brief lemma concerning conditional expectations under this change of measure (see Solé (2005, Lemma 4.4)), which is utilised in the sketch derivation in Section 2 and in the full proof of Theorem 2.1.

**Lemma D.1** (Conditional change of measure). *Define $F_t = \mathbb{E}_{\mathbb{P}}[F(X_T)|X_t]$. Then for any $H$, we have*

$$\frac{1}{F_t}\mathbb{E}_{\mathbb{P}}[FH|X_t] = \mathbb{E}_{\mathbb{Q}}[H|X_t], \qquad \mathbb{Q} - a.s. \tag{56}$$

*Proof.* Let us consider the change of measure $\frac{d\mathbb{Q}}{d\mathbb{P}}(X) = F(X_T)$ (where we assume the normalising constant is included in $F$; otherwise, one can rescale accordingly). A quick calculation shows that the local densities are given by $\frac{d\mathbb{Q}}{d\mathbb{P}}|_{\sigma(X_t)} = F_t$. For a test function $g(X_t)$, we therefore have

$$\mathbb{E}_{\mathbb{Q}}\big[g(X_t)\tfrac{1}{F_t}\mathbb{E}_{\mathbb{P}}[FH|X_t]\big] = \mathbb{E}_{\mathbb{P}}\big[g(X_t)\mathbb{E}_{\mathbb{P}}[FH|X_t]\big] \tag{57}$$
$$= \mathbb{E}_{\mathbb{P}}\big[g(X_t)FH\big] \tag{58}$$
$$= \mathbb{E}_{\mathbb{Q}}\big[g(X_t)H\big] \tag{59}$$
$$= \mathbb{E}_{\mathbb{Q}}\big[g(X_t)\mathbb{E}_{\mathbb{Q}}[H|X_t]\big]. \tag{60}$$

Here, we used the tower property of conditional expectations in the second and final lines, and in the first and third lines used the definition of the reweighted measure $\mathbb{Q}$. Since $g$ is an arbitrary test function, we thus see that $\frac{1}{F_t}\mathbb{E}_{\mathbb{P}}[FH|X_t] = \mathbb{E}_{\mathbb{Q}}[H|X_t]$ $\mathbb{Q}$-almost surely, as required. □

**Theorem 2.1** (**Self-consistency property (I)**). *For $0 < s < t < T$, the optimal control $u^*$ satisfies*

$$u^*(s,x) = \mathbb{E}[J_{t|s}^\top u^*(t, X_t^{u^*})|X_s^{u^*} = x]. \tag{SC1}$$

*Proof.* We begin by filling in the details of the sketch derivation in Section 2 showing that $u^*(t,x) = \mathbb{E}_{\mathbb{Q}}[J_{T|t}^\top \nabla \log F(X_T)|X_t = x]$ when $F$ is differentiable. Starting from Doob's $h$-transform and using Lemma D.1 with $H = \nabla \log F(X_T)$ in the final line, we have

$$u^*(t, x) = \nabla_x \log \mathbb{E}_\mathbb{P}[F(X_T) \mid X_t = x] \tag{61}$$

$$= \frac{\nabla_x \mathbb{E}_\mathbb{P}[F(X_T) \mid X_t = x]}{\mathbb{E}_\mathbb{P}[F(X_T) \mid X_t = x]} \qquad \nabla \log f = \frac{\nabla f}{f} \tag{62}$$

$$= \frac{\mathbb{E}_\mathbb{P}[J_{T|t}^\top \nabla F(X_T) \mid X_t = x]}{\mathbb{E}_\mathbb{P}[F(X_T) \mid X_t = x]} \qquad \text{interchange } \nabla_x \text{ and } \mathbb{E}_\mathbb{P} \tag{63}$$

$$= \frac{\mathbb{E}_\mathbb{P}[J_{T|t}^\top F(X_T) \nabla \log F(X_T) \mid X_t = x]}{\mathbb{E}_\mathbb{P}[F(X_T) \mid X_t = x]} \qquad \nabla \log f = \frac{\nabla f}{f} \tag{64}$$

$$= \mathbb{E}_\mathbb{Q}[J_{T|t}^\top \nabla \log F(X_T) \mid X_t = x] \qquad \text{Lemma D.1} \tag{65}$$

The interchanging of the derivative and the expectation in the third line is often referred to as the reparameterisation trick in the machine learning literature. It can be made rigorous using the theory of stochastic flows (Kunita, 1984; Elworthy et al., 1999); namely, if one considers a fixed realisation $\omega$ of the Brownian path between $t$ and $T$, then the terminal state $X_T$ is a function of the value $x$ at time $t$, which we can denote $x \mapsto \varphi_{t,T}(x, \omega) = X_T(\omega)$. Under mild assumptions on $b, \sigma$ (see Kunita (1986)), the stochastic flow $\varphi_{t,T}(\cdot, \omega)$ is continuously differentiable. Using Leibniz's rule, for continuously differentiable $F$ we then have,

$$\nabla_x \mathbb{E}_\mathbb{P}[F(X_T) | X_t = x] = \nabla_x \int F(\varphi_{t,T}(x, \omega)) \mathrm{d}\mu(\omega) \tag{66}$$

$$= \int \nabla_x \varphi_{t,T}(x, \omega)^\top \nabla F(\varphi_{t,T}(x, \omega)) \mathrm{d}\mu(\omega) \tag{67}$$

$$= \mathbb{E}_\mathbb{P}[J_{T|t}^\top \nabla F(X_T) | X_t = x]. \tag{68}$$

The self-consistency property follows from this expression by utilising the semigroup property of the Jacobian process. For $s < t$, we can 'invert' the property at the later time,

$$u^*(s, x) = \mathbb{E}_\mathbb{Q}[J_{T|s}^\top \nabla \log F(X_T) | X_s = x] \tag{69}$$

$$= \mathbb{E}_\mathbb{Q}[(J_{T|t} J_{t|s})^\top \nabla \log F(X_T) | X_s = x] \tag{70}$$

$$= \mathbb{E}_\mathbb{Q}\big[\mathbb{E}_\mathbb{Q}[J_{t|s}^\top J_{T|t}^\top \nabla \log F(X_T) | X_{[s,t]}] | X_s = x\big] \tag{71}$$

$$= \mathbb{E}_\mathbb{Q}\big[J_{t|s}^\top \mathbb{E}_\mathbb{Q}[J_{T|t}^\top \nabla \log F(X_T) | X_t] | X_s = x\big] \tag{72}$$

$$= \mathbb{E}_\mathbb{Q}\big[J_{t|s}^\top u^*(t, X_t) | X_s = x\big]. \tag{73}$$

Here, we used the tower rule in line 3, and the Markov property of $X$ in line 4.

Finally, we remark that this self-consistency property also holds in the singular case when $F = \delta_{x_T}$, which corresponds to diffusion bridges. To see this, take $\tilde{T} < T$, and apply the above result with $\tilde{T}$ in place of $T$ and with $F(X_{\tilde{T}}) = p(X_T = x_T | X_{\tilde{T}})$, which is smooth. We thus obtain the result for $s < t < \tilde{T}$, and as $\tilde{T}$ was arbitrary we see that the self-consistency property holds for $s < t < T$. □

## D.2. Proof of Theorem 3.1

We now move to proving the characterisation of the diffusion bridge given in Theorem 3.1, upon which our approach is based. In fact, all results in the paper essentially follow from such computations, which involve relating the self-consistency property to the gradient of the HJB equation by considering an Itô expansion of $J_{t|s}^\top u(t, X_t^u)$ along the controlled dynamics.

We will prove a slightly stronger result in Theorem D.3, from which Theorem 3.1 will follow. To begin, we introduce the following definitions.

**Definition D.2.** We call a (time-dependent) vector field $u$

- *bridge-preserving*, if the bridges of the controlled dynamics $X_t^u$ (3) coincide with those of the base dynamics $X_t$(1),

- *self-consistent*, if $u$ satisfies $u(s, x) = \mathbb{E}[J_{t|s}^\top u(t, X_t^u)|X_s^u = x]$,

- *of gradient form*, if there exists a time-dependent potential $\phi$ such that $u(t, x) = \nabla_x \phi(t, x)$,

- *of gradient form at one point in time*, if $u(t, \cdot) = \nabla\phi$ for some $t > 0$ and time-independent potential $\phi$.

The statement of our next result requires the generator of the base dynamics,

$$\mathcal{L} = \sum_i b_i \partial_i + \tfrac{1}{2} \sum_{ij} (\sigma\sigma^\top)_{ij} \partial_{ij}.$$

**Theorem D.3.** *The following are equivalent:*

  *(i) The vector field $u$ is bridge-preserving,*

 *(ii) we have $u = \nabla\phi$, and $\phi$ satisfies the Hamilton-Jacobi-Bellman (HJB) equation*

$$\partial_t \phi + \mathcal{L}\phi + \tfrac{1}{2}|\sigma\nabla\phi|^2 = 0, \tag{HJB}$$

*(iii) the vector field $u$ is of gradient form and self-consistent,*

*(iv) the vector field $u$ is of gradient form at one point in time and self-consistent.*

*Proof.* To simplify the notation, we present the proof here for a scalar diffusion term $\sigma$, but the proof proceeds similarly for general diffusion coefficients under our ellipticity assumption by making the appropriate modifications.

(i) $\iff$ (ii): This equivalence is well known (and SDEs with bridge preserving drifts are in the reciprocal class of the base process), see for example Léonard (2013); Rœlly (2013), or Vargas et al. (2024, Section E.5).

(ii) $\iff$ (iii): Taking the gradient of the HJB-equation and using $u = \nabla\phi$, we see that

$$0 = \partial_t u + \nabla\mathcal{L}\phi + \frac{\sigma^2}{2}\nabla|u|^2 = \partial_t u + \nabla(b \cdot u) + \frac{\sigma^2}{2}\Delta u + \frac{\sigma^2}{2}\nabla|u|^2, \tag{Grad-HJB}$$

using the facts that $\nabla\mathcal{L}\phi = \nabla(b \cdot u) + \frac{\sigma^2}{2}\nabla(\Delta\phi)$ and $\nabla(\Delta\phi) = \Delta u$. Here and throughout, it is understood the Laplacian is taken element-wise in $\Delta u$.

On the other hand, we can compute the time evolution of $t \mapsto J_{t|s}^\top u(t, X_t^u)$ along the controlled dynamics, using the product rule of Itô stochastic calculus:

$$
\begin{aligned}
\mathrm{d}(J_{t|s}^\top u(t, X_t^u)) &= J_{t|s}^\top \, \mathrm{d}u(t, X_t^u) + \mathrm{d}J_{t|s}^\top u(t, X_t^u) \\
&= J_{t|s}^\top \left( \partial_t u(t, X_t^u)\,\mathrm{d}t + \nabla u(t, X_t^u) \cdot \big(b(t, X_t^u)\,\mathrm{d}t + \sigma^2 u(t, X_t^u)\,\mathrm{d}t + \sigma\mathrm{d}B_t\big) + \frac{\sigma^2}{2}\Delta u(t, X_t^u)\,\mathrm{d}t \right) \\
&\quad + J_{t|s}^\top \nabla b(t, X_t^u) \cdot u(t, X_t^u)\,\mathrm{d}t \\
&= J_{t|s}^\top \underbrace{\left( \partial_t u + \nabla(b \cdot u) + \frac{\sigma^2}{2}\Delta u + \frac{\sigma^2}{2}\nabla|u|^2 \right)}_{\text{RHS of (Grad-HJB)}} (t, X_t^u)\,\mathrm{d}t + \sigma J_{t|s}^\top \nabla u(t, X_t^u) \cdot \mathrm{d}B_t.
\end{aligned}
$$

The last line of this calculation uses $\frac{1}{2}\nabla|u|^2 = (u \cdot \nabla)u$, which holds for vector fields of gradient form. Note that in the general case with a possibly spatially-dependent diffusion coefficient, performing similar calculations instead gives rise to the noise term $J_{t|s}^\top (\nabla u\,\sigma + u\,\nabla\sigma)(t, X_t^u) \cdot \mathrm{d}B_t$.

The equivalence between (ii) and (iii) follows from the observation that the drift in $\mathrm{d}(J_{t|s}^\top u(t, X_t^u))$ coincides with the right-hand side of (Grad-HJB). More precisely, from the above calculation the self-consistency property (SC1) can be re-expressed in the form

$$u(s,x) = \mathbb{E}[J_{t|s}^\top u(t, X_t^u)|X_s^u = x]$$

$$= \mathbb{E}\left[ J_{s|s}^\top u(s, X_s^u) + \int_s^t J_{t|s}^\top \left( \partial_t u + \nabla(b\cdot u) + \tfrac{\sigma^2}{2}\Delta u + \tfrac{\sigma^2}{2}\nabla|u|^2 \right)(r, X_r^u)\,\mathrm{d}r \Big| X_s^u = x \right],$$

owing to the fact that the martingale term $\nabla u(t, X_t^u)\cdot \mathrm{d}B_t$ has zero expectation. Using $J_{s|s} = Id$, this is equivalent to

$$\mathbb{E}\left[ \int_s^t \left( \partial_t u + \nabla(b\cdot u) + \tfrac{\sigma^2}{2}\Delta u + \tfrac{\sigma^2}{2}\nabla|u|^2 \right)(r, X_r^u)\,\mathrm{d}r \Big| X_s^u = x \right] = 0. \tag{75}$$

Now it is clear that (ii) implies (iii), because (Grad-HJB) implies (75), for all $s \le t$ and $x \in \mathbb{R}^d$. Conversely, we can obtain (Grad-HJB) from (75) by taking the derivative with respect to $t$ of (75) and setting $t = s$; hence (iii) implies (ii).

(iii) $\iff$ (iv): We first define the vorticity

$$\Omega_{ij}(t,x) := \partial_i u_j(t,x) - \partial_j u_i(t,x), \qquad t > 0, \quad x \in \mathbb{R}^d. \tag{76}$$

The Poincaré lemma implies that $\Omega(t,\cdot) = 0$ if and only if $u(t,\cdot)$ is of gradient form. As above, self-consistency yields the equation

$$\partial_t u + \tfrac{\sigma^2}{2}\Delta u + (\nabla b)u + (b\cdot\nabla)u + (u\cdot\nabla)u = 0. \tag{77}$$

Note that this does not simplify to (Grad-HJB), since we do not make the assumption $u = \nabla\phi$. Taking another spatial derivative and anti-symmetrising, we see that $\Omega$ solves the linear parabolic PDE

$$\left( \partial_t + (b+u)\cdot\nabla + \tfrac{\sigma^2}{2}\Delta \right)\Omega + (\nabla(b+u))\Omega + \Omega(\nabla(b+u))^\top = 0, \tag{78}$$

which clearly admits $\Omega \equiv 0$ as a solution. Under standard regularity assumptions, solutions to (78) are unique once $\Omega(t,\cdot)$ is prescribed for some $t$ (both forwards and backwards in time), see for example (Pazy, 2012). From this, it follows that (iv) implies (iii), and the converse implication follows directly from the definition. $\qquad\square$

**Theorem 3.1.** *Consider the controlled dynamics* (3) *with control drift* $u : [0,T] \times \mathbb{R}^d \to \mathbb{R}^d$. *Suppose that:*

  *(i)* $u$ *satisfies the self-consistency property* (SC1);

  *(ii)* $u$ *is such that it forces the controlled dynamics to terminate at* $X_T^u = x_T$;

  *(iii)* $u$ *is of gradient form for some time* $t$, *(that is,* $u(t,\cdot) = \nabla\phi(\cdot)$ *for some scalar function* $\phi$*).*

*Then the controlled process* $X_t^u$ *is the diffusion bridge.*

*Proof of Theorem 3.1.* The result follows as a corollary from Theorem D.3. More precisely, we use the implication $(iv) \iff (ii)$. Under the gradient form and consistency assumptions, the addition of the control $u$ does not change the process when conditioned on $X_T = x_T$. $\qquad\square$

### D.3. Generalisations of the Self-Consistency Property

We now provide the proofs of the analogous results for the generalised self-consistency property (SC-gen) presented in Section 4 and Appendix C. Recall that $\tilde{J}_{t|s}$ is the Jacobian process associated to the auxiliary process $\tilde{X}$, which evolves according to the SDE $\mathrm{d}\tilde{X}_t = (b+\tilde{b})(t, \tilde{X}_t)\mathrm{d}t + \sigma(t, \tilde{X}_t)\mathrm{d}B_t$ including an additional drift function $\tilde{b}$. Theorems 4.2 and C.1 follow from a minor modification to the proof of Theorem D.3.

**Theorem C.1 (Generalised self-consistency property).** *For any differentiable* $\tilde{b}$ *satisfying the conditions of Girsanov's theorem, the control drift* $u^*$ *of the solution satisfies*

$$u^*(s,x) = \mathbb{E}\left[ -\int_s^t \tilde{J}_{r|s}^\top \nabla\tilde{b}(r, X_r^{u^*})^\top u^*(r, X_r^{u^*})\mathrm{d}r + \tilde{J}_{t|s}^\top u^*(t, X_t^{u^*})|X_s^{u^*} = x \right], \quad \text{for } 0 < s < t < T. \quad \text{(SC-gen)}$$

**Theorem 4.2.** *The result of Th. 2.1 holds with the generalised self-consistency property* (SC-gen) *in place of* (SC1).

*Proof.* To prove this result, we will in fact prove a more general result from which (SC-gen) arises as a particular case. Consider a class of matrix-valued functions $M_s(t)$, with $M_s(t)$ invertible and $t \mapsto M_s(t)$ differentiable, and assume $M_s(s) = \mathrm{Id}$.

**Claim:** We will show that if $u^*$ is the control of the diffusion bridge solution then it satisfies

$$u^*(s,x) = \mathbb{E}\Big[ \int_s^t \big( M_s(r) \nabla b^\top - \partial M_s(r) \big) u^*(r, X_r^{u^*}) \mathrm{d}r + M_s(t) u^*(t, X_t^{u^*}) | X_s^{u^*} = x \Big]. \tag{79}$$

In light of the proof of Theorem D.3, to prove this claim we need only to show that the equivalence $(ii) \iff (iii)$ still holds. To do so, we mirror the proof of the standard case, and perform an Itô expansion of $M_s(t) u(t, X_t^u)$ along the controlled dynamics

$$\mathrm{d}(M_s(t) u(t, X_t^u)) = M_s(t) \mathrm{d}u(t, X_t^u) + \mathrm{d}M_s(t) u(t, X_t^u) \tag{80a}$$

$$= M_s(t) \left( \partial_t u(t, X_t^u) \mathrm{d}t + \nabla u(t, X_t^u) \cdot \big( b(t, X_t^u) \mathrm{d}t + \sigma^2 u(t, X_t^u) \mathrm{d}t + \sigma \mathrm{d}B_t \big) + \tfrac{\sigma^2}{2} \Delta u(t, X_t^u) \mathrm{d}t \right) \tag{80b}$$
$$+ \partial M_s(t) \cdot u(t, X_t^u) \mathrm{d}t$$

$$= M_s(t) \underbrace{\left( \partial_t u + \nabla(b \cdot u) + \tfrac{\sigma^2}{2} \Delta u + \tfrac{\sigma^2}{2} \nabla |u|^2 \right)(t, X_t^u)}_{\text{RHS of (Grad-HJB)}} \mathrm{d}t$$
$$+ \big( \partial M_s(t) - M_s(t) \nabla b^\top \big) u(t, X_t^u) \mathrm{d}t + \sigma M_s(t) \nabla u(t, X_t^u) \cdot \mathrm{d}B_t. \tag{80c}$$

The equivalence in the claim follows by the same argument as in the proof of Theorem D.3. Namely, if $u$ is the diffusion bridge control then by integrating (80c) conditioned on $X_s^u = x$ we obtain

$$u(t, X_t^u) - u(s,x) = \mathbb{E}\bigg[ \int_s^t \big( \partial M_s(r) - M_s(r) \nabla b^\top \big) u(r, X_r^u) \mathrm{d}r \,|\, X_s^u = x \bigg], \tag{81}$$

where we use that the solution satisfies (Grad-HJB) and the martingale term $M_s(r) \nabla u(r, X_r^u) \cdot \mathrm{d}B_r$ has zero expectation. Rearranging gives the self-consistency expression (79). Conversely, we can obtain (Grad-HJB) from (79) by taking the derivative with respect to $t$ and setting $t = s$.

The results of Theorem C.1 and Theorem 4.1 follow from this result by taking $M_s(t) = \tilde{J}_{t|s}^\top$ and noting that in this case we have $\partial M_s(t) = M_s(t) \nabla (b + \tilde{b})^\top$, from which substituting into (79) gives (SC-gen). $\square$

The above result relates to the family of 'reparameterisation matrices' $M_s(t)$ introduced in Domingo-Enrich et al. (2024); we comment more on this connection in Appendix F. We note that our presentation focuses on taking $M_s(t) = \tilde{J}_{t|s}^\top$, as this provides a probabilistic interpretation of the role that these matrices play. Nevertheless, we provide this more general version in the claim above as it may be useful in some settings (and note that it justifies the use of (SC2) for spatially-dependent $\sigma$).

**Change-of-Measure Computation** For completeness, we here also fill in the details of the change-of-measure computation outlined in Section 4. This derives the generalised self-consistency property (SC-gen) in a way that mirrors the introductory derivation of (SC1) given in Section 2. Again, we present the computations for scalar $\sigma$ and the general result follows with the appropriate modifications. Note that as in the earlier case, this relation can also be obtained directly from (80c) in the proof above.

Define $\tilde{\mathbb{P}}$ to be the path measure associated to the auxiliary process $\mathrm{d}\tilde{X}_t = (b + \tilde{b})(t, \tilde{X}_t)\mathrm{d}t + \sigma(t, \tilde{X}_t)\mathrm{d}B_t$. By Girsanov's theorem (Øksendal, 2003, Theorem 8.6.8), the Radon-Nikodym derivative is

$$\frac{\mathrm{d}\mathbb{P}}{\mathrm{d}\tilde{\mathbb{P}}}(X_{[s,T]}) = \exp\left( -\int_s^T \sigma^{-1} \tilde{b}(r, X_r)^\top \mathrm{d}B_r - \int_s^T \tfrac{(\sigma \sigma^\top)^{-1}}{2} \|\tilde{b}(r, X_r)\|^2 \mathrm{d}t \right).$$

For ease of notation, we will present the result for scalar diffusion coefficient $\sigma$; the proof of the general case follows with the appropriate changes. As in the proof of Theorem 2.1, we begin with Doob's $h$-transform, then perform the following calculations (where $B^{\mathbb{P}}, B^{\tilde{\mathbb{P}}}, B^{\mathbb{Q}}$ denote standard Brownian motions under their respective measures).

$$
\begin{aligned}
u^*(s,x) &= \nabla_x \log \mathbb{E}_{\mathbb{P}}[F(X_T)|X_s = x] \\
&= \frac{1}{\mathbb{E}_{\mathbb{P}}[F(X_T)|X_s = x]} \nabla_x \mathbb{E}_{\mathbb{P}}[F(X_T)|X_s = x] \\
&= \frac{1}{\mathbb{E}_{\mathbb{P}}[F(X_T)|X_s = x]} \nabla_x \mathbb{E}_{\tilde{\mathbb{P}}}[F(\tilde{X}_T)\tfrac{\mathrm{d}\mathbb{P}}{\mathrm{d}\tilde{\mathbb{P}}}|\tilde{X}_s = x] \\
&= \frac{1}{\mathbb{E}_{\mathbb{P}}[F(X_T)|X_s = x]} \mathbb{E}_{\tilde{\mathbb{P}}}\Big[ \tilde{J}_{T|s}^\top \nabla F(\tilde{X}_T)\tfrac{\mathrm{d}\mathbb{P}}{\mathrm{d}\tilde{\mathbb{P}}} \\
&\qquad + F(\tilde{X}_T)\tfrac{\mathrm{d}\mathbb{P}}{\mathrm{d}\tilde{\mathbb{P}}}\Big( -\int_s^T \tfrac{1}{\sigma^2}\tilde{J}_{r|s}^\top \nabla \tilde{b}(r,\tilde{X}_r)^\top \tilde{b}(r,\tilde{X}_r)\mathrm{d}r - \int_s^T \tfrac{1}{\sigma}\tilde{J}_{r|s}^\top \nabla \tilde{b}(r,\tilde{X}_r)^\top \mathrm{d}B_r^{\tilde{\mathbb{P}}} \Big)|\tilde{X}_s = x \Big] \\
&= \frac{1}{\mathbb{E}_{\mathbb{P}}[F(X_T)|X_s = x]} \mathbb{E}_{\mathbb{P}}\Big[ \tilde{J}_{T|s}^\top \nabla F(X_T) \\
&\qquad + F(X_T)\Big( -\int_s^T \tfrac{1}{\sigma^2}\tilde{J}_{r|s}^\top \nabla \tilde{b}(r,X_r)^\top \tilde{b}(r,X_r)\mathrm{d}r - \int_s^T \tfrac{1}{\sigma}\tilde{J}_{r|s}^\top \nabla \tilde{b}(r,X_r)^\top (\mathrm{d}B_r^{\mathbb{P}} - \tfrac{1}{\sigma}\tilde{b}(r,X_r)\mathrm{d}r) \Big)|X_s = x \Big] \\
&= \frac{1}{\mathbb{E}_{\mathbb{P}}[F(X_T)|X_s = x]} \mathbb{E}_{\mathbb{P}}\Big[ \tilde{J}_{T|s}^\top \nabla F(X_T) + F(X_T)\Big( -\int_s^T \tfrac{1}{\sigma}\tilde{J}_{r|s}^\top \nabla \tilde{b}(r,X_r)^\top \mathrm{d}B_r^{\mathbb{P}} \Big)|X_s = x \Big] \\
&= \mathbb{E}_{\mathbb{Q}}\Big[ \tilde{J}_{T|s}^\top \nabla \log F(X_T) - \int_s^T \tfrac{1}{\sigma}\tilde{J}_{r|s}^\top \nabla \tilde{b}(r,X_r)^\top (\mathrm{d}B_r^{\mathbb{Q}} + \sigma u^* \mathrm{d}t)|X_s = x \Big] \\
&= \mathbb{E}_{\mathbb{Q}}\Big[ \tilde{J}_{T|s}^\top \nabla \log F(X_T) - \int_s^T \tilde{J}_{r|s}^\top \nabla \tilde{b}(r,X_r)^\top u^*(r,X_r)\mathrm{d}r|X_s = x \Big].
\end{aligned}
$$

Much of this derivation follows similarly to the standard case. When we perform the change of measures, we include the appropriate modifications to the Brownian motions; namely, we have $\mathrm{d}B_r^{\mathbb{P}} = \mathrm{d}B_r^{\tilde{\mathbb{P}}} + \tfrac{1}{\sigma}\tilde{b}(r,X_r)\mathrm{d}r$, and $\mathrm{d}B_r^{\mathbb{P}} = \mathrm{d}B_r^{\mathbb{Q}} + \sigma u^* \mathrm{d}r$.

The remainder of the computation follows similarly to the proof of Theorem 2.1; as before, we 'invert' the property at a later time $t$, utilising the tower rule and the semigroup property of the Jacobian process $\tilde{J}_{t|s}$.

$$
\begin{aligned}
u^*(s,x) &= \mathbb{E}_{\mathbb{Q}}[\tilde{J}_{T|s}^\top \nabla \log F(X_T) - \int_s^T \tilde{J}_{r|s}^\top \nabla \tilde{b}(r,X_r)^\top u^*(r,X_r)\mathrm{d}r|X_s = x] \\
&= \mathbb{E}_{\mathbb{Q}}[-\int_s^t \tilde{J}_{r|s}^\top \nabla \tilde{b}(r,X_r)^\top u^*(r,X_r)\mathrm{d}r - \int_t^T \tilde{J}_{r|s}^\top \nabla \tilde{b}(r,X_r)^\top u^*(r,X_r)\mathrm{d}r + \tilde{J}_{T|s}^\top \nabla \log F(X_T)|X_t = x] \\
&= \mathbb{E}_{\mathbb{Q}}[-\int_s^t \tilde{J}_{r|s}^\top \nabla \tilde{b}(r,X_r)^\top u^*(r,X_r)\mathrm{d}r \\
&\qquad -\int_t^T (\tilde{J}_{r|t}\tilde{J}_{t|s})^\top \nabla \tilde{b}(r,X_r)^\top u^*(r,X_r)\mathrm{d}r + (\tilde{J}_{T|t}\tilde{J}_{t|s})\top \nabla \log F(X_T)|X_t = x] \\
&= \mathbb{E}_{\mathbb{Q}}\big[\mathbb{E}_{\mathbb{Q}}[-\int_s^t \tilde{J}_{r|s}^\top \nabla \tilde{b}(r,X_r)^\top u^*(r,X_r)\mathrm{d}r \\
&\qquad -\int_t^T (\tilde{J}_{r|t}\tilde{J}_{t|s})^\top \nabla \tilde{b}(r,X_r)^\top u^*(r,X_r)\mathrm{d}r + (\tilde{J}_{T|t}\tilde{J}_{t|s})^\top \nabla \log F(X_T)|X_{[s,t]}]\,|X_s = x\big] \\
&= \mathbb{E}_{\mathbb{Q}}\big[ -\int_s^t \tilde{J}_{r|s}^\top \nabla \tilde{b}(r,X_r)^\top u^*(r,X_r)\mathrm{d}r \\
&\qquad +\tilde{J}_{t|s}^\top \mathbb{E}_{\mathbb{Q}}[-\int_t^T \tilde{J}_{r|t}^\top \nabla \tilde{b}(r,X_r)^\top u^*(r,X_r)\mathrm{d}r + \tilde{J}_{T|t}^\top \nabla \log F(X_T)|X_t]|X_s = x\big] \\
&= \mathbb{E}\big[ -\int_s^t \tilde{J}_{r|s}^\top \nabla \tilde{b}(r,X_r^{u^*})^\top u^*(r,X_r^{u^*})\mathrm{d}r + \tilde{J}_{t|s}^\top u^*(t,X_t^{u^*})|X_s^{u^*} = x\big],
\end{aligned}
$$

which completes the computation.

### D.4. Derivation through $\alpha$-schedule

We also remark here that the generalised form of the self-consistency property in (79) can in fact also be derived from (SC1) by weighting according to a particular choice of matrix-valued $\alpha$-schedule, similar in style to the presentation in Pidstrigach et al. (2025, Lemma 3.2).

**Weighted self-consistency.** The self-consistency property (SC1) holds for any $s < t$; it can therefore be integrated over future times $t$ when constructing the training targets according to some weighting schedule $\alpha$, as done in (13). More generally, we can let $\alpha_s$ be a matrix-valued random measure on $(s, t]$, and write

$$\beta_s(r) := \alpha_s((s, r]), \qquad r \in [s, t].$$

We call $\alpha_s$ admissible if $\beta_s$ is adapted, of finite variation, satisfies $\beta_s(s) = 0$, $\beta_s(t) = I_d$, and obeys the integrability assumptions needed below. Then the weighted self-consistency property is

$$u(s, x) = \mathbb{E}\left[\int_{(s,t]} \alpha_s(dr)\, J_{r|s}^\top u(r, X_r^u) \,\big|\, X_s^u = x\right]. \tag{$\alpha$-SC}$$

For deterministic weights, this follows directly by integrating (SC1) over $r$ according to the weights. For stochastic weights, the right way to interpret the condition is that $\alpha_s$ is non-anticipative: its cumulative process $\beta_s(r)$ may depend on the path up to time $r$, but not on later parts of the trajectory. Point masses are allowed, and simply correspond to jumps of the cumulative weight $\beta_s$.

Note that this formulation corresponds to the weighted (SC1) training target (13) when using the deterministic scalar measure

$$\alpha_s(dr) = \frac{\alpha_r}{A_s} I_d \, dr, \qquad A_s = \int_s^T \alpha_r \, dr.$$

Thus (13) is obtained by averaging the future self-consistency targets $J_{r|s}^\top u(r, X_r^u)$ according to a normalized schedule.

Indeed, at the solution the process $Y_{r|s} := J_{r|s}^\top u(r, X_r^u)$ is a martingale, which is the process-level form of (SC1). If $\alpha_s = d\beta_s$, then integration by parts gives

$$\int_{(s,t]} \alpha_s(dr)\, Y_{r|s} = Y_{t|s} - \int_s^t \beta_s(r)\, dY_{r|s}.$$

The second term is a martingale integral and therefore has zero conditional expectation. Since $\mathbb{E}[Y_{t|s} \mid X_s^u = x] = u(s, x)$, this gives ($\alpha$-SC).

We now relate this weighted form ($\alpha$-SC) to the general matrix-valued self-consistency target. In (79), we proved a consistency property for general differentiable, invertible matrices $M_s(r)$ (of which taking $M_s(r)$ to be the auxiliary Jacobian $\tilde{J}_{t|s}$ is particular case). The next lemma shows that the general target (79) can also be obtained from ($\alpha$-SC) by choosing a specific adapted $\alpha$-weighting schedule.

To do so, on the open interval $(s, t)$ we choose the density of $\alpha_s$ so that the factor $J_{r|s}^\top$ in ($\alpha$-SC) is cancelled and replaced by the desired matrix coefficient. To obtain the terminal term in (79), the measure must additionally include a point mass at $t$.

**Lemma D.4** (Recovering (79) by weighting (SC1)). *Fix $0 < s < t < T$, and assume $M_s : [s, t] \to \mathbb{R}^{d \times d}$ is continuously differentiable, invertible, and normalized by*

$$M_s(s) = I_d.$$

*We assume also that $J_{r|s}^\top$ is invertible. Let*

$$\alpha_s^M(dr) := \left(M_s(r)\nabla b(r, X_r^u)^\top - \partial_r M_s(r)\right)(J_{r|s}^\top)^{-1} \mathbf{1}_{(s,t)}(r)\, dr + M_s(t)(J_{t|s}^\top)^{-1} \delta_t(dr).$$

*Then $\int_{(s,t]} \alpha_s^M(dr) = I_d$ and ($\alpha$-SC) with $\alpha_s = \alpha_s^M$ gives*

$$u(s, x) = \mathbb{E}\left[\underbrace{\int_s^t \left(M_s(r)\nabla b(r, X_r^u)^\top - \partial_r M_s(r)\right)u(r, X_r^u)\, dr}_{\text{Term 1 in (79)}} + \underbrace{M_s(t)u(t, X_t^u)}_{\text{Term 2 in (79)}} \,\Big|\, X_s^u = x\right].$$

*Thus, under the normalization $M_s(s) = I_d$, the weighted self-consistency property recovers* (79).

*Proof.* Set $K_{r|s} := (J_{r|s}^\top)^{-1}$. We define the cumulative matrix-valued weight

$$\beta_s^M(r) := \begin{cases} I_d - M_s(r)(J_{r|s}^\top)^{-1}, & r \in [s,t), \\ I_d, & r = t. \end{cases}$$

Then $\alpha_s^M = d\beta_s^M$. Since $\partial_r K_{r|s} = -\nabla b(r, X_r^u)^\top K_{r|s}$, we have for $r \in (s,t)$,

$$\partial_r \beta_s^M(r) = -\partial_r\big(M_s(r)K_{r|s}\big) = \Big(M_s(r)\nabla b(r, X_r^u)^\top - \partial_r M_s(r)\Big)K_{r|s}.$$

Moreover,

$$\Delta\beta_s^M(t) = \beta_s^M(t) - \beta_s^M(t-) = I_d - \big(I_d - M_s(t)K_{t|s}\big) = M_s(t)K_{t|s}.$$

Hence

$$\alpha_s^M(dr) = \partial_r \beta_s^M(r)\, \mathbf{1}_{(s,t)}(r)\, dr + \Delta\beta_s^M(t)\, \delta_t(dr),$$

which is the claimed expression. Since $M_s(s) = I_d$ and $K_{s|s} = I_d$, we have $\beta_s^M(s) = 0$. Therefore

$$\int_{(s,t]} \alpha_s^M(dr) = \beta_s^M(t) - \beta_s^M(s) = I_d.$$

Substituting $\alpha_s^M$ into ($\alpha$-SC) gives

$$u(s,x) = \mathbb{E}\left[\int_{(s,t]} \alpha_s^M(dr)\, J_{r|s}^\top u(r, X_r^u)\,\bigg|\, X_s^u = x\right].$$

Using $K_{r|s} J_{r|s}^\top = I_d$, the absolutely continuous part gives

$$\int_s^t \Big(M_s(r)\nabla b(r, X_r^u)^\top - \partial_r M_s(r)\Big)u(r, X_r^u)\, dr,$$

while the atom at $t$ gives

$$M_s(t)K_{t|s}J_{t|s}^\top u(t, X_t^u) = M_s(t)u(t, X_t^u).$$

Combining these two terms yields the claimed identity. $\qquad\square$

# E. Experimental Details and Additional Results

We implement our method using JAX (Bradbury et al., 2018). All experiments were carried out on a Nvidia GeForce RTX 2080Ti GPU unless otherwise specified. For the neural adjustment term $\eta_\theta$, we generally use a gradient-form parameterisation $\eta_\theta = \nabla\psi_\theta$ unless otherwise specified. We parameterise $\psi_\theta$ using the MLP architecture with sinusoidal time-embedding used in De Bortoli et al. (2021), with main hidden dimensions [128,128] and time-embedding hidden dimensions [64,64]. For fair comparison, we also used the same gradient-form architecture for the NGDB and BEL algorithms, which we found performed slightly better than a standard architecture; for SDB and FB we used the standard architecture as this performed better. When training NGDB, we used a Brownian auxiliary process. In all experiments, we use the Adam optimiser (Kingma & Ba, 2015). Below, we describe the individual experiment setups and implementation details used. For the alternative methods, we use the default hyperparameters provided in their respective codebases unless otherwise specified; we provide the links to these codebases in Appendix G. Code for our experiments is available at
`https://github.com/samuel-howard/control_consistency_bridges`.

**Metrics** In several of the experimental settings, we have access to the ground-truth diffusion bridge solution. In these experiments, we report the KL divergence $\mathrm{KL}(\mathbb{P}^*\|\mathbb{P}^\theta)$ relative to this ground-truth, which can be calculated according to the expression

$$\mathrm{KL}(\mathbb{P}^*\|\mathbb{P}^\theta) = \mathbb{E}\Big[\tfrac{1}{2}\int_0^T \|\sigma(t,X_t^*)^\top u^*(t,X_t^*) - \sigma(t,X_t^*)^\top u_\theta(t,X_t^*)\|^2 \mathrm{d}t\Big]. \tag{82}$$

When we do not have access to the ground-truth, we instead report the KL divergence $\mathrm{KL}(\mathbb{P}^\theta\|\mathbb{P})$ relative to the base process. Amongst stochastic processes bridging between the initial and terminal states, the diffusion bridge solution is the one that minimises this divergence, so provided that the obtained solutions transport to the endpoint correctly then this metric provides an indication of how well the problem has been solved. This metric can be computed as

$$\mathrm{KL}(\mathbb{P}^\theta\|\mathbb{P}) = \mathbb{E}\Big[\tfrac{1}{2}\int_0^T \|\sigma(t,X_t^\theta)^\top u_\theta(t,X_t^\theta)\|^2 \mathrm{d}t\Big]. \tag{83}$$

In our experiments, we report values for these metrics computed using 1,000 trajectories. For the SDB method (Heng et al., 2025), we report these values for the forward bridge, which is learned by reversing the learned backwards bridge.

In all experiments, our method trained significantly faster than NGDB (see Table 2); this speed-up can be attributed to the fact that our method does not require backpropagation through the simulated trajectories. Both methods are implemented in JAX, allowing for fair comparison.

### E.1. Ornstein-Uhlenbeck Process

We consider sampling bridges of Ornstein-Uhlenbeck processes of the form

$$\mathrm{d}X_t = -\alpha X_t \mathrm{d}t + \sigma \mathrm{d}B_t, \tag{84}$$

with starting state $x_0 = e_1$, termination state $x_1 = e_2$, termination time $T = 1$, and diffusion coefficient $\sigma = 0.1$. Ornstein-Uhlenbeck bridges are a standard setting in the literature for quantitatively evaluating performance, as the ground-truth control $u^*(t,x)$ can be calculated in closed-form as

$$u^*(t,x) = \frac{2\alpha e^{-\alpha(1-t)}}{(1-e^{-2\alpha(1-t)})\sigma^2}(x_1 - xe^{-\alpha(1-t)}). \tag{85}$$

The training curves shown in Figure 3a are for $\alpha = 2.0$ and $d = 2$. The results in Figure 3b comparing the effect of dimension use $\alpha = 2.0$, and the results comparing the effect of OU parameter in Figure 3c use $d = 2$. To enable a fair comparison between the methods, we use the same gradient-form neural architecture for both, and use the same learning rate, number of gradient update steps, and number of trajectory simulations per step. We used 500 time-discretisation steps, simulated 64 trajectories at each step, and used a learning rate of 1e-4. For Figures 3b and 3c, we ran the algorithms for 10,000 steps; we sometimes found the value for NGDB to start increasing again later in training, so the reported results are the lowest value for NGDB over the 10,000 training iterations.

Both methods learn the true drift to a very high degree of accuracy, and we remark that the difference in KL that is visible between the two methods is predominantly due to the behaviour towards the terminal time $T$. The methods behave comparably earlier in time, but the error of NGDB significantly increases approaching the endpoint. We hypothesise this is a consequence of the differing types of training objective; NGDB uses a KL-based objective, while our approach fits to the desired endpoint behaviour directly in the loss construction so is more accurate towards the terminal time. We were unable to learn the bridge dynamics successfully using the methods SDB, FB, and BEL, which is unsurprising given that the experiments involve rare events under the base dynamics.

### E.2. Cox–Ingersoll–Ross model

The Cox–Ingersoll–Ross (CIR) model (Cox et al., 1985) is a 1-dimensional stochastic process that evolves according to the SDE

$$\mathrm{d}X_t = a(b - X_t)\mathrm{d}t + \varepsilon\sqrt{X_t}\mathrm{d}B_t. \tag{86}$$

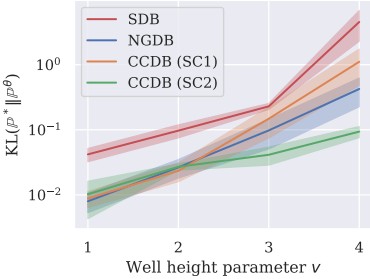

*Figure 5.* Comparison of performance on the double-well experiment, for different barrier height parameters $v$.

The bridges for this process are also known in closed form, as the transition densities can be written as $ce^{-u-v}\left(\frac{v}{u}\right)^{q/2}I_q(2\sqrt{uv})$, where $c = \frac{2a}{(1-e^{-a(T-t))\varepsilon^2}}, q = \frac{2ab}{\varepsilon^2} - 1, u = cxe^{-a(T-t)}, v = cx_T$, and $I_q$ is the modified Bessel function of the first kind of order $q$. This again allows us to report $\mathrm{KL}(\mathbb{P}^*\|\mathbb{P}^\theta)$ and verify that the learned diffusion bridges are correct.

In the experiment, we set $a = b = 1.0$, use a noise parameter of $\varepsilon = 1.0$, and aim to learn the bridge from $x_0 = 2.0$ to $x_1 = 2.0$. For each method, we used 500 time-discretisation steps and 2,000 training steps. For our method and for NGDB, we use a learning rate of 1e-4, and simulated 64 trajectories at each step. For FB and SDB we used a learning rate of 1e-3. When computing the values of $\mathrm{KL}(\mathbb{P}^*\|\mathbb{P}^\theta)$, we truncated the final five steps to avoid numerical issues when computing the ground-truth towards the terminal time.

### E.3. Double-Well Experiment

In the double-well experiments, we consider overdamped Langevin dynamics $\mathrm{d}X_t = -\nabla U(X_t)\mathrm{d}t + \sigma \mathrm{d}B_t$ using a classical double-well potential

$$U(x) = v(x^2 - 1)^2.$$

We aim to traverse from $x_0 = 1$ to $x_1 = -1$ and set $\sigma = 1.0$; this setting was previously considered in Pidstrigach et al. (2025). For each method, we used 500 time-discretisation steps and 4,000 training steps. For our method and NGDB we used a learning rate of 1e-3 and simulated 64 trajectories at each step, and we use the default hyperparameters for SDB and BEL. We note that we were unable to obtain good results for the double-well experiments using the FB method; we hypothesise that this is due to the behaviour of the adjoint process used for the training samples, which Baker et al. (2025) remark can exacerbate the difficulty of learning the correct dynamics for rare events.

The results reported in Table 1 are for a well height of $v = 3.0$, which gives a strong peak between the wells. In Figure 5, we also compare the methods for different heights $v$ of the well barrier. We see that the methods that learn from controlled dynamics (that is, our method and NGDB) outperform SDB which learns from the unconditional dynamics; in particular, the performance of SDB becomes significantly worse for the largest value, which is unsurprising as the uncontrolled simulations will rarely cross between the wells in this case. For lower well heights, our method performs comparably with NGDB for both choices of self-consistency properties (SC1), (SC2). As the well height increases, we see that using (SC2) performs better; this reflects the discussion in Section 4 and demonstrates how avoiding large growth in the Jacobian terms can improve performance.

### E.4. Cell Diffusion Process

The cell differentiation model of Wang et al. (2011) has previously been used to provide a qualitative evaluation of diffusion bridge methods in Heng et al. (2025); Baker et al. (2025); Yang et al. (2025). The model dynamics are given by the 2-dimensional SDE

$$\mathrm{d}X_t = \begin{bmatrix} \frac{X_{t,1}^4}{2^{-4}+X_{t,1}^4} + \frac{2^{-4}}{2^{-4}+X_{t,2}^4} - X_{t,1} \\ \frac{X_{t,2}^4}{2^{-4}+X_{t,2}^4} + \frac{2^{-4}}{2^{-4}+X_{t,1}^4} - X_{t,2} \end{bmatrix} \mathrm{d}t + \sigma \, \mathrm{d}B_t. \tag{87}$$

This provides an example in which the reference drift $b$ is not of gradient form, so following the discussion in Section 3 we include the reference drift in our parameterisation. We consider the three settings used in Yang et al. (2025): a *normal* event,

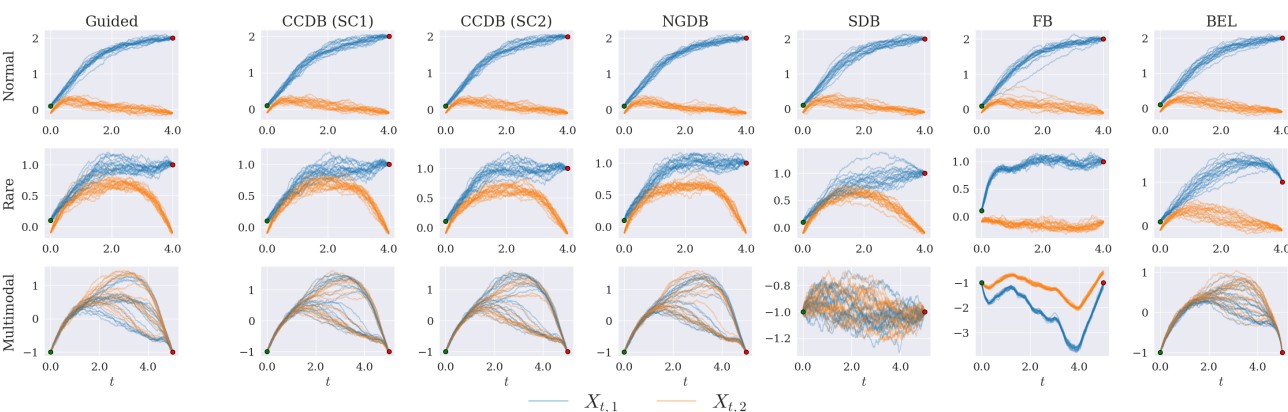

*Figure 6.* Visualisation of the learned trajectories for each method in the *normal*, *rare*, and *multimodal* cell diffusion examples. The experimental setting is taken from Yang et al. (2025), using the cell differentiation model from Wang et al. (2011). Results in the leftmost column are obtained using the MCMC-based guided proposal approach from Schauer et al. (2017).

a *rare* event, and a *multi-modal* example. For the normal event, the starting state is $x_0 = [0.1, -0.1]$, the termination state is $x_T = [2.0, -0.1]$, and the termination time is $T = 4$. For the rare event, the starting state is again $x_0 = [0.1, -0.1]$, the termination state is $x_T = [1.0, -0.1]$, and the termination time is again $T = 4$. For the multi-modal example, the starting state is $x_0 = [-1.0, -1.0]$, the termination state is $x_T = [-1.0, -1.0]$, and the termination time is $T = 5$. In all settings, the diffusion coefficient is $\sigma = 0.1$, and we train for 1000 steps. For our approach and NGDB, we use a learning rate of 1e-3, and otherwise use the default parameters for the alternative methods.

As there is no ground-truth available in this setting, in Table 1 we report the KL divergence $\mathrm{KL}(\mathbb{P}^\theta \| \mathbb{P})$ relative to the base process. For a qualitative comparison, we additionally provide visualisations of the paths obtained by each method in the different cases in Figure 6. In the plot we also include a comparison with a non-ML baseline, displaying samples obtained using the MCMC-based guided proposal method from Schauer et al. (2017). We use the implementation and hyperparameters from Yang et al. (2025) (other than for the multimodal example, for which we used $\eta = 0.995$ and ran several MCMC chains).

From the trajectories in Figure 6, we see that CCDB performs very similarly in each setting to NGDB, which was shown in Yang et al. (2025) to perform strongly in these experiments. Both methods accurately approximate the correct dynamics, even in the rare and multimodal examples. In contrast, the methods that learn from uncontrolled dynamics perform well in the normal setting, but struggle in the other cases as expected.

### E.5. Müller-Brown Potential

We follow the experimental setup used in Du et al. (2024). The dynamics follow the overdamped Langevin SDE $\mathrm{d}X_t = -\nabla U(X_t)\mathrm{d}t + \sigma \mathrm{d}B_t$, where the potential $U$ is defined as

$$U(x, y) = -200 \exp\left(-(x-1)^2 - 10y^2\right) \tag{88}$$

$$-100 \exp\left(-x^2 - 10(y - 0.5)^2\right) \tag{89}$$

$$-170 \exp\left(-6.5(0.5 + x)^2 + 11(x + 0.5)(y - 1.5) - 6.5(y - 1.5)^2\right) \tag{90}$$

$$+15 \exp\left(0.7(1 + x)^2 + 0.6(x + 1)(y - 1) + 0.7(y - 1)^2\right), \tag{91}$$

with initial state $x_0 = [-0.559, 1.442]$, terminal state $x_T = [0.624, 0.028]$, diffusion term $\sigma(t, x) = 3\,\mathrm{Id}$, and termination time $T = 0.05$.

We used 1000 discretisation steps. For our method and for NGDB, we ran for 4000 training steps, simulated 64 trajectories at each outer step, and used a learning rate of 3e-4; for DL (Du et al., 2024) we use the default parameters provided in their codebase. For our method, we clipped the losses at 1e4, which we found to improve stability.

This experiment provides another example showcasing the benefit of using the self-consistency property (SC2) in our algorithm, compared to using the first version (SC1). This helps to avoid large growth in the Jacobian terms caused by the

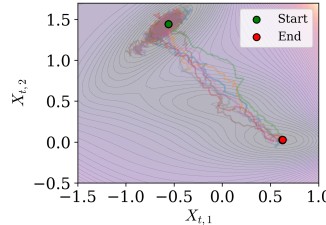 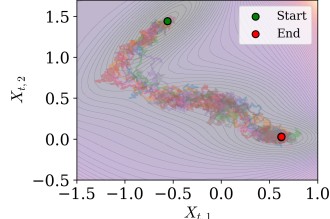 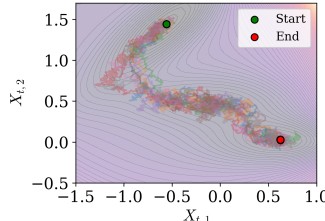

*(a)* Using the self-consistency property (SC1) gives unstable learning due to the large Jacobian terms, and thus fails to learn the correct dynamics.

*(b)* Using the self-consistency property (SC2) avoids large growth in the Jacobians, enabling successful learning of the correct dynamics.

*(c)* Similar performance in $100d$ Müller-Brown example suggests increased dimensionality does not degrade performance when intrinsic difficulty remains unchanged.

*Figure 7.* Comparison of the trajectories obtained using our method, in the Müller-Brown examples.

*Table 4.* Comparison of results for the $100d$ Müller–Brown experiment. Metrics are evaluated using the leading two dimensions corresponding to the Müller–Brown potential. The values for $\text{KL}(\mathbb{P}^\theta\|\mathbb{P})$ show mean±std over 5 runs, while the Max Energy values report the average mean and average std over the 5 runs.

|  | $\text{KL}(\mathbb{P}^\theta\|\mathbb{P}) \downarrow$ | Max Energy |
|---|---|---|
| Ours (using (SC2)) | 27.0±1.2 | -35.2±5.13 |
| NGDB (Yang et al., 2025) | 29.1±0.8 | -33.7±5.8 |
| DL (Du et al., 2024) | 39.2±0.4 | -32.0±6.2 |

steep potential. Using (SC1) generally caused learning to be unstable and usually fail, other than for some specific choices of $\alpha$-schedule (see the trajectories in Figure 7). In contrast, using (SC2) gave consistent and reliable training.

**$100d$ Müller-Brown Example**   To assess performance in higher dimensions, we also report results for an augmented Müller-Brown potential in 100 dimensions (following a similar $5d$ example in Sipka et al. (2023)). The first two dimensions match the standard Müller-Brown potential above, and the added dimensions follow a quadratic potential. We use a larger network with hidden dimensions [256,256,256] and time-embedding hidden dimensions [128,128], and train each method for 8000 steps; otherwise, we use the same hyperparameters as above.

For the obtained paths, in Table 4 we report the same quantitative results as in Table 3. To make comparable with those in Table 3, we use the first two components of the obtained trajectories to evaluate the metrics, so the additional dimensions only influence training and not evaluation (otherwise the KL computations are dominated by the behaviour in the added dimensions). Trajectories for the first two dimensions using our method are visualised in Figure 7c.

We find that our method displays largely similar performance in this higher-dimensional example compared to the standard $2d$ case, and the ability to learn the bridge does not appear to deteriorate significantly. This suggests the difficulty of the problem is primarily governed by the complexity of the transition path rather than the ambient dimensionality, and that our approach appears not to be substantially impacted by increases in dimensionality when the underlying problem structure remains unchanged.

### E.6. Alanine Dipeptide

**Molecular Dynamics**   Transition path sampling aims to learn the dynamics of molecules conditioned on transitioning between two metastable states. The behaviour of the molecule is typically modelled using overdamped or underdamped Langevin dynamics; here we focus on overdamped dynamics as our framework considers elliptic diffusions, and remark that extending our method to hypoelliptic settings is an important direction for future work. The overdamped dynamics evolve according to the SDE,

$$\mathrm{d}X_t = -(\gamma M)^{-1}\nabla U(X_t)\mathrm{d}t + (\gamma M)^{-\frac{1}{2}}\sqrt{2k_B T} \cdot \mathrm{d}B_t, \tag{92}$$

where $M$ denotes the mass vector, $\gamma$ denotes the friction coefficient, $T$ is the temperature, and $k_B$ is the Boltzmann constant.

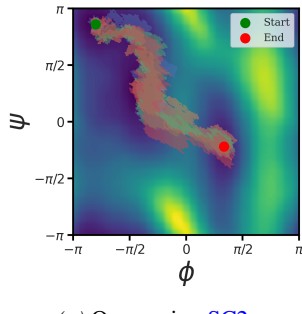

*(a)* Ours, using SC2

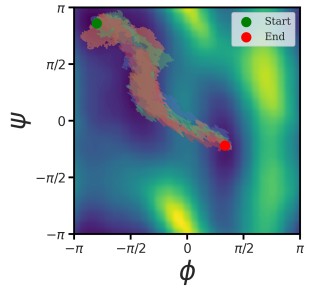

*(b)* NGDB (Yang et al., 2025)

*Figure 8.* Visualisations of the obtained trajectories transitioning from the $C5$ state (upper left) to the $C7ax$ state (lower right), displayed on Ramachandran plots showing the dihedral angles. *(Left)* Our method, using (SC2). *(Right)* NGDB.

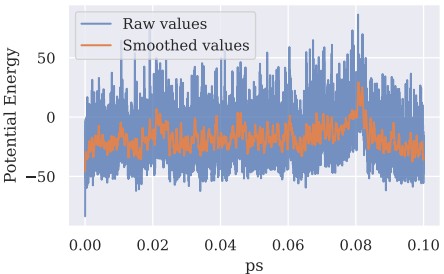

*Figure 9.* Comparison of the raw potential evaluations and smoothed potential evaluations along a single trajectory.

*Table 5.* Comparison of energy transition state (ETS) values for the alanine dipeptide experiment. The values for PIPS and TPS-DPS are those reported in Seong et al. (2025).

|  |  | ETS (kJmol$^{-1}$) |
|---|---|---|
| Overdamped | Ours (using (SC2)) | 28.7±5.8 |
|  | NGDB (Yang et al., 2025) | 26.5±6.8 |
| Underdamped | PIPS (Holdijk et al., 2023) | 28.2±10.9 |
|  | TPS-DPS (Seong et al., 2025) | 18.4±10.9 |

**Experimental Details** We consider transitioning between the $C5$ and $C7ax$ conformational states of alanine dipeptide, a small 22-atom molecule commonly used to evaluate transition path sampling methods. Following the experimental setup of Du et al. (2024), we use an AMBER14 forcefield (Maier et al., 2015) implemented in OpenMM (Eastman et al., 2017). We use DMFF (Wang et al., 2023) to allow for differentiation through the forcefield, enabling efficient implementation in JAX. The initial and final states are taken to be those used in Seong et al. (2025), which are located at the respective local minima of the potential and are aligned using the Kabsch algorithm (Kabsch, 1976). We use a temperature of $300K$, and friction coefficient of $\gamma = 1.0$. As we use overdamped dynamics rather than the more common underdamped setting usually considered in TPS works, our dynamics are considerably rougher and we are required to make small time steps to avoid numerical issues. We therefore consider a time interval of $T = 0.1ps$ and make 30,000 discretisation steps, which we found to avoid numerical issues while being sufficiently long as to not distort the obtained transition paths.

We include the base drift in our parameterisation, to help avoid regions of high energy which would cause numerical issues. To encourage departure from the original basin, we scaled the Brownian drift term by a factor of 4, and clipped this guiding term towards the endpoint to avoid numerical issues in the energy evaluation. We remark that different scaling factors caused the dynamics to transition at different times within the interval, but all traced out similar paths between the endpoints. We use the average $\alpha$-schedule, a learning rate of 1e-4, and use (SC2) to avoid large Jacobian terms, following our findings in the previous experiments. At each iteration, we simulate 16 paths, and perform 10 inner gradient steps on these samples. We train for 100 iterations, amounting to 1000 gradient steps. Training took approximately 15 minutes.

We also apply NGDB in this setting. Because of the added computational requirements, we run for 400 training steps and use a Nvidia A100 GPU to avoid memory issues (note that NGDB cannot make multiple gradient updates on the same samples). Training took 3 hours, which is significantly longer than our method despite the more powerful hardware; this clearly demonstrates the scalability advantages of our approach. We remark that we were unable to obtain reasonable paths with the Doob's Lagrangian method; this may be due to the fact that during inference, errors can accumulate in the sampling procedure causing numerical issues in the simulation (note that simulations are not used during training for this method).

**Discussion of Results** In Figure 8, we provide Ramachandran plots for the obtained paths, which visualise how the backbone dihedral angles (denoted $\phi, \psi$) evolve during the trajectories. As we consider overdamped Langevin dynamics,

we cannot directly compare our results to the underdamped setting considered in works that specifically focus on TPS. Nevertheless, we can qualitatively compare the obtained trajectories to see if our approach provides physically plausible paths. We see that the two methods give similar paths, and that the paths are consistent with existing results in the literature (see, for example, similar plots in (Seong et al., 2025)). Both methods, however, appear to learn only one of the modes; similar behaviour is also reported by other methods in Seong et al. (2025). We note that overdamped dynamics may give different results to underdamped dynamics, as molecules do not have momentum so are less likely to traverse certain paths. We remark that the difference between trajectories obtained by our method and NGDB in Figure 8 is primarily due to the difference in the drift parameterisation, as NGDB uses the guiding proposal from Schauer et al. (2017) which results in the transition occurring later in the path trajectory.

In Table 5, we also report the energy of the transition states (ETS) for the obtained trajectories, which is the maximum value of the potential energy $U$ along the path. As we use overdamped dynamics, there is a large amount of noise in the trajectories and thus in the potential evaluations. To obtain meaningful ETS values, we therefore constrain the bond lengths (as is common practice in molecular dynamics simulations), and also smooth the noisy energy evaluations. To do so, we use a Savitzky–Golay filter with window length 201; we find this choice to not overly distort the trajectory shape (see Figure 9) and be fairly robust to the choice of window length. Nevertheless, care should be taken interpreting these values because of the overdamped setting. In Table 5, we also provide values for existing strongly performing TPS methods PIPS (Holdijk et al., 2023) and TPS-DPS (Seong et al., 2025) in the underdamped setting, using the values reported in Seong et al. (2025). While not directly comparable, their similarity again suggests that our obtained paths are reasonable.

**Future Work**   The transition path sampling examples presented here demonstrate the increased scalability of our approach compared to existing diffusion bridge algorithms, and show that our method remains stable in these more challenging settings. Extending our methodology to hypoelliptic diffusions to allow for using underdamped dynamics, as well as incorporating domain-specific implementation details, may enable extending to more complex systems. In particular, the examples presented here could benefit from a good initialisation, and extensions to harder problems may require improved initial paths such as those obtained from IDPP (Smidstrup et al., 2014) (as used in Lee et al. (2025); Park et al. (2025)). Other implementation improvements could employ an annealing schedule (Seong et al., 2025), utilise improved architectures, train on shorter sliding window intervals (Lee et al., 2025), or incorporate trust-regions (Blessing et al., 2025). Investigating such extensions to our methodology for targeting TPS problems specifically is a promising direction for future work.

### E.7. $\alpha$-Schedule Experiments

In Table 6, we perform the same experiments as in the main body, and show the results for the *next-step*, *average*, and *endpoint* $\alpha$-schedules. We use the same hyperparameters as in the main body (aside from adding loss clipping of 10 for the CIR model, for (SC1) with next-step $\alpha$-schedule). Generally, we find the average schedule to perform strongest so we report those results in the main body; it outperforms the endpoint schedule in all examples, and next-step prediction in the majority. In cases where the Jacobian can lead to large target variance (like in the double-well example, when using (SC1)), next-step prediction performs better than average prediction, which is reflective of the discussion in Section 4.

*Table 6.* Quantitative results for experiments, comparing $\alpha$-schedules (mean $\pm$std, over 5 runs).

| | CIR model, $\mathrm{KL}(\mathbb{P}^*\|\mathbb{P}^\theta)$ ↓ | Double Well, $v = 3.0$, $\mathrm{KL}(\mathbb{P}^*\|\mathbb{P}^\theta)$ ↓ | Cell Diffusion, $\mathrm{KL}(\mathbb{P}^\theta\|\mathbb{P})$ ↓ | | |
| --- | --- | --- | --- | --- | --- |
| | | | Normal | Rare | Multimodal |
| Using (SC1), next-step | 0.021±0.007 | 0.126±0.069 | 7.38±0.49 | 74.6±14.1 | 959.9±55.3 |
| Using (SC1), average | 0.023±0.010 | 0.148±0.073 | 6.53±0.14 | 66.1±0.9 | 792.8±0.9 |
| Using (SC1), endpoint | 0.026±0.005 | 3.49±2.40 | 6.79±0.10 | 69.1±1.1 | 793.0±1.0 |
| Using (SC2), next-step | 0.042±0.022 | 0.067±0.044 | 6.95±0.29 | 68.6±1.9 | 1136.2±288.9 |
| Using (SC2), average | 0.055±0.009 | 0.041±0.013 | 6.53±0.17 | 66.0±0.5 | 792.4±1.0 |
| Using (SC2), endpoint | 0.068±0.027 | 0.278±0.101 | 7.64±0.59 | 67.4±0.9 | 795.5±3.2 |

### E.8. Sticking-the-Landing Adjustments

We now compare performance of our algorithm with and without the STL adjustments discussed in Appendix B.3. We use the same hyperparameters as in the previous experiments, and report the results in Table 7. Including the STL adjustments generally appears to improve performance slightly, with the largest improvements seen in cases where large Jacobians arise (that is, the double-well and Müller-Brown experiments). However, as the STL adjustments require computing the terms $(\nabla u\,\sigma + u\,\nabla\sigma)(t, X_t^u)\cdot\delta B_t$, they increase the computational cost of the method and training steps are approximately 1.5 times slower.

*Table 7.* Quantitative results for experiments, comparing performance with and without STL adjustments (mean $\pm$std, over 5 runs).

| | CIR model, $\text{KL}(\mathbb{P}^*\|\mathbb{P}^\theta)\ \downarrow$ | Double Well, $v=3.0$, $\text{KL}(\mathbb{P}^*\|\mathbb{P}^\theta)\ \downarrow$ | Cell Diffusion, $\text{KL}(\mathbb{P}^\theta\|\mathbb{P})\ \downarrow$ | | |
| --- | --- | --- | --- | --- | --- |
| | | | Normal | Rare | Multimodal |
| Using (SC1), without STL | 0.023±0.010 | 0.148±0.073 | 6.53±0.14 | 66.1±0.9 | 792.8±0.9 |
| Using (SC1), with STL | 0.016±0.001 | 0.042±0.024 | 6.47±0.16 | 65.7±0.6 | 792.4±1.0 |
| Using (SC2), without STL | 0.055±0.009 | 0.041±0.013 | 6.53±0.17 | 66.0±0.5 | 792.4±1.0 |
| Using (SC2), with STL | 0.051±0.007 | 0.022±0.015 | 6.48±0.14 | 66.1±0.5 | 792.6±0.8 |

### E.9. Neural Parameterisation

Recall that there is a large degree of flexibility in the choice of neural parameterisation. In the experiments reported in Table 1, we use the standard parameterisations outlined in Section 3 and Appendix B.1, and in particular enforce the gradient-form property throughout time by setting the neural adjustment to be of form $\eta_\theta = \nabla\psi_\theta$. One can also use a general neural vector field $\eta_\theta$; in Table 8 we report results for the general parameterisation and compare to those using $\eta_\theta = \nabla\psi_\theta$ from the main paper.

*Table 8.* Quantitative results for experiments, comparing parameterisations using a general neural adjustment $\eta_\theta$ with a gradient-form neural adjustment $\eta_\theta = \nabla\psi_\theta$ (mean $\pm$std, over 5 runs).

| | CIR model, $\text{KL}(\mathbb{P}^*\|\mathbb{P}^\theta)\ \downarrow$ | Double Well, $v=3.0$, $\text{KL}(\mathbb{P}^*\|\mathbb{P}^\theta)\ \downarrow$ | Cell Diffusion, $\text{KL}(\mathbb{P}^\theta\|\mathbb{P})\ \downarrow$ | | |
| --- | --- | --- | --- | --- | --- |
| | | | Normal | Rare | Multimodal |
| Using (SC1), with $\eta_\theta = \nabla\psi_\theta$ | 0.023±0.010 | 0.148±0.073 | 6.53±0.14 | 66.1±0.9 | 792.8±0.9 |
| Using (SC1), with general $\eta_\theta$ | 0.025±0.009 | 0.080±0.033 | 6.49±0.12 | 66.2±0.6 | 979.5±2.0 |
| Using (SC2), with $\eta_\theta = \nabla\psi_\theta$ | 0.055±0.009 | 0.041±0.013 | 6.53±0.17 | 66.0±0.5 | 792.4±1.0 |
| Using (SC2), with general $\eta_\theta$ | 0.057±0.008 | 0.028±0.013 | 6.51±0.15 | 66.1±0.5 | 792.8±1.1 |

### E.10. Sensitivity to Choice of Self-Consistency Property

Recall from Section 4 and Appendix C that we can obtain a family of self-consistency properties (SC-gen) indexed by an additional drift $\tilde{b}$ included in an 'auxiliary' process. The property (SC1) corresponds to $\tilde{b} = 0$, and (SC2) corresponds to $\tilde{b} = -b$. As in the previous experiments, we would expect the different objectives to perform largely similarly in settings without significant growth in the Jacobians, but for there to be a larger change in performance when the Jacobian terms become larger. To further illustrate this effect, we here provide some additional results for the double well experiment, in which we vary the height of the well barrier and interpolate between the objectives (SC1) and (SC2).

We also remark that one can interpret (SC2) as a more 'damped' self-consistency objective compared to (SC1), as the running costs include the current control. (SC1) gives more aggressive updates, which can accelerate training in well-behaved settings but at a possible cost to stability in settings with large Jacobians. Below we show the KL divergence to the ground truth during training in the OU experiment. We see that (SC1) does indeed display slightly faster convergence to the solution than (SC2), consistent with this intuition.

*Table 9.* Results for different choices of $\tilde{b}$ interpolating from (SC1) to (SC2), for different barrier heights $v$ in the double-well experiment (mean $\pm$ std over 5 runs, using average $\alpha$-schedule, 1e-4 learning rate, and 2000 training steps).

|  | $v = 2$ | $v = 3$ | $v = 4$ |
|---|---|---|---|
| $\tilde{b} = 0$ (SC1) | $0.039 \pm 0.011$ | $0.204 \pm 0.100$ | $0.584 \pm 0.346$ |
| $\tilde{b} = -\frac{1}{3}b$ | $0.036 \pm 0.012$ | $0.091 \pm 0.032$ | $0.192 \pm 0.045$ |
| $\tilde{b} = -\frac{2}{3}b$ | $0.034 \pm 0.006$ | $0.072 \pm 0.026$ | $0.146 \pm 0.033$ |
| $\tilde{b} = -b$ (SC2) | $0.035 \pm 0.006$ | $0.081 \pm 0.004$ | $0.153 \pm 0.016$ |

*Table 10.* Convergence speed of training objectives (SC1) and (SC2) in the Ornstein-Uhlenbeck example (KL divergence to ground-truth, mean over 5 runs).

| Training step | 200 | 400 | 600 | 800 | 1000 |
|---|---|---|---|---|---|
| Using (SC1) | 0.12 | 0.078 | 0.067 | 0.052 | 0.046 |
| Using (SC2) | 0.15 | 0.088 | 0.070 | 0.061 | 0.055 |

## F. Connections to Stochastic Optimal Control Methods

To conclude, we draw connections between our self-consistency framework and recent developments in the stochastic optimal control (SOC) and generative modelling literatures (Domingo-Enrich et al., 2024; 2025; Domingo-Enrich, 2024), aiming to clarify their relations and provide new perspective on these families of training objectives. For a 'running cost' function $f : [0, T] \times \mathbb{R}^d \to \mathbb{R}$ and terminal cost $g : \mathbb{R}^d \to \mathbb{R}$, the quadratic optimal control problem aims to find the control drift $u : [0, T] \times \mathbb{R}^d \to \mathbb{R}$ that minimises

$$\mathbb{E}\left[ \int_0^T \tfrac{1}{2}\|(\sigma^\top u)(t, X_t^u)\|^2 + f(t, X_t^u)\,\mathrm{d}t + g(X_T^u) \right], \tag{93}$$

where $X_t^u$ evolves according to the controlled dynamics (3). It can be shown (see, for example, Nüsken & Richter (2021)) that the solution corresponds to a reweighted version of the reference path measure

$$\frac{\mathrm{d}\mathbb{Q}}{\mathrm{d}\mathbb{P}}(X) \propto \exp(-\mathcal{W}(X)), \tag{94}$$

where $\mathcal{W}$ is the 'work' functional given by $\mathcal{W}(X) = \int_0^T f(t, X_t)\,\mathrm{d}t + g(X_T)$. Intuitively, this corresponds to an exponential reweighting according to the 'cost' of an individual trajectory. The optimal control of the solution can be written as an expectation according to the uncontrolled process (Kappen, 2005),

$$u^*(s, x) = \nabla \log \mathbb{E}_{\mathbb{P}}\left[ \exp\left( -\int_s^T f(t, X_t)\,\mathrm{d}t - g(X_T)\right) | X_s = x \right]. \tag{95}$$

Note that this is often written as $u(s, x) = -\nabla V(s, x)$ for the value function $V$. We remark that this presentation differs from the usual presentation in that the control $u$ is rescaled by a factor of $\sigma$; we use this notation here because it is more natural for the self-consistency properties we present below.

Heuristically, the diffusion bridge problem considered in this work can be considered as a limiting case of this general SOC problem, in which we have zero running costs $f = 0$ and the terminal cost $g$ approaches infinity everywhere other than the desired endpoint $x_T$.

**Combining Self-Consistency with Terminal Conditions** In this paper, we showed how the diffusion bridge can be characterised by two properties—a *self-consistency property*, which relates the value of the optimal control $u^*$ between times $s$ and $t$, and a *terminal condition*, which enforces known information about the problem. By combining a chosen self-consistency property with the terminal condition, we obtain a wide class of algorithms for learning the diffusion bridge.

This self-consistency framework also provides an analogous class of algorithms for the SOC problem (93), and in fact this class recovers several recently proposed algorithms as special cases. We derive the corresponding self-consistency properties

below, which now incorporate the additional running cost $f$. The proofs proceed as in Appendix D, with the appropriate minor modifications caused by the inclusion of $f$ in the HJB equation.

For diffusion bridges, recall that the terminal information was enforced by jumping to the known endpoint at the final step. For the SOC problem in (93), this is instead replaced by setting the terminal condition $u(T, X_T) = \nabla g(X_T)$.

### F.1. Self-Consistency Property for SOC

The self-consistency properties (SC1), (SC2), (SC-gen) hold for any reweighting $\frac{d\mathbb{Q}}{d\mathbb{P}}(X) \propto F(X_T)$, so they hold for the SOC problem (93) with $f = 0$ by setting $F = e^{-g}$. By following the same derivation, we can obtain a similar property for the general SOC problem (93) with non-zero running costs. Defining $\mathcal{W}_{s,T} = \int_s^T f(r, X_r)dr + g(X_T)$, we have

$$u^*(s,x) = \nabla \log \mathbb{E}_{\mathbb{P}}\Big[\exp\big(-\mathcal{W}_{s,T}\big)|X_s = x\Big] \tag{96}$$

$$= \frac{1}{\mathbb{E}_{\mathbb{P}}\big[\exp\big(-\mathcal{W}_{s,T}\big)|X_s = x\big]}\nabla_x\mathbb{E}_{\mathbb{P}}\Big[\exp\big(-\mathcal{W}_{s,T}\big)|X_s = x\Big] \tag{97}$$

$$= \frac{1}{\mathbb{E}_{\mathbb{P}}\big[\exp\big(-\mathcal{W}_{s,T}\big)|X_s = x\big]}\mathbb{E}_{\mathbb{P}}\Big[\exp\big(-\mathcal{W}_{s,T}\big)\cdot\big(-\int_s^T J_{r|s}^\top \nabla f(r, X_r)dr - J_{T|s}^\top \nabla g(X_T)\big)|X_s = x\Big] \tag{98}$$

$$= \mathbb{E}_{\mathbb{Q}}[-\int_s^T \nabla f(r, X_r)J_{r|s}dr - \nabla g(X_T)J_{T|s}|X_s = x], \tag{99}$$

where in the last step we used Lemma D.1. Inverting the relation at a later time $t$ as in the proof of Theorem 2.1, we obtain the following *self-consistency property for SOC*, which compared to (SC1) now also includes the running cost function $f$,

**Theorem F.1** (Self-consistency for SOC). *The control drift $u^*$ of the SOC solution satisfies,*

$$u^*(s,x) = \mathbb{E}\left[-\int_s^t J_{r|s}^\top \nabla f(r, X_r^{u^*})dr + J_{t|s}^\top u^*(t, X_t^{u^*})|X_s^{u^*} = x\right] \qquad \text{for } s < t, \tag{100}$$

*along with the known terminal condition $u^*(T, x) = \nabla g(x)$.*

**Adjoint Matching** In particular, the recently proposed *adjoint matching* (Domingo-Enrich et al., 2025) method is recovered by setting $t = T$ in this self-consistency relation; their 'lean adjoint' $\tilde{a}_s$ is equal to $\int_s^T J_{r|s}^\top \nabla f(r, X_r^u)dr - J_{T|s}^\top \nabla g(X_T^u)$. This method has proven highly effective for fine-tuning of flow-based generative models, and has also been extended to neural sampling settings (Havens et al., 2025; Liu et al., 2025; Choi et al., 2025).

### F.2. Generalised Self-Consistency Property for SOC

As in the presentation in Section 4, we can perform a change of measure before interchanging the derivative and expectation in (98), so that we obtain Jacobian terms of an auxiliary process $d\tilde{X}_t = (b+\tilde{b})(t, \tilde{X}_t)dt + \sigma(t, \tilde{X}_t)dB_t$ rather than the base process $X$. Following the same derivation as in the main presentation, we obtain the generalised self-consistency property for SOC (analogous to (SC-gen)), which similarly to above now also includes the running cost function $f$.

**Theorem F.2** (Generalised self-consistency for SOC). *For any differentiable $\tilde{b}$ satisfying the conditions of Girsanov's theorem, the control drift $u^*$ of the SOC solution satisfies,*

$$u^*(s,x) = \mathbb{E}\left[-\int_s^t \tilde{J}_{r|s}^\top \Big(\nabla f(r, X_r^{u^*}) + \nabla\tilde{b}(r, X_r^{u^*})^\top u^*(r, X_r^{u^*})\Big)dr + \tilde{J}_{t|s}^\top u^*(t, X_t^{u^*})|X_s^{u^*} = x\right] \qquad \text{for } s < t, \tag{101}$$

*along with the known terminal condition $u^*(T, x) = \nabla g(x)$, where $\tilde{J}_{t|s}$ is associated with the auxiliary process $\tilde{X}_t$.*

Taking $\tilde{J}_{t|s} = \text{Id}$, we recover the SOC analogue of Equation (SC2),

$$u^*(s,x) = \mathbb{E}\left[\int_s^t \Big(-\nabla f(r, X_r^{u^*}) + \nabla b(r, X_r^{u^*})^\top u^*(r, X_r^{u^*})\Big)dr + u^*(t, X_t^{u^*})|X_s^{u^*} = x\right] \qquad \text{for } s < t. \tag{102}$$

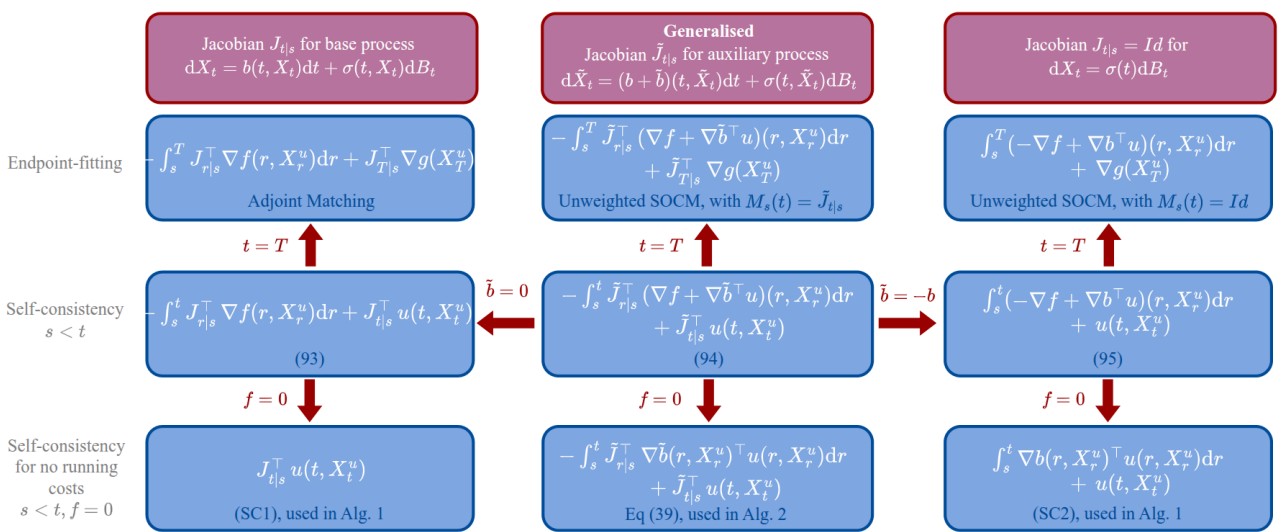

*Figure 10.* Connections between self-consistency properties for stochastic optimal control, showing the integrand $\Phi(X^u)$ such that $u(s, X^u_s) = \mathbb{E}[\Phi(X^u)|X^u_s = x]$. *(Left)* The expressions contain the Jacobian process $J_{t|s}$ for the base dynamics $X$, analogous to (SC1) from the main paper. *(Middle)* Generalised versions, containing the Jacobian $\tilde{J}_{t|s}$ for an auxiliary process $\tilde{X}$. These are analogous to the generalised self-consistency property in (SC-gen). *(Right)* The expressions take $J_{t|s} = \text{Id}$ for the base dynamics $X$, analogous to (SC2) from the main paper. *Top line:* When restricted to the terminal time $t = T$, these self-consistency relations recover recently introduced objectives in the SOC and generative modelling literature (Domingo-Enrich et al., 2025; Domingo-Enrich, 2024). *Bottom Line:* In the case of zero running costs, the SOC self-consistency properties coincide with those presented in the main paper, which form the basis of our method for learning diffusion bridge dynamics.

**Connection to Reparameterisation Matrices**    As in the discussion in Appendix D.3 and Appendix D.4, the property (101) can be further generalised to incorporate a wider matrix family $M_s(t)$ by making the appropriate adjustments to the proof of Theorem C.1. We again focus our presentation here on considering $M_s(t) = \tilde{J}^\top_{t|s}$ to retain their probabilistic interpretation. We comment here on how this general form of the self-consistency property relates to the family of 'reparameterisation matrices' used in Domingo-Enrich et al. (2024). In Domingo-Enrich et al. (2024), the reparameterisation matrices arise in the following 'path-wise reparameterisation trick for SOC'.

**Proposition F.3** (Proposition 1, Domingo-Enrich et al. (2024)). *For each $t \in [0, T]$, let $M_t : [t, T] \to \mathbb{R}^{d \times d}$ be an arbitrary continuously differentiable matrix-valued function such that $M_t(t) = \text{Id}$. We have that*

$$\nabla_x \mathbb{E}_\mathbb{P}\left[ \exp\left( -\int_s^T f(t, X_t)\mathrm{d}t - g(X_T)\right)|X_s = x\right] = \mathbb{E}_\mathbb{P}\Bigg[ \Bigg( -\int_s^T M_s(t)\nabla f(t, X_t)\mathrm{d}t - M_s(T)\nabla g(X_T)$$
$$+ \int_s^T (M_s(t)\nabla b(t, X_t) - \partial_t M_s(t))(\sigma^{-1})^\top \mathrm{d}B_t \Bigg)$$
$$\times \exp\left( -\int_s^T f(t, X_t)\mathrm{d}t - g(X_T)\right)|X_s = x\Bigg].$$
$$(103)$$

Note that the connection between this result and the above generalised self-consistency property for SOC (101) arises by taking $t = T$ and choosing the reparameterisation matrices to be the auxiliary Jacobian $M_s(t) = \tilde{J}^\top_{t|s}$. Indeed, recalling the Jacobian dynamics (5), we see that

$$M_s(t)\nabla b(t, X_t) - \partial_t M_s(t) = \tilde{J}^\top_{t|s}\nabla b(t, X_t) - \partial_t \tilde{J}^\top_{t|s} = -\tilde{J}^\top_{t|s}\nabla \tilde{b}(t, X_t). \quad (104)$$

We can substitute this into (103), and by making the change of measure from $\mathbb{P}$ to $\mathbb{P}^{u^*}$ (recalling that $\mathrm{d}B^\mathbb{P}_r = \mathrm{d}B^\mathbb{Q}_r + \sigma u^* \mathrm{d}t$, and that the stochastic integral has zero expectation), we obtain (101) for the case $t = T$.

**Connection to Unweighted SOCM**    Proposition F.3 is used in Domingo-Enrich et al. (2024) to derive the SOCM objective, which also includes exponential importance weights. Unfortunately, these terms cause prohibitively high variance for larger

scale problems. In Domingo-Enrich (2024, Section 3.2), it is proposed to optimise an *unweighted SOCM* loss without these importance weights, though the resulting algorithm is reported to have weak theoretical guarantees. From the discussion above, we see that optimising for the generalised self-consistency property (101) corresponds to the unweighted SOCM loss in which $M_s(t) = \tilde{J}_{t|s}$. Crucially, our analysis shows we do not compromise the theoretical guarantees of the resulting algorithm. The generalised form (in particular, setting $\tilde{b} = -b$ as in (SC2)) provided significant benefits for our diffusion bridge algorithm in many settings; we anticipate that similar benefits may also hold for more general SOC problems, and investigating such losses is a promising direction for future work.

**Summary of Connections**    We illustrate the relationships discussed above in Figure 10, to help clarify how the various training objectives arise from the shared self-consistency framework, and to shed light on the connections and intuition behind them. Moreover, the self-consistency properties hold for general $s < t$, so give rise to a broad family of objectives by time-weighting according to chosen $\alpha$-schedules. This is crucial for the diffusion bridge setting studied in this work due to the singular behaviour at the endpoint, but may additionally be advantageous for SOC problems of form (93). Finally, the generalised form of the self-consistency property (101) (and indeed the corresponding forms for the wider matrix family $M_s(t)$) provides a wider class of theoretically-grounded algorithms, which may also provide benefits beyond the diffusion bridge setting considered in this work.

# G. Licenses

The following assets were used in this work.

- Neural Guided Diffusion Bridges (NGDB) (Yang et al., 2025),
  `https://github.com/gefanyang/neuralbridge`

- Simulating Diffusion Bridges with Score Matching (SDB) (Heng et al., 2025),
  `https://github.com/jeremyhengjm/DiffusionBridge`

- Score matching for bridges without learning time-reversals (FB) (Baker et al., 2025), MIT License
  `https://github.com/libbylbaker/forward_bridge`

- Conditioning Diffusions Using Malliavin Calculus (BEL) (Pidstrigach et al., 2025), MIT License
  `https://github.com/jakiw/conditioning-diffusions`

- Doob's Lagrangian (DL) (Du et al., 2024), MIT License
  `https://github.com/plainerman/Variational-Doob`

- TPS-DPS (Seong et al., 2025)
  `https://github.com/kiyoung98/tps-dps`

- OpenMM (Eastman et al., 2017), MIT License
  `https://github.com/openmm/openmm`

- DMFF (Wang et al., 2023), LGPL-3.0 license
  `https://github.com/deepmodeling/DMFF`

