# OpenReview forum: "Control Consistency Losses for Diffusion Bridges"
_ICML.cc/2026/Conference — ICML 2026 spotlight_

### Official Review · Reviewer_vnPs · 2026-03-03

**Soundness:** 4
**Presentation:** 4
**Significance:** 3
**Originality:** 3
**Overall Recommendation:** 5
**Confidence:** 5

**Summary:**

This paper proposes a novel framework for learning general diffusion bridges by leveraging the self-consistency property of the optimal control function. The primary motivation is to bypass the high computational cost associated with backpropagating through SDE trajectories, a common bottleneck in existing NGDB method. The authors derive a self-consistency condition that the optimal control must satisfy and translate this into a supervised learning objective that can be optimized online. The method's efficacy is evaluated across a range of tasks, including low-dimensional analytical models and high-dimensional molecular transitions, with performance compared against established diffusion-based baselines using both quantitative metrics and qualitative visualizations. Finally, a theoretical connections between the derived self-consistency properties and the ones used in diffusion generative models has been discussed.

**Compliance With Llm Reviewing Policy:**

Affirmed.

**Key Questions For Authors:**

I have some relatively minor questions for the authors:
1. Line 132: how does eq. (10) still hold when $F = \delta_{x_T}$, how is $\nabla \log F(x)$ defined then?
2. Theorem 3.1 (ii): it is stated that "... it forces the controlled dynamics to terminate at $X^u_T = x_T$",  which can satisfied "by construction". How is such construction done? by defining $u$ in limit?
3. Line 203 (right column): how are the requirements of enforced termination met when $b$ is of gradient form and $\sigma$ is independent of $x$?
4. Line 266: I believe it is a wrong reference to the non-enumerated equation after eq. (13)? (apparently there is no $\lambda$ in eq. (14)), it also applies for the one in Alg.1

**Limitations:**

Yes

**Strengths And Weaknesses:**

**Strengths**

The paper is theoretically sound and well-presented, offering a clear and motivated progression from the initial self-consistency property to the final objective function. The authors provide a detailed analysis regarding the regression target and neural parameterizations. Another strong contribution is the generalization used to address the instability of the Jacobian growth. The established theory is further bolstered by a comprehensive set of numerical experiments, and the results successfully validate the claims, showing the proposed method significantly improved the training speed while maintaining high performance. Finally, the work is well-organized, with an extensive appendix that provides sufficient context and additional depth to the results.

**Weaknesses**

Despite the high quality of the work, a few limitations warrant consideration. A primary theoretical limitation is the standing assumption of the elliptic diffusions, which may restrict the broader application of the method in computational chemistry, where the underdamping Langevin dynamics are standard (as noted in the paper). Furthermore, as demonstrated, the choice of $\alpha$ schedule has critical impact on the method's performance, even though in the conducted experiments in the paper, the authors show that the ``average prediction'' schedule outperforms others, while in general cases, a more robust or adaptive setting is preferred. Finally, the authors propose to use STL strategies to replace the original regression target as an expectation with a path-wise integral to reduce the variance. However, it remains unclear how well such a path-wise objective handles multimodal distributions, especially the potential modes are quite distinct, which would benefit from more rigorous verifications.

---

> ### Author Rebuttal · Authors · 2026-03-30
>
> We thank the reviewer for their detailed and positive review, and for their helpful comments! We address each of their questions below.
>
> **Q1 - Justifying eq (10) in singular case:**
> The presentation in Section 2 is intended to give a simple and intuitive derivation of the self-consistency property (SC1), which holds for a general reweighting function $F$ in (2). In this simple derivation, the lines (6)-(10) require differentiability of $F$, which of course would not hold in the limiting case $\delta_{x_T}$ corresponding to diffusion bridges, and consequently the expression (10) would not be well-defined in this singular case. However, by 'inverting' the steps at an earlier time $t<T$, we can obtain the self-consistency property (SC1) which crucially avoids the endpoint - by working with this relation instead, we can design an objective that is still well-defined even in the limiting case $F = \delta_{x_T}$.
>
> We derive (SC1) rigorously in the singular case $F = \delta_{x_T}$ in App D.1. To do so, one should take $\tilde{T}<T$, and apply the above result with $\tilde{T}$ in place of $T$ and with ${F(X_{\tilde{T}}) = p(X_T = x_T|X_{\tilde{T}})}$, which is smooth. We thus obtain the result for $s<t<\tilde{T}$, and as $\tilde{T}$ was arbitrary we see that the self-consistency property holds for $s<t<T$. We propose to add an extra sentence in the main paper clarifying these points in more detail, and to point to the complete derivation in the Appendix.
>
> **Q2 - Enforcing termination requirements by construction:**
> This constructive enforcement of the terminal condition is done in two ways. Firstly, we include a Brownian bridge guiding term $\frac{x_T - X_t}{T-t}$ in the parameterisation which strongly steers the samples towards the terminal point; the neural network then only learns the adjustment to this. The second way utilises the discretisation of the SDE - at the final discretisation step (at time $T-\delta$, say), we can jump directly to the endpoint $x_T$. In doing so, the control function $u(T-\delta, \cdot)$ is completely determined by construction, and the endpoint condition is directly enforced.
>
> Related to this point, the reviewer may also be interested in our response to Reviewer ymHm, in which we discuss theoretical properties related to this bridge parameterisation.
>
> **Q3 - Enforcing gradient-form requirements:**
> We agree that this point could be clarified. The result in Theorem 3.1 states that the control drift $u$ needs to be of gradient-form for some time $t$. As described above, at the final discretisation step we jump directly to the endpoint $x_T$, which determines the value of $u(T-\delta, \cdot)$ by construction. Specifically, it enforces that $u(T-\delta, x) = \frac{x_T - x}{\delta \sigma_{T-\delta}^2} - \tfrac{1}{\sigma^2_{T-\delta}} b(T-\delta, x)$. Note that the term $\tfrac{x_T - x}{\delta \sigma^2_{T-\delta}}$ is of gradient form. If $b$ is of gradient form, then so is $u$ at the final discretisation time $T-\delta$, thus satisfying the required condition of Theorem 2.1. One can also make suitable parameterisation adjustments to cover other cases.
>
> **Q4 - Incorrect equation number:**
> Thanks for spotting this! We'll update this in a revised version.
>
> **Ellipticity assumption:**
> We agree that this is a limitation of our approach, though we note it is a common assumption in the literature (e.g. [1]). We believe it may be possible to extend our self-consistency framework to the underdamped setting, though this would require several changes to the methodology. Investigating such extensions for applications in computational chemistry is a promising direction for future work (see the discussion in App E.7).
>
> **$\alpha$-schedule:**
> We investigate the choice of $\alpha$-schedule in Table 6 in Appendix E.7, and generally found the average schedule to perform reliably well across different settings. The choice of $\alpha$-schedule provides another tool to combat potential growth in the Jacobian; indeed, from the experiments in Table 6 we see that the differences are more pronounced in settings with large Jacobian terms.
>
> **Multimodal settings:**
> We agree that further investigation of performance in multimodal settings is an interesting direction. We found our method to perform well in finding both modes in the multimodal cell diffusion example (see Fig 6 in App E.4), however it found only one mode in the alanine dipeptide example (though we note that this is also the case for many methods designed specifically for TPS, e.g. see [2]). Future work could investigate combining our self-consistency methodology with enhanced sampling methods (similarly to works like [3]) to target multimodal settings specifically.
>
> We hope that these comments have addressed your questions and concerns. If you have any further questions, please let us know!
>
> [1] Heng et al 2021, https://arxiv.org/abs/2111.07243
>
> [2] Seong et al 2024, https://arxiv.org/abs/2405.19961
>
> [3] Nam et al 2025, https://arxiv.org/abs/2510.11923

---

> > ### Author Rebuttal · Reviewer_vnPs · 2026-04-03
> >
> > I appreciate the authors' thorough response to my comments. My concerns have been properly addressed, and I agree that the proposed revisions will strengthen the final paper. I maintain my positive score and recommend the paper for publication.

---

> > > ### Author Response · Authors · 2026-04-08
> > >
> > > We are glad to hear that our rebuttal has addressed your questions! We would like to thank the reviewer for their positive and helpful comments during the review process.

---

### Official Review · Reviewer_zM76 · 2026-03-11

**Soundness:** 2
**Presentation:** 3
**Significance:** 2
**Originality:** 2
**Overall Recommendation:** 4
**Confidence:** 5

**Summary:**

The paper presents a method to simulate diﬀusion processes conditioned on a terminal state (diﬀusion bridges). This problem traditionally is addressed using MCMC or neural approaches that learn the control drift of the conditioned dynamics. However, many existing neural methods rely on samples from the uncontrolled dynamics during training, which makes them unreliable, for example when the training samples do not adequately cover the region near the terminal position. To address this issue, the authors derive a self-consitency property of the optimal control of the conditioned diﬀusion. When combined with a terminal condition enforcing that trajectories reach the desired endpoint and with a gradient-form assumption on the control, this property uniquely characterizes the diﬀusion bridge dynamics.
Based on this, the authors propose the Control Consistency Diﬀusion Bridge (CCDB) algorithm, which learns the control drift by minimizing a family of self-consistency losses. The training of the control function is achieved in an online manner using the trajectories simulated from the current controlled process. Moreover, the proposed fixed-point style objective avoids backpropagation through simulated trajectories, which reduces computational cost and improves scalability. The authors also discuss suitable neural approximations of the drift that
enforce the bridging constraint and gradient structure by construction.
The authors show the improvements of their algorithm with respect to numerical methods that learn unconditional diﬀusions (SDB, FB, BEL), and provide a comparison with a more performant neural method (NGDB); diﬀerent parametric dynamics and more complex systems arising in computational chemistry are tested.

**Compliance With Llm Reviewing Policy:**

Affirmed.

**Key Questions For Authors:**

1. Is it possible to show a more specific convergence result/argument for the proposed fixed-point training procedure?
2. Are there scenarios in which the proposed method substantially improves the performance with respect to NGDB?
3. The generalized consistency formulation allows removing the explicit dependence on the Jacobian by introducing a suitable reference drift \tilde{b}. However, the performance of this
formulation may depend on the choice of \tilde{b}. How sensitive is the method to the choice of \tilde{b} in order to mitigate the instability caused by the Jacobian?
4. Are there regimes where SC1 performs better despite its potential instability?

**Limitations:**

Yes

**Strengths And Weaknesses:**

Strengths :
Soundness: The motivations and core ideas of the work are clearly articulated, the derivations are mathematically consistent and the results are established under reasonable assumptions. The proofs appear rigorous. The presented numerical experiments appear convincing.
Presentation: The paper is generally well written and well structured, the results are clearly stated. The comparison with related literature is extensively detailed. Various numerical methods are tested in the simulations and compared to the proposed algorithm showing some improvements and the limitations.
Significance: The training part based on the self-consistency loss avoid back propagation in simulated trajectories appears as a possible research direction.
Originality: The particular self-consistency identity derived for the diﬀusion bridge control, and its use to train the control without backpropagation through simulated trajectories, appear to be novel.

Weaknesses:
Soundness: Numerical tests: While the proposed approach is competitive with NGDB on several benchmarks, it does not consistently outperform this baseline. In some experiments the reported metrics are comparable or slightly worse than NGDB.
Significance: The paper addresses the well-known problem of simulation of diﬀusion processes under terminal condition. The contribution can be seen as a specific solution within this framework rather than a fundamentally new paradigm. The numerical improvements are not always evident. The main empirical advantage of the method appears to be the reduction in computational time rather than a clear improvement in approximation quality.

---

> ### Author Rebuttal · Authors · 2026-03-30
>
> We thank the reviewer for their detailed and positive review, and for their constructive comments which will help to strengthen our paper. We address each of their questions below.
>
> **Performance relative to NGDB:**
> We agree with the reviewer that the key advantage of our method is computational rather than necessarily improving solution quality, which we state in the introduction and at the start of Section 5. NGDB is a very strong baseline that requires full differentiation through the trajectory - our aim to to achieve comparable performance with this strong method without requiring differentiation through the trajectory (which provides significant computational benefits).
>
> The computational benefits of our approach over NGDB are demonstrated in Table 2, where the training is approximately three times faster. This performance gap increases further as the scale increases, as can be seen from the discussion in Appendix E.6.
>
> The results in Table 1 show that our method generally performs similarly to NGDB. Both of these methods significantly outperform the other approaches SDB, FB and BEL. When examining the obtained solutions, we observed that the methods generally performed similarly earlier in time, but the error of NGDB often increased relative to our method when approaching the terminal time $T$. We anticipate that this is due to the differences in the way the training objectives are constructed - our approach explicitly enforces the endpoint behaviour in the training objective, resulting in better accuracy near $T$. This was particularly clear in the OU example, which explains why our method significantly outperformed NGDB in the plots in Figure 3. We propose to add a section in the Appendix with plots to illustrate this effect.
>
> **Convergence properties:**
> We certainly agree that understanding the convergence of the fixed point training procedure is an important research direction. A convergence result would likely require analysing the contractivity behaviour of the fixed-point operator. The most tractable setting to analyse this would be in the linear setting $dX_t = -\alpha X_t + dB_t$ using the next-step prediction with time-step $\delta$; the fixed-point operator is then $(Tu)(s,x) = e^{-\alpha \delta} E[u(s+\delta, X_{s+\delta}^u) | X_s^u=x]$. A possible strategy could be to then use Gronwall's inequality to obtain a contraction under certain conditions.
>
> Making this approach rigorous and extending to the more general setting would be challenging, and thus we believe it is beyond the scope of the present paper but is certainly an exciting direction for future work. Indeed, such analysis would have impact beyond the diffusion bridge setting considered here, potentially also giving theoretical insights into related literature such as the adjoint-matching method for diffusion model finetuning (Domingo-Enrich et al. 2024).
>
> **Sensitivity to $\tilde{b}$:**
> The sensitivity to the choice of $\tilde{b}$ is mainly determined by the problem setting - in cases where the Jacobians are well-behaved there will be little difference, but if the Jacobians will become large (like in the double-well, Muller-Brown, or alanine dipeptide examples) there will be a larger change in performance (this effect can be seen in Tables 1, 6, 7 and in Figure 5).
>
> To give more detail on this, we include here some additional results that reflect the discussion above. We report results for $\tilde{b}$ interpolating from the SC1 to SC2 objectives, in the double well experiment for varying barrier heights $v$, and see that the sensitivity is greater for larger height.
>
> | | $v=2$ | $v=3$ | $v=4$ |
> |-|-|-|-|
> | $\tilde{b}=0$ (SC1) | 0.039$\pm$ 0.011 | 0.204$\pm$ 0.100  | 0.584$\pm$ 0.346 |
> | $\tilde{b}=-\tfrac{1}{3}b$ | 0.036$\pm$ 0.012 | 0.091$\pm$ 0.032 | 0.192$\pm$ 0.045 |
> | $\tilde{b}=-\tfrac{2}{3}b$ | 0.034$\pm$ 0.006 | 0.072$\pm$ 0.026 | 0.146$\pm$ 0.033 |
> | $\tilde{b}=-b$ (SC2) | 0.035$\pm$ 0.006 | 0.081$\pm$ 0.004  | 0.153$\pm$ 0.016 |
>
> **Differences between SC1 and SC2:**
> One can interpret SC2 as a more 'damped' self-consistency objective compared to SC1, as the running costs include the current control. SC1 gives more aggressive updates, which can accelerate training in well-behaved settings but at a possible cost to stability in settings with large Jacobians. Below we show the KL divergence to the ground truth during training in the OU experiment (mean over 5 runs). We see that SC1 does indeed display faster convergence to the solution than SC2, consistent with this intuition (we also observed similar behaviour in other examples with well-behaved Jacobians). We propose to add this discussion to the Appendix.
>
> | Training step | 200 | 400 | 600 | 800 | 1000 |
> |-|-|-|-|-|-|
> | SC1 | 0.12 | 0.078  | 0.067  | 0.052 | 0.046 |
> | SC2 | 0.15 | 0.088  | 0.070  | 0.61  | 0.055 |
>
> We hope that these comments have addressed your questions and concerns. If you have any further questions, please let us know!

---

> > ### Author Rebuttal · Reviewer_zM76 · 2026-04-02
> >
> > I am satisfied with the answers of the authors, and upgrade my note to accept.

---

> > > ### Author Response · Authors · 2026-04-08
> > >
> > > We are glad to hear that our rebuttal has addressed your questions! We would like to thank the reviewer for their positive and constructive comments during the review process, and we are delighted to hear you will raise your score.

---

### Official Review · Reviewer_zNHm · 2026-03-12

**Soundness:** 3
**Presentation:** 3
**Significance:** 2
**Originality:** 3
**Overall Recommendation:** 4
**Confidence:** 4

**Summary:**

This paper tackles the challenging problem of simulating "diffusion bridges". Specifically, it focuses on learning the optimal control $u(t, x)$ to steer a Stochastic Differential Equation (SDE) such that its terminal state $X_T$ maximizes an arbitrary reward function or matches a target $F(X_T)$. Traditional methods that learn the optimal control $u(t, x)$ to steer the SDE typically rely on computationally expensive Backpropagation Through Time (BPTT) across full simulated trajectories.To resolve this, the authors propose a novel algorithm based on a "self-consistency" property. Instead of simulating the full path, the algorithm simulates short forward chunks from time $s$ to time $t$. During this short step, it tracks the Jacobian process, $J_{t|s} = \nabla_{X_s}X_t$, which measures how sensitive the future state is to current perturbations. The authors mathematically prove that the optimal control must obey the self-consistency relation:
$u(s, x) = \mathbb{E} [J_{t|s} u(t, X_t) | X_s = x].$
The algorithm trains a neural network $u_\theta$ by minimizing the temporal difference between the current control and the Jacobian-projected future control (using a target network). Combined with a terminal boundary penalty forcing the trajectory toward $x_T$, the model learns the optimal drift iteratively online. Experiments on physical and biological tasks (OU, CIR, Double-Well, Cell Differentiation) demonstrate that this method matches the accuracy of state-of-the-art guided bridges while running 2x to 3x faster.

**Compliance With Llm Reviewing Policy:**

Affirmed.

**Final Justification:**

The authors’ rebuttal fully addressed my concerns. In particular, the additional clarification on Jacobian-related stability, implementation choices, and the positioning of the experimental scope was helpful, and I am satisfied with the response. I'll maintain my score accordingly.

**Key Questions For Authors:**

- In highly non-linear systems or over very long bridging horizons, the Jacobian $J_{t|s}$ can explode or suffer from extreme variance. Did you observe gradient clipping or specialized learning rate scheduling being strictly necessary in the Double-Well experiments as the barrier height increased?

- How do you propose stabilizing the self-consistency loss for highly chaotic SDEs? High-Dimensional Scalability: The current experiments cap out at relatively low dimensions. Does computing the Jacobian-vector product in your consistency loss scale linearly enough to apply this method to image-to-image translation (e.g., Schrödinger Bridges on CIFAR-10)?

**Limitations:**

Yes

**Strengths And Weaknesses:**

**Strength**

- The formulation of the self-consistency loss represents a significant breakthrough for training diffusion bridges. By framing the optimization as a Temporal Difference (TD) learning problem rather than a trajectory-level KL divergence, the authors completely eliminate the need for Backpropagation Through Time (BPTT) and adjoint sensitivity methods. The algorithm updates weights continuously ("online") during forward simulation, bypassing the memory bottlenecks and vanishing/exploding gradients typical of differentiating through SDE solvers. This directly translates to the impressive 2x-3x wall-clock speedups compared to the closest baseline (NGDB).

- The derivation of the self-consistency property using the Jacobian process is a highly elegant piece of stochastic calculus. Moving from Doob's h-transform to an actionable, forward-time consistency constraint is mathematically sound and conceptually novel in the context of continuous-time generative models.

**Weakness**

- The method relies heavily on simulating the Jacobian process $J_{t|s}$ forward in time. For highly chaotic dynamics, steep potentials (like high-barrier double wells), or long time horizons, the Jacobian can exhibit exponential growth or high variance, potentially destabilizing the consistency loss. While the authors briefly acknowledge this, the limitations of Jacobian-based losses in highly non-linear settings are not fully stress-tested.

- Lack of High-Dimensional Validation: All experiments (OU, CIR, Double-Well, Cell Diffusion) are strictly low-dimensional systems. While diffusion bridges are highly relevant in these physical sciences, modern machine learning heavily utilizes diffusion bridges for high-dimensional image generation and translation. It is unclear if tracking the Jacobian process scales favorably (in terms of memory and variance) to environments like image pixels or large protein backbones.

---

> ### Author Rebuttal · Authors · 2026-03-30
>
> We thank the reviewer for their detailed and positive review, and for their constructive comments. We address their points below.
>
> **Clarifying the methodology:**
> We note that our method does not explicitly track the Jacobians forwards in time over short chunks; rather, we solve for the training targets backwards along the full trajectories, and the effective time-intervals are determined by the choice of $\alpha$-schedule weighting. This backwards computation uses VJPs, so can be performed at a similar cost to the simulation of $X_t$ giving good scalability.
>
> **Behaviour of Jacobian:**
> The reviewer is correct that the behaviour of the Jacobian can affect performance, particularly in settings where the it gets large (e.g. passing over peaks on a potential). Consistent with this intuition, we see that SC2 outperforms SC1 on the double well, Muller-Brown (MB), and Alanine Dipeptide (ALDP) examples, but they perform similarly in the other experiments. Also consistent with this discussion, App E.3 shows that the performance gap between SC1 and SC2 increases with the barrier height in the double-well experiment. The $\alpha$-schedule can also be used to address the behaviour caused by possible Jacobian growth. Table 6 compares different schedules; again, we see that they perform comparably in most settings, but for the double-well the endpoint-schedule performs significantly worse.
>
> One way to interpret this is that SC2 is a more 'damped' objective compared to SC1; SC1 gives more aggressive updates, which can accelerate training in well-behaved settings but at a possible cost to stability in settings with large Jacobians (please also see our response to Reviewer zM76).
>
> We hope that the experiments in Tables 1,6,7 and Fig 5, and the additional results in the other rebuttal, help to elucidate these trade-offs. We propose to add a more detailed discussion in Section 5 regarding the considerations discussed here.
>
> **Implementation details, and stabilising training:**
> We did not observe gradient clipping, learning rate scheduling, or similar implementation details to have a major impact - by far the most effective change for stabilising training is to use SC2 rather than SC1. In this case, there is no problematic Jacobian behaviour (it is the identity, and we instead pick up 'running cost' terms).
>
> **Experimentation, and effect of dimension:**
> We have aligned our experimental setups with those used in the diffusion bridge literature that we consider [1,2,3], and we believe that our experimental evaluation compares well with this line of works. Moreover, our improved scalability allows us to explore settings not previously considered in these works, for example the MB and ALDP examples.
>
> We note that for diffusion bridges, challenges arises due to unstable behaviour along certain directions rather than from the dimension itself. In many examples, dimension can be large, but the underlying difficulty is more straightforward - e.g. ALDP has 63 dims, but the dynamics are largely governed by 2 Collective Variables (the dihedral angles). The MB potential, though low dimensional, is widely used as a benchmark in computational chemistry as it captures much of the difficulty of a more realistic higher-dim problem. We see this effect in our experiments; in App E.5 we also study a 100$d$ MB potential with quadratic terms in the additional 98 dims. The performance and training dynamics are largely unchanged from the 2$d$ case, demonstrating our method is robust to the dimension increase when the underlying 'difficulty' remains the same.
>
> We certainly agree that assessing the performance of our algorithm in problems with more complex instability behaviour is important, and this is something we intend to assess in more detail in future work (please see discussion below).
>
> **Extending to new experimental settings:**
> Given the improved scalability of our method, this opens the door to larger scale experiments not previously considered by the diffusion bridge literature. We believe that the most compelling higher-dimensional use-case is Transition Path Sampling (TPS).
>
> We have presented promising initial results in this direction in our ALDP example, but assessing in more challenging examples requires integrating implementation tricks and domain knowledge from the TPS literature. For example, this requires extending our methodology to underdamped dynamics, using improved initialisations, annealing schemes, sliding window intervals, and trust-regions (see App E.6 for a full discussion). Given these modifications, we believe that extending our method to more complex TPS examples warrants a separate work that builds upon the present paper, and we intend to pursue this in future work.
>
> We hope that these comments have addressed your questions and concerns. If you have any further questions, please let us know!
>
> [1] https://arxiv.org/abs/2111.07243
>
> [2] https://arxiv.org/abs/2407.15455
>
> [3] https://arxiv.org/abs/2502.11909

---

> > ### Author Rebuttal · Reviewer_zNHm · 2026-04-03
> >
> > The authors’ rebuttal fully addressed my concerns. In particular, the additional clarification on Jacobian-related stability, implementation choices, and the positioning of the experimental scope was helpful, and I am satisfied with the response. I'll maintain my score accordingly.

---

> > > ### Author Response · Authors · 2026-04-08
> > >
> > > We are glad to hear that our rebuttal has addressed your questions! We would like to thank the reviewer for their positive and constructive feedback in the review process.

---

### Official Review · Reviewer_ymHm · 2026-03-13

**Soundness:** 4
**Presentation:** 4
**Significance:** 3
**Originality:** 4
**Overall Recommendation:** 5
**Confidence:** 4

**Summary:**

The following summary presents my understanding of the main steps of the proposed method. The goals are to (i) verify correctness of my understanding and (ii) provide a self-contained account that might benefit readers.

The paper considers the problem of sampling diffusion bridges: given an unconditioned diffusion
$$\\mathrm{d}X_t = b(t,X_t)\\,\\mathrm{d}t + \\sigma(t,X_t)\\,\\mathrm{d}B_t, \\qquad X_0 = x_0,$$
with path measure $\\mathbb{P}$, simulate trajectories conditioned to hit a prescribed terminal state $x_T$ at time $T$. More generally, one can consider reweighting the terminal state by a function $F(X_T)$ (diffusion bridges correspond to the singular choice $F = \\delta_{x_T}$). By Doob's $h$-transform, the controlled SDE achieving this reweighting is
$$\\mathrm{d}X^u_t = \\big(b + \\sigma\\sigma^\\top u\\big)(t,X^u_t)\\,\\mathrm{d}t + \\sigma(t,X^u_t)\\,\\mathrm{d}B_t,$$
with optimal control
$$u^{\\ast}(t,x) = \\nabla_x \\log \\mathbb{E}\_{\\mathbb{P}}[F(X_T) \\mid X_t = x].$$

Most prior neural approaches learn $u^{\\ast}$ from samples of the unconditioned process $\\mathbb{P}$. Since the learned control is then deployed under the controlled dynamics $\\mathbb{P}^u$, training samples from $\\mathbb{P}$ may not adequately cover the relevant regions, especially for rare events where $x_T$ is unlikely under $\\mathbb{P}$. Training on trajectories from $\\mathbb{P}^u$ itself avoids this, but in the approach of Yang et al. (2025) (NGDB) this requires backpropagation through the simulated paths, which is costly.

This paper sidesteps both issues by deriving a self-consistency property of $u^{\\ast}$ that can be enforced as a fixed-point loss on trajectories simulated from the current control, without differentiating through them.

Starting from this expression for $u^{\\ast}$, one can rewrite it (using $\\nabla \\log f = \\nabla f / f$, interchanging $\\nabla_x$ and $\\mathbb{E}\_{\\mathbb{P}}$, and a conditional change-of-measure lemma (Lemma D.1, reported for completeness)) as
$$u^{\\ast}(t,x) = \\mathbb{E}\_{\\mathbb{Q}}\\big[J_{T|t}^\\top \\nabla \\log F(X_T) \\mid X_t = x\\big],$$
where $J_{t|s} = \\nabla_{X_s} X_t$ is the Jacobian of the base (not controlled) dynamics, i.e. the sensitivity of $X_t$ to perturbations of $X_s$. The interchange of $\\nabla_x$ and $\\mathbb{E}\_{\\mathbb{P}}$ can be made rigorous via the theory of stochastic flows. The Jacobian satisfies the semigroup property $J_{t|s} = J_{t|r}\\, J_{r|s}$ (the chain rule for the flow map). Using this together with the tower property, the expression can be "inverted" to earlier times, yielding the self-consistency property (SC1): for $0 < s < t < T$,
$$u^{\\ast}(s,x) = \\mathbb{E}\\big[J_{t|s}^\\top\\, u^{\\ast}(t, X_t^{u^{\\ast}}) \\mid X_s = x\\big].$$
This relates the control at time $s$ to its value at a later time $t$ along solution trajectories, and crucially avoids derivatives of $F$, making it suitable for diffusion bridges where $F = \\delta_{x_T}$ is not differentiable. Note that $J_{t|s}$ is the Jacobian of the base dynamics, while the trajectories $X_t^{u^{\\ast}}$ follow the controlled dynamics.

The Jacobian evolves according to the matrix SDE
$$\\mathrm{d}J_{t|s} = \\nabla b(t,X_t)\\,J_{t|s}\\,\\mathrm{d}t + \\nabla\\sigma(t,X_t)\\,J_{t|s}\\,\\mathrm{d}B_t, \\qquad J_{s|s} = \\mathrm{Id},$$
which can be seen heuristically by differentiating the Euler discretisation $X_{t+\\delta t} = X_t + b\\,\\delta t + \\sigma\\,\\delta B_t$ with respect to $X_s$: each step multiplies the current Jacobian by $\\mathrm{Id} + \\nabla b\\,\\delta t + \\nabla\\sigma\\,\\delta B_t$, and the continuous limit gives the SDE above.

The conditional expectation form of (SC1) directly motivates an MSE objective:
$$\\mathcal{L}(\\theta) = \\mathbb{E}\_{\\substack{s \\sim \\mathbf{U}\_{[0,T]} \\\\ t \\sim \\mathbf{U}\_{[s,T]}}}\\Big[\\big\\|u_\\theta(s, X_s^{\\bar\\theta}) - J_{t|s}^\\top\\, \\bar{u}(t, X_t^{\\bar\\theta})\\big\\|^2\\Big],$$
where $\\bar\\theta$ denotes the stop-gradient of the current parameters, and $J_{t|s}$ is computed along the trajectory $X^{\\bar\\theta}$ simulated under the current control.

Self-consistency alone is necessary but not sufficient: as expected we also need a boundary condition and, more unexpectedly, a gradient-form constraint. Theorem 3.1 shows that combining (i) SC1, (ii) a terminal condition forcing $X_T^u = x_T$, and (iii) $u$ being of gradient form at some time $t$ (i.e. $u(t,\\cdot) = \\nabla\\phi$ for a scalar $\\phi$), uniquely characterises the diffusion bridge. Self-consistency then propagates the gradient-form property to all times. In practice, when the base drift $b$ is of gradient form, the full drift is parameterised as $f_\\theta(t, X_t) = \\frac{x_T - X_t}{T - t} + \\sigma(t, X_t)\\,\\eta_\\theta(t, X_t)$, where the first term is a Brownian-bridge-like drift toward $x_T$ and $\\eta_\\theta$ is a neural correction; condition (ii) is enforced by jumping directly to $x_T$ at the final discretisation step (when $b$ is not of gradient form, $b$ is included explicitly in the parameterisation). Taking $\\eta_\\theta = \\nabla \\psi_\\theta$ for a scalar network $\\psi_\\theta$ enforces (iii), and (i) is enforced via the loss. The guiding term $\\frac{x_T - X_t}{T-t}$ is classical.

When the base Jacobian $J_{t|s}$ grows large (e.g. near energy barriers), the SC1 targets suffer from high variance.

To address this, the paper performs a Girsanov change of measure before pulling $\\nabla_x$ through $\\mathbb{E}\_{\\mathbb{P}}$, introducing an auxiliary process with drift $b + \\tilde{b}$ whose Jacobian $\\tilde{J}\_{t|s}$ can be kept well-behaved (e.g. taking $\\tilde{b} = -b$ gives a driftless auxiliary process with $\\tilde{J}\_{t|s} = \\mathrm{Id}$ when $\\sigma$ is state-independent), at the cost of additional correction terms in the self-consistency relation (SC2).

The paper considers $\\alpha$-schedules that control the temporal horizon over which self-consistency is enforced (instead of the simple target presented above). The training targets are defined as $\\mathcal{U}\_s := \\frac{1}{A_s}\\int_s^T \\alpha_t\\, J_{t|s}^\\top\\, \\bar{u}\_{\\bar\\theta}(t,X_t^{\\bar\\theta})\\,\\mathrm{d}t$ with $A_s = \\int_s^T \\alpha_t\\,\\mathrm{d}t$, and different choices of $\\alpha$ interpolate between next-step prediction ($\\alpha$ concentrated at $s+\\delta t$), average prediction (uniform $\\alpha$), and endpoint prediction ($\\alpha$ concentrated at $T - \\delta t$), trading target variance against propagation speed of the terminal information.

**Compliance With Llm Reviewing Policy:**

Affirmed.

**Key Questions For Authors:**

1. Can you give some intuition on what the gradient-form condition (iii) in Theorem 3.1 precisely rules out? Self-consistency + bridging alone are not sufficient: what kind of non-gradient controls satisfy (i) and (ii) but are not the diffusion bridge?

2. The parameterisation $f_\\theta(t, X_t) = \\frac{x_T - X_t}{T - t} + \\sigma(t, X_t)\\,\\eta_\\theta(t, X_t)$ absorbs the $1/(T-t)$ singularity into the deterministic guiding term. Does this imply that a bounded (or Lipschitz) neural network $\\eta_\\theta$ now suffices to represent the controlled drift, whereas without the guiding term one would need the network to learn the singularity?

3. Theorem 3.1 guarantees uniqueness of the fixed point, but the stop-gradient training iteration is not guaranteed to converge to it. Are there settings (e.g. linear dynamics) where the iteration can be shown to be contractive? More broadly, what research directions do you see toward establishing convergence guarantees for self-consistency training, possibly drawing on the TD-learning convergence literature?

**Limitations:**

yes

**Strengths And Weaknesses:**

## Strengths

**Clear derivation**: The paper presents a clean chain of derivations from the Doob $h$-transform to the self-consistency property, the training loss, and the uniqueness result. I checked the main derivations in the text and the key proofs in the appendix and did not spot errors.

**Principled method**: The approach is well-motivated: self-consistency gives a fixed-point characterisation of the optimal control that can be enforced online on trajectories from the current policy, avoiding both the coverage issues of training on unconditioned samples and the cost of backpropagating through trajectories. The uniqueness result (Theorem 3.1) provides a clear theoretical justification, and the generalised SC2 variant is a natural remedy for Jacobian growth.

**Broad experimental coverage**: The paper tests on a good range of settings (OU, CIR, double-well, cell diffusion, Muller-Brown, alanine dipeptide), with comparisons to several baselines. Training is reported to be roughly 3x faster than NGDB. That said, the experimental settings are standard benchmarks from the diffusion bridge literature. Recent competing methods (TPS-DPS, Seong et al. 2025; Doob's Lagrangian, Du et al. 2024) have been applied to more challenging molecular systems. A demonstration on a system of comparable complexity would make the empirical contribution more compelling.

## Minor Weaknesses

**Presentation of Section 4**: The generalised self-consistency (SC2) in Section 4 reads a bit as if it was discovered after the main framework was already developed and then appended. Integrating the discussion of Jacobian growth and the motivation for SC2 earlier would make the narrative more cohesive.

**Semigroup property**: The paper states the semigroup property $J_{t|s} = J_{t|r}\\,J_{r|s}$ without justification. It would be worth noting explicitly that this follows from the chain rule applied to the flow map $X_s \\mapsto X_t$.

**Singularity in the derivation**: While the final self-consistency property (SC1) avoids derivatives of $F = \\delta_{x_T}$, the intermediate steps of the derivation (specifically eq. (6)-(10)) do pass through $\\nabla \\log F(X_T)$, which is ill-defined for $F = \\delta_{x_T}$. The paper handles this in Appendix D.1. Explicitly mentioning this point and providing the intuition for its solution would improve the presentation. Numerically, the singularity manifests through the drift blowing up as $t \\to T$.

**$\\alpha$-schedules introduced without motivation**: The paper jumps directly to the general $\\alpha$-weighted targets $\\mathcal{U}\_s$ without first presenting a simpler target (such as the one mentioned in the Summary) for the loss or explaining upfront why weighting over temporal horizons is beneficial.

**Solving for training targets**: The backward recursion for computing $\\mathcal{U}\_s$ along simulated trajectories (Section 3.1 and Appendix B.2) is quite dense and could benefit from a clearer step-by-step presentation.

**No convergence analysis for the fixed-point iteration**: Theorem 3.1 establishes that the fixed point of the self-consistency loss is unique (given bridging and gradient form), but the paper provides no analysis of whether the stop-gradient training iteration converges to it. The Limitations section acknowledges that "the fixed-point nature of our algorithm may be less stable than the KL-based objective used in NGDB," but this is left as an open concern. This seems analogous to the instability of TD learning with nonlinear function approximation in RL, and deserves more discussion.

**Relationship between CCDB and NGDB losses**: The paper frames its contribution as avoiding backpropagation through trajectories, but does not explain why a stop-gradient approach cannot simply be applied to NGDB's KL objective to achieve the same benefit. Clarifying the structural reason that the self-consistency formulation enables stop-gradient training, whereas NGDB's KL objective does not (see Yang et al. 2025 for details of the NGDB loss), would strengthen the positioning of the contribution.

## References

- Du et al. (2024), "Doob's Lagrangian: A sample-efficient variational approach to transition path sampling"
- Seong et al. (2025), "Transition path sampling with improved off-policy training of diffusion path samplers"
- Yang et al. (2025), "Neural guided diffusion bridges"

---

> ### Author Rebuttal · Authors · 2026-03-30
>
> We thank the reviewer for their detailed and positive review, and for their helpful comments! We address each of their questions below.
>
> **Presentation of Section 4:**
> We agree that discussing the motivation for SC2 earlier would make its introduction more natural. Following your suggestion, we will add a sentence in the introduction, to mention the possible growth of Jacobian in certain settings, and that we will introduce a second objective SC2 to help address this.
>
> **Semigroup property:**
> We agree that mentioning why this holds would be useful, and we will add a sentence explaining that this is a consequence of the chain rule when we introduce the Jacobian object.
>
> **Singularity in the derivation:**
> We will add a sentence to the main paper derivation clarifying this point, with a brief discussion of the full derivation that is the Appendix.
>
> **$\alpha$-schedule:**
> We propose to add a sentence before Eq (12) explaining the intuition behind the  $\alpha$-schedule, to make its introduction more natural. Namely, the self-consistency properties hold for any $s<t,$ and the choice of $\alpha$ determines how 'far into the future' the algorithm attempts to enforce this property.
>
> **Solving for training targets:**
> We appreciate that the presentation of the backwards recursions is a little dense. To give an alternative  interpretation, we propose to write out the continuous dynamics of the training targets. If we let $I_s = \int_s^T J_{t|s}^\top u(t, X_t^u) dt$, then $dI_s = - \alpha_s u(x, X_s^u) ds - \nabla b(s, X_s^u)^\top I_s ds$. The recursions written in the paper correspond to a discretisation of these dynamics backwards in time. We hope this interpretation can provide more intuition for these expressions.
>
> **Convergence analysis, and possible directions to investigate this:**
> We agree that understanding the convergence of the fixed point iterations is an important research direction. There certainly may be useful ideas from the TD learning literature (though we note our approach would correspond to the gradient of the value function, rather than the value function itself). Please see our response to Reviewer zM76 for some of our thoughts regarding possible directions for studying this.
>
> **Relationship to NGDB:**
> The NGDB method uses the loss function $L(\theta) = \mathbb{E} [ \int_0^T \tfrac{1}{2} \lVert v_\theta(s, X_s^\theta) \rVert^2 - G(s, X_s^\theta) ds]$, where $v_\theta$ is a neural correction to a chosen guiding drift, and $G$ is a function that depends on the choice of guiding drift. As you suggest, one could place a stop-gradient on the $X_s^\theta$ in this loss to avoid differentiation through the trajectories. However, if one were to do this then the loss function would just encourage the correction term $v_\theta$ to collapse to zero along the simulated trajectories, and thus it would not learn the correct dynamics. It is therefore not clear how one would modify the NGDB loss to avoid differentiation through the trajectory; indeed, this is noted by the NGDB authors in their conclusion as an important direction for future research. We propose to add this discussion to the Appendix to help clarify the distinction between the methods.
>
> **Gradient condition:**
> The gradient condition rules out rotational components in the control drift. While the true control is gradient-form, it is possible for a self-consistent $u$ to include rotational terms. Enforcing the gradient condition at a single time $t$ is enough to prevent such terms. We will add a sentence clarifying this intuition after Theorem 3.1.
>
> **Lipschitzness of network under parameterisation:** This is a very interesting question, and we thank the reviewer for raising it. Indeed, we can show the following: Define $\eta(t,x) := \sigma^{-1}(t,x) \left (f(t,x) - \frac{x_T - x}{T-t} \right)$, where $f=b + (\sigma \sigma^\top)u$ is the full drift of the conditioned SDE. Under mild regularity and boundedness assumptions on $b$ and $\sigma$, the correction $\eta$ is indeed locally bounded and Lipschitz.
>
> *Proof sketch:* The drift $u$ is related to the log gradient of the transition density for the base SDE (by Doob's $h$-transform). By considering the EM discretisation at the last step (from $T - \delta t$ to $T$), the transition kernel is (conditionally) Gaussian, and we can calculation the coefficients in the asymptotic expansion as $\delta t \rightarrow 0$. Defining $\eta$ in this manner leads to cancellations and shows that $\eta = \mathcal{O}(1)$ as $\delta t \rightarrow 0$. On a more rigorous level, one can appeal to short-time asymptotics of heat kernels (e.g. Chen et al, 'Logarithmic heat kernel estimates without curvature restrictions').
>
> We will include a carefully stated proposition and proof, as this result provides a strong theoretical justification for our parameterisation.
>
> We hope that these comments have addressed your questions and concerns. If you have any further questions, please let us know!

---

> > ### Author Rebuttal · Reviewer_ymHm · 2026-04-02
> >
> > I thank the authors for their detailed and thoughtful responses to my questions and remarks.
> > I have no further questions, as all the points I raised have been satisfactorily addressed.
> >
> > The proposed revisions will strengthen the final manuscript, in particular through the addition of new results on the local Lipschitz continuity of the neural network under the proposed parametrization. I also believe that the clarified discussion of training targets will improve the presentation and enhance the overall readability of the paper.

---

> > > ### Author Response · Authors · 2026-04-08
> > >
> > > We are glad to hear that our rebuttal has addressed your questions! We would like to thank the reviewer for their positive and helpful comments during the review process.

---

### Decision · Program_Chairs · 2026-04-30

**Decision:**

Accept (spotlight)

**Comment:**

This paper derives a set of control consistency losses for diffusion bridges and demonstrates competitive empirical results on a number of test systems while maintaining high computational efficiency relative to methods that require back-propagating through trajectories. The reviews emphasize the clarity of the exposition and the theoretical and computational advantages of the approach.